# CHANNEL-AWARE CONTRASTIVE CONDITIONAL DIFFUSION FOR MULTIVARIATE PROBABILISTIC TIME SERIES FORECASTING

## ABSTRACT

Forecasting faithful trajectories of multivariate time series from practical scopes is essential for reasonable decision-making. Recent methods majorly tailor generative conditional diffusion models to estimate the target temporal predictive distribution. However, it remains an obstacle to enhance the exploitation efficiency of given implicit temporal predictive information to bolster conditional diffusion learning. To this end, we propose a generic channel-aware Contrastive Conditional Diffusion model entitled **CCDM** to achieve desirable Multivariate probabilistic forecasting, obviating the need for curated temporal conditioning inductive biases. In detail, we first design a channel-centric conditional denoising network to manage intra-variate variations and cross-variate correlations, which can lead to scalability on diverse prediction horizons and channel numbers. Then, we devise an ad-hoc denoising-based temporal contrastive learning to explicitly amplify the predictive mutual information between past observations and future forecasts. It can coherently complement naive step-wise denoising diffusion training and improve the forecasting accuracy and generality on unknown test time series. Besides, we offer theoretic insights on the benefits of such auxiliary contrastive training refinement from both neural mutual information and temporal distribution generalization aspects. The proposed CCDM can exhibit superior forecasting capability compared to current state-of-the-art diffusion forecasters over a comprehensive benchmark, with best MSE and CRPS outcomes on 79.17% and 87.5% cases. Our code is publicly available at `https://github.com/anonymous/CCDM`.

## 1 INTRODUCTION

Multivariate probabilistic time series forecasting aims to quantify the stochastic temporal evolutions of multiple continuous variables and benefit decision-making in various engineering fields, such as weather prediction (Li et al., 2024), renewable energy dispatch (Dumas et al., 2022), traffic planning (Huang et al., 2023) and financial trading (Gao et al., 2024). Modern methods majorly customize time series generative models (Salinas et al., 2019; Li et al., 2022; Yoon et al., 2019; Rasul et al., 2020) and produce diverse plausible trajectories to decipher the intricate temporal predictive distribution which is conditioned on past observations. Due to the excellent mode coverage capacity and training stability of diffusion models (Song et al., 2020; Ho et al., 2020), a flurry of conditional diffusion forecasters (Lin et al., 2023; Yang et al., 2024) are recently developed by designing effective temporal conditioning mechanisms to discover informative patterns from historical time series.

Despite existing advances, current time series diffusion models still struggle to learn a precise and generalizable multivariate predictive distribution on challenging prediction tasks. The first barrier is *how to design an effective conditional denoising network* to account for multivariate temporal correlations in provided observations as well as varying degrees of noise imposed on target sequences. To address this denoiser architectural issue, CSDI (Tashiro et al., 2021) and TMDM (Li et al., 2023) employ spatiotemporal attention modules to characterize intra-channel and inter-channel [1] relations. SSSD (Alcaraz & Strodthoff, 2022) and LDT (Feng et al., 2024) utilize structured state space and latent diffusion models to handle high-dimensional time series more efficiently. However, existing

---

[1] A channel shares the same meaning with a variate, with each channel indicating a univariate time series.

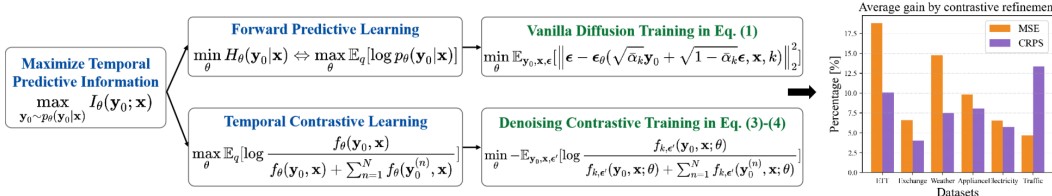

Figure 1: The schematic of the devised information-theoretic denoising-based contrastive diffusion learning. The bar chart depicts the average gains by contrastive refinement on six real-world datasets.

time series denoisers fall short in identifying the faithful channel-specific and cross-channel properties with the step-wise noise disturbing each variate sequences. Improper treatment for diffused noise can cause training instability and hurt the capacity to tackle long-term dependencies and complex inter-channel correlations. Inspired by recent success of channel-centric structure in long-term multivariate forecasting (Chen et al., 2024a; Liu et al., 2023), we propose a *composite channel-aware manipulation strategy* to design the conditional denoising network, which can cope with the side effect of noise corruption and recover the plausible heterogeneous temporal correlations.

The second barrier is *how to enhance the exploitation efficiency of the implicit temporal predictive information* hidden in limited historical time series. It has been revealed that learning to unveil the useful temporal patterns like decomposed modes (Deng et al., 2024) or spectral biases (Crabbé et al., 2024) in collected dataset can boost the diversity and accuracy of generated profiles. But diffusion forecasters fail to fully unleash the intrinsic predictive information merely by naive noise regression training. To this end, existing works propose *auxiliary diffusion training strategies* to amplify the helpful temporal features for better prediction quality. In particular, they employ specific time series inductive biases to promote temporal conditioning schemes or guide iterative inference procedures. Pretraining conditional encoders by deterministic point prediction (Shen & Kwok, 2023; Li et al., 2023) is a viable method, which produces more accurate medians and sharper prediction intervals. Coupling unique temporal features like multi-granularity dynamics (Fan et al., 2024; Shen et al., 2023) or target quantitative metrics (Kollovieh et al., 2024) to regularize the sequential diffusion process can also steer the reverse generation process towards plausible trajectories. However, these auxiliary refinements need to expose prior knowledge on task-specific temporal properties and tailor ad-hoc regulations for diffusion training and sampling. They are not consistent with standard step-wise temporal denoising learning and a generic way to improve time series diffusion models.

Motivated by a neural information view in (Tsai et al., 2020), naive conditional time series diffusion learning can be deemed as a *forward predictive way* to maximize the temporal mutual information between past observations and target forecasts. Above auxiliary learning methods can empirically enrich the predictive temporal information. However, single noise prediction training is inadequate to reveal the entire task-specific information. In light of the composite objective integrating contrastive learning to procure more robust task-related representations (Tsai et al., 2020), we propose to further enhance the *prediction-related mutual information* captured by denoising diffusion in a *complementary contrastive way*, where both positive and negative time series are inspected at each diffusion step. We illustrate such temporal contrastive refinement on conditional diffusion forecasting in Fig. 1, which mitigates over-fitting and attains better generality on unknown test data.

In this work, we propose a contrastive conditional diffusion model termed CCDM which can explicitly maximize the predictive mutual information for multivariate probabilistic forecasting. The efficient *channel-aware denoiser architecture* and complementary *denoising-based contrastive refinement* are two recipes to boost diffusion forecasting capacity. Our *main contributions* are summarized as: (1) We design a composite channel-aware conditional denoising network, which merges channel-independent dense encoders to extract univariate dynamics and channel-wise diffusion transformers to aggregate cross-variate correlations. It gives rise to efficient iterative inference and better scalability on various channel numbers and prediction horizons. (2) We propose to explicitly amplify the predictive information between generated forecasts and past observations via a coherent denoising-based temporal contrastive learning, which can be seamlessly aligned with vanilla step-wise denoising diffusion training and thus efficient to implement. (3) Extensive simulations validate the superior forecasting capability of CCDM. It can attain better accuracy and reliability versus other excellent models on various forecasting settings, especially for long-term and large-channel scenarios.

## 2 PRELIMINARIES

### 2.1 PROBLEM FORMULATION

In this paper, we look into the task of multivariate probabilistic time series forecasting. Given the past observation $\mathbf{x} \in \mathbb{R}^{L \times D}$ as conditioning time series, the goal is to generate a group of $S$ plausible forecasts $\{\hat{\mathbf{y}}_0^{(s)} \in \mathbb{R}^{H \times D}\}_{s=1}^S$ from the learned conditional predictive distribution $p_\theta(\mathbf{y}_0|\mathbf{x})$. Here, $D$ is the number of channels, $L$ and $H$ indicate the lookback window length and prediction horizon respectively. $\theta$ stands for the parameters of a conditional diffusion forecaster which represents the real predictive distribution $q(\mathbf{y}_0|\mathbf{x})$. We allocate diverse values to horizon $H$ and channel number $D$ to construct a holistic benchmark which can completely evaluate the capability of different conditional diffusion models on various forecasting scenarios.

### 2.2 CONDITIONAL DENOISING DIFFUSION MODELS

Conditional diffusion models have exhibited impressive capability on a wide variety of controllable multi-modal synthesis tasks (Chen et al., 2024b). It dictates a bi-directional distribution transport process between raw data $\mathbf{y}_0$ and prior Gaussian noise $\mathbf{y}_K \in \mathcal{N}(\mathbf{0}, \mathbf{I})$ via $K$ diffusion steps. The forward process gradually degrades clean $\mathbf{y}_0$ to fully noisy $\mathbf{y}_K$ and can be fixed as a Markov chain: $q(\mathbf{y}_{0:K}) = q(\mathbf{y}_0) \prod_{k=1}^K q(\mathbf{y}_k|\mathbf{y}_{k-1})$, where $q(\mathbf{y}_k|\mathbf{y}_{k-1}) := \mathcal{N}(\mathbf{y}_k; \sqrt{1-\beta_k}\mathbf{y}_{k-1}, \beta_k \mathbf{I})$ and $\beta_k$ is the degree of imposed step-wise Gaussian noise. We can accelerate the forward sampling procedure and obtain closed-form latent state $\mathbf{y}_k$ at arbitrary step $k$ by a noteworthy property (Ho et al., 2020): $\mathbf{y}_k = \sqrt{\bar{\alpha}_k}\mathbf{y}_0 + \sqrt{1-\bar{\alpha}_k}\boldsymbol{\epsilon}$, where $\bar{\alpha}_k := \prod_{s=1}^k (1-\beta_s)$ and $\boldsymbol{\epsilon} \sim \mathcal{N}(\mathbf{0}, \mathbf{I})$. The reverse generation process converts known Gaussian to realistic prediction data $\mathbf{y}_0$ given input conditions $\mathbf{x}$, which can be cast as a parameterized Markov chain: $p_\theta(\mathbf{y}_{0:K}|\mathbf{x}) = p(\mathbf{y}_K) \prod_{k=K}^1 p_\theta(\mathbf{y}_{k-1}|\mathbf{y}_k, \mathbf{x})$. The overall training objective can be simplified as minimizing the step-wise denoising loss below:

$$\mathcal{L}_k^{denoise} = \mathbb{E}_{\mathbf{y}_0, \mathbf{x}, \boldsymbol{\epsilon}}[\left\| \boldsymbol{\epsilon} - \boldsymbol{\epsilon}_\theta(\sqrt{\bar{\alpha}_k}\mathbf{y}_0 + \sqrt{1-\bar{\alpha}_k}\boldsymbol{\epsilon}, \mathbf{x}, k) \right\|_2^2]. \tag{1}$$

A potential issue for current conditional diffusion models lies in forging an effective conditioning mechanism that can enhance the alignment between given conditions $\mathbf{x}$ and produced data $\mathbf{y}_0$, like the coherent semantics between textual descriptions and visual renderings (Esser et al., 2024), or the conformity of generated vehicle motions to scenario constraints (Jiang et al., 2023). However, such data consistency is hard to represent for temporal conditional probability modeling. We thus explicitly learn to amplify the prediction-related temporal information conveyed from past conditioning time series to generated trajectories. Such predictive mutual information can reflect underlying temporal properties in historical sequences, to which the produced forecasts should comply.

### 2.3 NEURAL MUTUAL INFORMATION MAXIMIZATION

As discussed above, to more efficiently represent the useful predictive modes involved in conditioning time series, we choose to explicitly maximize the prediction-oriented mutual information when learning the conditional diffusion forecaster. Learning to maximize mutual information is effective to boost the consistency between two associated variables (Song & Ermon, 2019), which has been actively applied to self-supervised learning (Liang et al., 2024b) and multi-modal alignment (Liang et al., 2024a). Regarding conditional diffusion learning, there also exist several related works (Wang et al., 2023; Zhu et al., 2022) which explicitly employ mutual information maximization to enhance high-level semantic coherence between input prompts and generated samples. While we propose a complementary way to equip the conditional diffusion forecaster with this tool to bolster the utilization of informative temporal patterns. Besides, we provide a distinct composite loss design and more concrete interpretations on the benefits of the contrastive scheme to ordinary conditional diffusion.

Among the two practical methods to maximize the intractable mutual information (Tsai et al., 2020), contrastive learning aids to strengthen the association by discriminating intra-class from inter-class samples. Contrastive predictive coding (Oord et al., 2018) realizes such objective by optimizing the contrastive lower bound with low variance via the prevalent InfoNCE loss:

$$\mathcal{L}_{InfoNCE} = -\mathbb{E}_{(\mathbf{y}_0, \mathbf{x}) \sim q(\mathbf{y}_0, \mathbf{x}), \mathbf{y}_0^{(n)} \sim q^n(\mathbf{y}_0)}[\log \frac{f(\mathbf{y}_0, \mathbf{x})}{f(\mathbf{y}_0, \mathbf{x}) + \sum_{n=1}^N f(\mathbf{y}_0^{(n)}, \mathbf{x})}]. \tag{2}$$

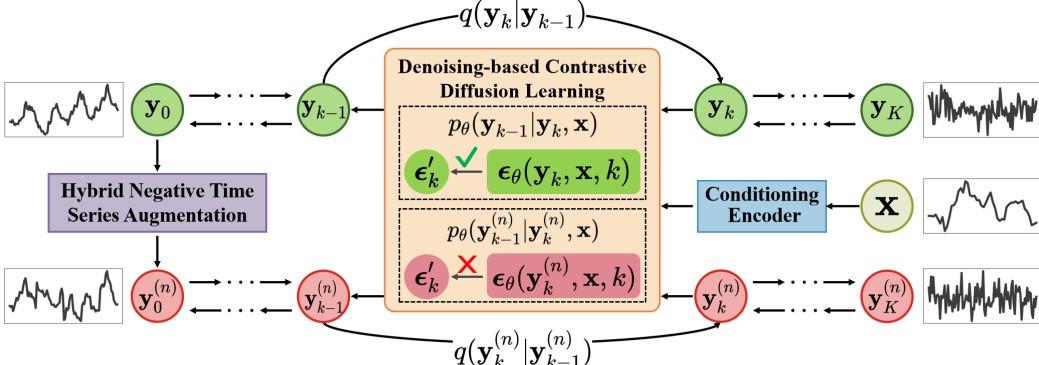

Figure 2: The framework of denoising-based contrastive conditional diffusion forecaster. Detailed negative time series construction methods are clarified in Appendix A.9.1.

During each iteration, we create a set of $N$ negative samples via the negative construction operation $q^n(\mathbf{y}_0)$ on positive data $\mathbf{y}_0$. $f(\mathbf{y}_0, \mathbf{x})$ accounts for the density ratio $\frac{q(\mathbf{y}_0|\mathbf{x})}{q(\mathbf{y}_0)}$ and can be *any types of positive real functions*. This flexible form of the density ratio function offers a natural initiative of the following denoising-based contrastive conditional diffusion.

*Forward predictive learning* is another way to boost the inter-dependency by fully reconstructing target $\mathbf{y}_0$ conditioned on given $\mathbf{x}$. This reconstruction learning can be realized by learning a deterministic mapping or conditional generative model from $\mathbf{x}$ to $\mathbf{y}_0$. As $I(\mathbf{y}_0; \mathbf{x}) = H(\mathbf{y}_0) - H(\mathbf{y}_0|\mathbf{x})$, and $H(\mathbf{y}_0)$ is irrelevant to discovering the entanglement between $\mathbf{x}$ and $\mathbf{y}_0$, thereby maximizing $I(\mathbf{x}; \mathbf{y}_0)$ boils down to optimizing the predictive lower bound $-H(\mathbf{y}_0|\mathbf{x}) = \mathbb{E}_{q(\mathbf{x},\mathbf{y}_0)}[\log p_\theta(\mathbf{y}_0|\mathbf{x})]$, which is aligned with the likelihood-based objective of naive conditional diffusion learning. (Tsai et al., 2020) claims that combining both predictive and contrastive learning tactics can significantly raise the quality of obtained task-related features. Accordingly, we equip vanilla conditional time series diffusion with a denoising-based InfoNCE contrastive loss to further boost the temporal predictive information between past conditions and future forecasts. A concise motivation of this information-theoretic contrastive diffusion forecasting is depicted in Fig. 1.

## 3 METHOD: CHANNEL-AWARE CONTRASTIVE CONDITIONAL DIFFUSION

In this section, we elucidate two innovations of the tailored CCDM for generative multivariate time series forecasting, including the hybrid channel-aware denoiser architecture depicted in Fig. 3 and denoising-based contrastive diffusion learning demonstrated in Fig. 2.

### 3.1 CHANNEL-AWARE CONDITIONAL DENOISING NETWORK

Recent progress on multivariate prediction methods (Liu et al., 2023; Ilbert et al., 2024) show that proper integration of channel management strategies in time series backbones is critical to discover univariate dynamics and cross-variate correlations. But this problem has not been well explored in multivariate diffusion forecasting and previous conditional denoiser structures do not obviously distinguish such heterogeneous channel-centric temporal properties. To this end, we design a channel-aware conditional denoising network which incorporates composite channel manipulation modules, i.e. channel-independent dense encoders and channel-mixing diffusion transformers. This architecture can efficiently represent intra-variate and inter-variate temporal correlations in past conditioning $\mathbf{x}$ and future predicted $\mathbf{y}_0$ under different noise levels, as well as being robust to diverse prediction horizons and channel numbers.

**Channel-independent dense encoders.** We develop two channel-independent MLP encoders to extract unique temporal variations in each individual channel of observed condition $\mathbf{x}$ and corrupted latent state $\mathbf{y}_k$ at each diffusion step. The core ingredient in latent and condition encoders is the channel-independent dense module (CiDM) borrowed from TiDE (Das et al., 2023a), which stands out as a potent MLP building-block for universal time series analysis models (Das et al., 2023b). A salient element in CiDM is the skip-connecting MLP residual block which can improve temporal pattern expressivity. The $D$ linear layers in parallel are shared and used for separate channel feature

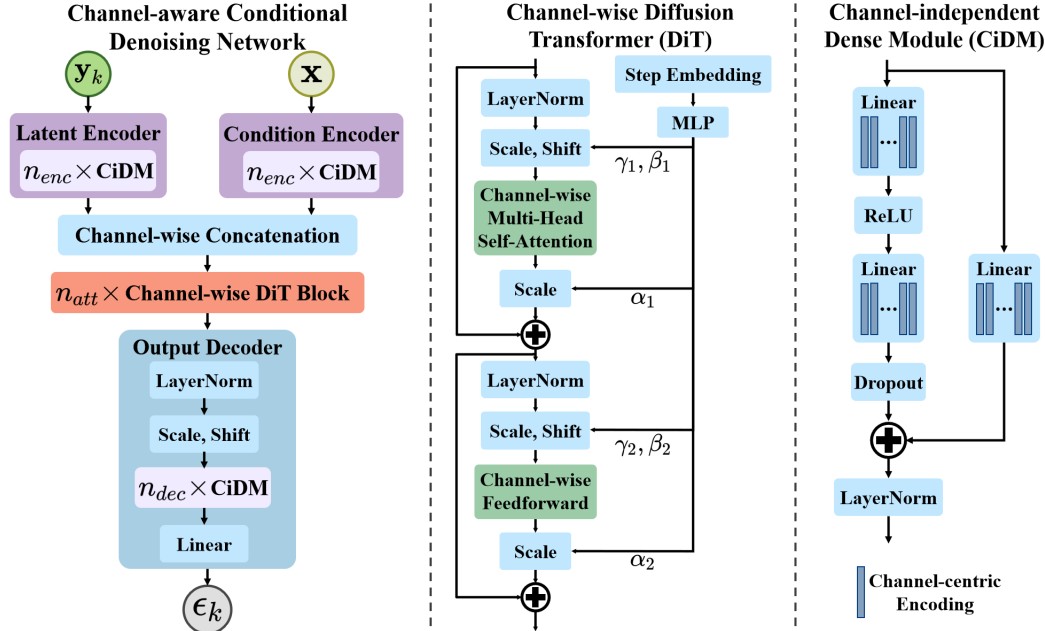

Figure 3: The diagram of channel-aware conditional denoiser architecture. Left: the whole network. Middle: channel-mixing DiT blocks. Right: channel-independent MLP dense modules.

embedding. We stack $n_{enc}$ CiDM modules of $e_{hid}$ hidden dimension to transform both $\mathbf{x}$ and $\mathbf{y}_k$ into $e_{hid} \times D$ size. These two input encoders can be easily adjusted to accommodate different context windows and hidden feature dimensions.

**Channel-wise diffusion transformers.** To regress step-wise Gaussian noise $\boldsymbol{\epsilon}_k$ on raw $\mathbf{y}_0$ more precisely, we should fully exploit implicit temporal information in pure conditioning $\mathbf{x}$ and polluted target $\mathbf{y}_k$. We concatenate the latent encoding of $\mathbf{x}$ and $\mathbf{y}_k$ along the channel axis and then leverage $n_{att}$-depth channel-wise diffusion transformer (DiT) blocks to aggregate heterogeneous temporal modes from various channels. DiT is an emergent diffusion backbone for open-ended text-to-image synthesis which merits eminent efficiency, scalability and robustness (Peebles & Xie, 2023; Esser et al., 2024). Two critical components in DiT are multi-head self-attention for feature fusion and adaptive layer norm (adaLN) layers to absorb other conditioning items (e.g. diffusion step embedding, text labels) as learnable scale and shift parameters. Although DiT has been repurposed by TimeDiT (Cao et al., 2024) and LDT (Feng et al., 2024) to model the multivariate predictive distribution, our adapted channel-centric DiT module differs from them in two ways. First, we switch the point-wise attention over the time dimension to a channel-wise attention along the variate axis, which can represent cross-channel correlations in $\mathbf{x}$ and $\mathbf{y}_k$ beyond temporal dependencies. Second, to improve time series denoising learning, we directly concatenate the conditioning $\mathbf{x}$ with corrupted $\mathbf{y}_k$ and capture their temporal features by attention, which can fully utilize the useful predictive patterns in limited historic observations. Whereas TimeDiT and LDT simply pass the given $\mathbf{x}$ to adaLN layers, which may cause the predictive information loss. We analyze the impact of these two unique structure designs in Appendix A.10. Afterwards, we develop an output decoder with $n_{dec}$ CiDMs plus a last adaLN to yield the prediction of imposed noise $\boldsymbol{\epsilon}_k$ given $\mathbf{x}$ and $\mathbf{y}_k$.

## 3.2 DENOISING-BASED TEMPORAL CONTRASTIVE REFINEMENT

Unlike previous empirically designed temporal conditioning schemes to make better exploitation of past predictive information, we instead propose to explicitly maximize the prediction-related mutual information $I(\mathbf{y}_0; \mathbf{x})$ between past observations $\mathbf{x}$ and future forecasts $\mathbf{y}_0$ via an adapted denoising-based contrastive strategy. We will employ the learnable denoising network $\boldsymbol{\epsilon}_\theta(\cdot)$ to represent the contrastive lower bound of $I(\mathbf{y}_0; \mathbf{x})$ presented by Eq. 2, and exhibit this information-theoretic contrastive refinement is complementary and aligned with original conditional denoising diffusion optimization, which is actually another forward predictive method to maximize $I(\mathbf{y}_0; \mathbf{x})$.

To improve the diffusion forecasting capacity more essentially, the developed contrastive learning item is wished to directly benefit naive step-wise denoising-based training procedure, i.e. regularizing noise elimination behaviors of the conditional denoiser $\epsilon_\theta(\cdot)$. Since the density ratio function $f(\mathbf{y}_0, \mathbf{x})$ constituting the contrastive mutual information lower bound in Eq. 2 can be any positive-valued forms, this flexibility naturally motivates us to prescribe $f(\cdot)$ using the step-wise denoising objective in Eq. 1, for both a positive sample $\mathbf{y}_0$ and a group of negative samples $\mathbf{y}_0^{(n)}$:

$$f_{k,\epsilon'}(\mathbf{y}_0, \mathbf{x}; \theta) = \exp(-||\epsilon' - \epsilon_\theta(\sqrt{\bar{\alpha}_k}\mathbf{y}_0 + \sqrt{1 - \bar{\alpha}_k}\epsilon', \mathbf{x}, k)||_2^2/\tau); \tag{3a}$$

$$f_{k,\epsilon'}(\mathbf{y}_0^{(n)}, \mathbf{x}; \theta) = \exp(-||\epsilon' - \epsilon_\theta(\sqrt{\bar{\alpha}_k}\mathbf{y}_0^{(n)} + \sqrt{1 - \bar{\alpha}_k}\epsilon', \mathbf{x}, k)||_2^2/\tau); \tag{3b}$$

where $\tau$ is the temperature coefficient in the softmax-form contrastive loss. In Appendix A.9.2, we also provide another cosine similarity form of $f(\cdot)$ to enhance the denoiser optimization. The negative time series are constructed by a hybrid time series augmentation method which alters both temporal variations and point magnitudes (See Appendix A.9.1 for details.). Then, we can derive the contrastive refinement loss which is coincident with vanilla step-wise denoising diffusion training:

$$\mathcal{L}_k^{contrast} = -\mathbb{E}_{\mathbf{x}, \mathbf{y}_0, \{\mathbf{y}_0^{(n)}\}_{n=1}^N, \epsilon'}[\log \frac{f_{k,\epsilon'}(\mathbf{y}_0, \mathbf{x}; \theta)}{f_{k,\epsilon'}(\mathbf{y}_0, \mathbf{x}; \theta) + \sum_{n=1}^N f_{k,\epsilon'}(\mathbf{y}_0^{(n)}, \mathbf{x}; \theta)}]. \tag{4}$$

Apparently, the devised denoising-based temporal contrastive learning can not only seamlessly coordinate with standard diffusion training at each step $k$, but also improve the conditional denoiser behaviors in out-of-distribution (OOD) regions. These OOD areas are constituted by the low-density diffusion paths of negative samples, which are not touched by merely executing denoising learning along the high-density probability paths of positive samples.

### 3.3 OVERALL LEARNING OBJECTIVE

The naive denoising diffusion model trained by log-likelihood maximization (Ho et al., 2020) totally owns $K$-step valid training items. To align with this step-wise denoising distribution learning, we can amortize the contrastive regularization in Eq. 4 to each training step, and derive the overall learning objective below:

$$\max_\theta \mathbb{E}_{q(\mathbf{y}_0, \mathbf{x})}[\log p_\theta(\mathbf{y}_0|\mathbf{x}) + \lambda K \cdot I_\theta(\mathbf{y}_0; \mathbf{x})], \tag{5}$$

where $\log p_\theta(\mathbf{y}_0|\mathbf{x})$ can be decomposed as $\sum_{k=1}^K \mathcal{L}_k^{denoise}$ and indicates the predictive distribution learning. Whilst $\max_\theta I_\theta(\mathbf{y}_0; \mathbf{x})$ governs the information-theoretic contrastive learning. Then, the practical training loss of the devised CCDM at each diffusion step can be presented as:

$$\mathcal{L}_k^{CCDM} = \mathbb{E}_{\mathbf{y}_0, \mathbf{x}, k \sim U[1, K]}(\mathcal{L}_k^{denoise} + \lambda \mathcal{L}_k^{contrast}). \tag{6}$$

So far, we obtain the overall step-wise training procedure for CCDM, which is a $\lambda$-weighted combination of the vanilla denoising term in Eq. 1 and auxiliary contrastive item in Eq. 4. The whole training algorithm is clarified in Appendix A.3, which is efficient, end-to-end and seamlessly coupled with original simplified denoising diffusion.

**Theoretical insights.** Beyond the method described above, we also offer two-fold theoretical interpretations on how time series diffusion forecasting can benefit from auxiliary contrastive training. From the *neural mutual information perspective*, we show that maximizing $I_\theta(\mathbf{y}_0; \mathbf{x})$ is equivalent to minimizing KL-divergence $\mathcal{D}_{KL}[q(\mathbf{y}_0|\mathbf{x})||p_\theta(\mathbf{y}_0|\mathbf{x})]$ between the real predictive distribution and diffusion-model-approximated distribution (See Appendix A.1.2 for a detailed proof). It is well-known that minimizing $\mathcal{D}_{KL}[q(\mathbf{y}_0|\mathbf{x})||p_\theta(\mathbf{y}_0|\mathbf{x})]$ can be an efficient surrogate for the maximum likelihood learning to improve the log-likelihood $\log p_\theta(\mathbf{y}_0|\mathbf{x})$ of diffusion models (Zhang et al., 2024; Song et al., 2021). As learning the faithful predictive likelihood is necessary for time series probabilistic forecasting (Salinas et al., 2020), complementing mutual information-theoretic contrastive training can gain better likelihood and thus improve the forecasting capacity of time series diffusion models. From the *distribution generalization perspective*, explicitly optimizing the probabilities of unexpected negative samples can render $\epsilon_\theta(\cdot)$ see more OOD regions that purely denoising on positive in-distribution samples do not encompass. In time series learning domain, there always exists distribution shift between unforeseen testing data and historical training data (Kim et al., 2021). The contrastive term in Eq. 4 intuitively minimizes the possibility $\log p_\theta(\mathbf{y}_0^{(n)})$ of undesirable spurious forecasts by directly impeding $\epsilon_\theta(\cdot)$ from correctly removing the noise over negative

$\mathbf{y}_0^{(n)}$. This contrastive training helps $\epsilon_\theta(\cdot)$ avoid low-density areas formed by negative instances and undergo more OOD areas during in-distribution training. As claimed in (Wu et al., 2024), boosting the denoiser robustness in OOD regions in testing stage is crucial to sample plausible forecasts.

Moreover, we reveal the upper bound of conditional diffusion forecasting errors in Proposition 1. It obviously reflects that the diffusion forecasting capacity is inextricably intertwined with the step-wise noise regression accuracy of obtained $\epsilon_\theta(\cdot)$ on unknown test time series. Hence, leveraging temporal contrastive refining or other auxiliary training regimes to boost conditional time series denoising behaviors is conducive to improve final prediction outcomes.

**Proposition 1.** *Let $q_\theta^{te}(\mathbf{y}_0|\mathbf{x})$ be the ground truth distribution of test time series, and $p_\theta^{te}(\mathbf{y}_0|\mathbf{x})$ be the approximated predictive distribution by the developed conditional diffusion model. Let the KL-divergence between $q^{te}(\mathbf{y}_0|\mathbf{x})$ and $p_\theta^{te}(\mathbf{y}_0|\mathbf{x})$ represent the resulting probabilistic forecasting error. Then the denoising diffusion-induced forecasting error is upper-bounded:*

$$\mathcal{D}_{KL}\left[q^{te}(\mathbf{y}_0|\mathbf{x})||p_\theta^{te}(\mathbf{y}_0|\mathbf{x})\right] \le \mathbb{E}_{\mathbf{x},\mathbf{y}_0,\epsilon_k,k}\left[A_k\left\|\epsilon_\theta\left(\sqrt{\bar{\alpha}_k}\mathbf{y}_0 + \sqrt{1-\bar{\alpha}_k}\epsilon_k,\mathbf{x},k\right) - \epsilon_k\right\|_2^2\right] + C. \tag{7}$$

*Such upper bound is determined by the denoising behaviors of learned $\epsilon_\theta(\cdot)$ on unknown test time series. $A_k$ is a step-wise constant related to noise schedule, and $C$ is a constant depending on test data quantities.* See Appendix A.1.1 for the proof.

# 4 EXPERIMENTS

## 4.1 EXPERIMENTAL SETUP

**Datasets.** We choose six multivariate time series datasets, i.e. `ETTh1`, `Exchange`, `Weather`, `Appliance`, `Electricity`, `Traffic`, which cover a wide range of temporal dynamics and channel number $D$ to completely gauge the probabilistic forecasting performance. We manually establish a more comprehensive benchmark with diverse values of lookback window $L$ and prediction horizon $H$, distinct from previous models which merely attest their generative forecasting capacity on a single short-term setup. Refer to Appendix. A.4 for more details on datasets.

**Evaluation metrics.** We adopt two standard metrics to assess the quality of both probabilistic and deterministic forecasts resulting from the generated prediction intervals. CRPS and CRPS_sum are used to assess the reliability of the estimated predictive distribution, and MSE and MAE are used to quantify the accuracy of calculated point forecasts. See Appendix A.5 for more details on metrics.

**Baselines.** We select five currently remarkable denoising diffusion-based generative forecasters for comparisons, including TimeGrad (Rasul et al., 2021), CSDI (Tashiro et al., 2021), SSSD (Alcaraz & Strodthoff, 2022), TimeDiff (Shen & Kwok, 2023), TMDM (Li et al., 2023). Since these models do not shed light on outcomes on long-term probabilistic forecasting scenarios, we fully reproduce them on the newly constructed benchmark. See Appendix A.8.2 for comparisons with more excellent non-diffusion models.

**Implementation details.** We normally execute the end-to-end contrastive diffusion training in Eq. 6 using 100 epochs. To reduce the contrastive learning costs on those cases which consume enormous computational resources, we also employ a cost-efficient two-stage training strategy. Concretely, we firstly pretrain a low-cost naive diffusion forecaster by Eq. 1 and fine-tune it by the total contrastive manner in Eq. 6 with only 30 epochs. We keep the temperature coefficient $\tau = 0.1$ and randomly generate $S = 100$ multivariate profiles to compose prediction intervals. See Appendix A.6 for more details on network architecture and contrastive training configurations in different forecasting setups. All experiments are conducted on a single NVIDIA A100 GPU.

## 4.2 OVERALL RESULTS

We demonstrate the devised CCDM model can outperform existing diffusion forecasters on most of the generative forecasting cases in Table 1. Concretely, CCDM can attain the best outcomes on 19/24 deterministic and 21/24 probabilistic evaluations, with 7.74% and 26.16% average improvement of MSE and CRPS on these cases. Especially on two most difficult datasets `Electricity` and `Traffic`, CCDM garners notable progress of 13.48%, 13.64% on MSE and 22.93%, 21.63%

Table 1: Overall comparisons w.r.t MSE and CRPS on six real-world datasets with diverse horizon $H \in \{96, 168, 336, 720\}$. The best and second-best results are boldfaced and underlined.

| Methods | | CCDM | | TMDM | | TimeDiff | | SSSD | | CSDI | | TimeGrad | |
|---|---|---|---|---|---|---|---|---|---|---|---|---|---|---|
| Metrics | | MSE | CRPS | MSE | CRPS | MSE | CRPS | MSE | CRPS | MSE | CRPS | MSE | CRPS |
| ETTh1 | 96 | **0.3715** | **0.2856** | 0.4692 | 0.3952 | 0.4025 | 0.3942 | 1.0984 | 0.5622 | 1.1013 | 0.5794 | 1.1730 | 0.6223 |
| | 168 | **0.4137** | **0.3027** | 0.5296 | 0.4163 | 0.4397 | 0.4170 | 0.6067 | 0.4046 | 1.1013 | 0.5794 | 1.1554 | 0.5970 |
| | 336 | 0.5146 | **0.3391** | 0.5862 | 0.4655 | **0.4943** | **0.4488** | 0.9330 | 0.5421 | 1.0459 | 0.6223 | 1.1403 | 0.5883 |
| | 720 | **0.5545** | **0.4856** | 0.7083 | 0.5335 | 0.5779 | 0.5145 | 1.3776 | 0.7035 | 1.0081 | 0.5952 | 1.2529 | 0.6498 |
| | Avg | **0.4636** | **0.3533** | 0.5733 | 0.4526 | 0.4786 | 0.4418 | 1.0039 | 0.5531 | 1.0642 | 0.5941 | 1.1804 | 0.6144 |
| Exchange | 96 | **0.0905** | **0.1545** | 0.1278 | 0.2112 | 0.1106 | 0.2349 | 0.5551 | 0.4569 | 0.2551 | 0.2901 | 1.8655 | 1.0439 |
| | 168 | **0.1638** | **0.2159** | 0.2791 | 0.3210 | 0.2050 | 0.3187 | 0.4517 | 0.3602 | 0.8050 | 0.5093 | 1.1638 | 0.8374 |
| | 336 | **0.4407** | **0.3517** | 0.4572 | 0.4426 | 0.5834 | 0.5472 | 0.5641 | 0.4106 | 0.6179 | 0.4786 | 1.9264 | 1.0465 |
| | 720 | 1.1685 | **0.5864** | 2.5625 | 1.0828 | **0.9096** | 0.7128 | 1.3686 | 0.6386 | 1.3816 | 0.7423 | 2.4034 | 1.1478 |
| | Avg | 0.4659 | 0.3271 | 0.8567 | 0.5144 | **0.4522** | 0.4534 | 0.7349 | 0.4666 | 0.7649 | 0.5051 | 1.8398 | 1.0189 |
| Weather | 96 | **0.2452** | **0.1826** | 0.2768 | 0.2273 | 0.3842 | 0.3441 | 0.6103 | 0.3878 | 0.2608 | 0.2127 | 0.5628 | 0.3445 |
| | 168 | **0.2407** | **0.1898** | 0.2864 | 0.2519 | 0.3566 | 0.3192 | 0.2796 | 0.2060 | 0.2930 | 0.2286 | 0.4141 | 0.2880 |
| | 336 | **0.2840** | 0.2230 | 0.3494 | 0.3007 | 0.4805 | 0.3591 | 0.3189 | 0.2355 | 0.2918 | **0.2193** | 0.5462 | 0.3549 |
| | 720 | 0.5599 | 0.4074 | 0.3975 | 0.3365 | 0.5052 | 0.3880 | 0.6880 | 0.4179 | **0.3803** | **0.2770** | 0.4774 | 0.3221 |
| | Avg | 0.3325 | 0.2507 | 0.3275 | 0.2791 | 0.4316 | 0.3526 | 0.4742 | 0.3118 | **0.3065** | **0.2344** | 0.5001 | 0.3274 |
| Appliance | 96 | **0.6227** | **0.3889** | 0.6858 | 0.4678 | 0.7328 | 0.5740 | 1.1954 | 0.6504 | 0.6823 | 0.4334 | 1.6748 | 0.8397 |
| | 168 | **0.6266** | **0.4020** | 0.7153 | 0.5232 | 0.6468 | 0.5562 | 0.7841 | 0.4776 | 0.7176 | 0.4560 | 1.8901 | 0.8858 |
| | 336 | **0.9119** | **0.5036** | 1.0310 | 0.6590 | 0.9531 | 0.6822 | 1.8822 | 0.8002 | 1.0565 | 0.5675 | 1.8506 | 0.8661 |
| | 720 | 1.5599 | 0.8594 | **1.3937** | **0.8272** | 1.4327 | 0.8809 | 3.3226 | 1.1225 | 1.7347 | 0.7982 | 2.4393 | 1.0083 |
| | Avg | **0.9303** | **0.5385** | 0.9565 | 0.6193 | 0.9414 | 0.6733 | 1.7961 | 0.7627 | 1.0478 | 0.5638 | 1.9637 | 0.9000 |
| Electricity | 96 | **0.1897** | **0.2046** | 0.1954 | 0.3113 | 0.1960 | 0.3123 | 0.2444 | 0.2346 | 0.2560 | 0.2571 | 0.3733 | 0.3259 |
| | 168 | **0.1575** | **0.1893** | 0.1908 | 0.3037 | 0.1907 | 0.3043 | 0.2001 | 0.2249 | 0.1754 | 0.1985 | 0.3676 | 0.3083 |
| | 336 | **0.1651** | **0.1983** | 0.2042 | 0.3165 | 0.2047 | 0.3172 | 0.1941 | 0.2245 | 0.1803 | 0.2043 | 0.4249 | 0.3497 |
| | 720 | **0.1959** | **0.2184** | 0.2282 | 0.3338 | 0.2277 | 0.3336 | 0.3743 | 0.3680 | 0.9932 | 0.5678 | 0.4299 | 0.3479 |
| | Avg | **0.1771** | **0.2027** | 0.2047 | 0.3163 | 0.2048 | 0.3169 | 0.2532 | 0.2630 | 0.4012 | 0.3069 | 0.3989 | 0.3330 |
| Traffic | 96 | 1.0291 | **0.3911** | 0.9692 | 0.5894 | **0.9684** | 0.5859 | 1.0363 | 0.4445 | 1.1154 | 0.4240 | 1.2259 | 0.4667 |
| | 168 | **0.6881** | **0.3077** | 0.8632 | 0.5254 | 0.8553 | 0.5192 | 0.9551 | 0.4289 | 1.6000 | 0.6701 | 1.3282 | 0.5510 |
| | 336 | **0.6683** | **0.3284** | 0.8874 | 0.5562 | 0.8834 | 0.5538 | 0.9283 | 0.5140 | 1.5724 | 0.6780 | 1.0447 | 0.3817 |
| | 720 | **0.8392** | **0.4304** | 1.0258 | 0.6383 | 1.0270 | 0.6387 | 1.0635 | 0.5515 | 1.5428 | 0.6696 | 1.1753 | 0.4604 |
| | Avg | **0.8062** | **0.3644** | 0.9364 | 0.5773 | 0.9335 | 0.5744 | 0.9958 | 0.4847 | 1.4577 | 0.6104 | 1.1935 | 0.4650 |
| 1st Count | | **40** | | 1 | | 3 | | 0 | | 4 | | 0 | |

on CRPS. These prominent increases reflect the devised channel-centric structure and contrastive refinement on the diffusion forecaster can enhance its representation efficiency of implicit predictive information on diverse prediction scenarios. The second-best model CSDI also manifests excellent forecasting ability especially on `Weather`, which has complex multivariate temporal correlations. The hybrid attention module in CSDI can well capture these relations but it entices high computational overhead and over-fitting to other datasets. TMDM and TimeDiff also attain small MSE on few cases due to their extra deterministic pre-training operations on conditioning encoders. Note that we completely replicate TimeGrad on the whole benchmark for the first time even with severe inference costs, and validate it can actually realize reasonable forecasting results. In Fig. 4, we depict different diffusion produced prediction intervals on one case. We can clearly see that CCDM's interval is much more faithful, while TimeDiff's area is sharper but loses diversity and accuracy. See Appendix A.11 for more forecasting result showcases and Appendix A.7 on time cost analysis.

### 4.3 ABLATION STUDY

To investigate respective effects of each component, we remove the proposed denoising-based contrastive learning and channel-wise DiT structure, and exhibit the average metric degradation over different prediction horizons in Table 2. Without auxiliary contrastive diffusion training, we observe a mean performance drop of 10.21% and 8.13% on MSE and CRPS over the whole benchmark. This notable decrease indicates that the dedicated denoising-based contrastive refinement can enhance the utilization efficiency of conditional temporal predictive information and yield a more genuine multivariate predictive distribution. Due to the restriction of computational costs, such contrastive gains on `Electricity` and `Traffic` datasets are relatively smaller. We can amplify contrastive benefits on large-scale datasets by increasing the batch size and negative number within an iteration in the future. Regarding the influence of composite channel-aware management in conditional denoiser, we replace the channel-wise DiT modules by the same depth of linear dense encoders and incur a

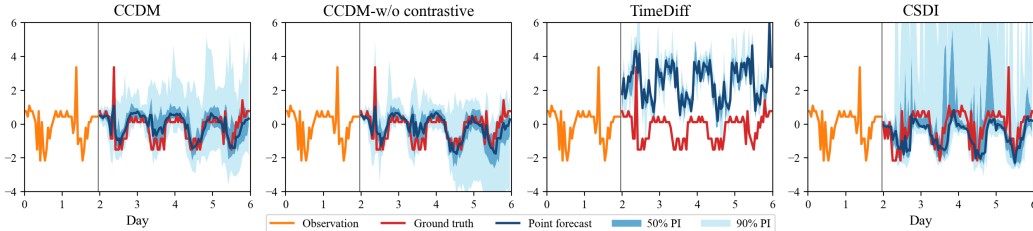

Figure 4: Comparison of generated point forecasts and prediction intervals on an Electricity channel.

full channel-independence architecture. The average reduction on MSE and CRPS over the whole test settings are 22.75% and 29.06%. This considerable drop reveals that the channel-mixing attention can empower the denoising network to integrate useful cross-variate temporal features in past observations and corrupted targets. Besides, the elevation degree induced by channel-centric DiT is consistent with the true variate correlations in real-world datasets. For instance, the performance decrease is less salient on `Electricity` dataset where the electricity consumption of different customers is not highly related to each other. Whilst on `ETTh1` and `Weather` datasets whose sensory measurements are heavily inter-correlated, the channel-mixing DiT can improve the diffusion forecasting capacity more vastly.

Table 2: Average MSE and CRPS degradation resulting from the ablation of denoising-based contrastive learning or channel-wise DiT module. Full results can be found in Appendix A.8.3.

| Models | w/o contrastive refinement | | | | w/o channel-wise DiT | | | |
|---|---|---|---|---|---|---|---|---|
| Metrics | MSE | Degradation | CRPS | Degradation | MSE | Degradation | CRPS | Degradation |
| ETTh1 | 0.5508 | 18.81% | 0.3889 | 10.08% | 0.5956 | 28.47% | 0.5816 | 64.62% |
| Exchange | 0.4966 | 6.59% | 0.3403 | 4.04% | 0.4924 | 5.69% | 0.3555 | 8.68% |
| Weather | 0.3816 | 14.77% | 0.2695 | 7.50% | 0.4843 | 45.65% | 0.3336 | 33.07% |
| Appliance | 1.0220 | 9.86% | 0.5818 | 8.04% | 1.1183 | 20.21% | 0.7231 | 34.28% |
| Electricity | 0.1887 | 6.55% | 0.2144 | 5.77% | 0.1973 | 11.41% | 0.2137 | 5.43% |
| Traffic | 0.8439 | 4.68% | 0.4130 | 13.34% | 1.0084 | 25.08% | 0.4675 | 28.29% |

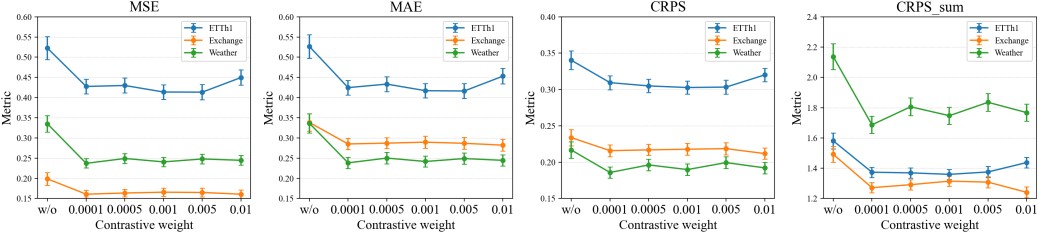

Figure 5: Results by varying contrastive weight $\lambda$ on three datasets with $H = 168$. Note that w/o indicates CCDM is obtained without contrastive training, i.e. $\lambda = 0$. The mean and standard error of 4 metrics are obtained from 10 independently repeated runs.

## 4.4 CONTRASTIVE REFINEMENT ANALYSIS

Below, we empirically investigate the efficacy of the devised denoising-based temporal contrastive refinement, including three vital factors for contrastive learning practice and its generality on other existing diffusion forecasters.

**Influence of contrastive weight $\lambda$.** The complementary step-wise denoising-based contrastive loss in 4 can enhance the alignment between diffusion generated forecasts and given temporal predictive information. To elucidate how different degrees of contrastive refining can affect naive diffusion optimization, we escalate the contrastive weight $\lambda$ in Eq. 5 from 0.0001 to 0.01 and display corresponding outcomes in Fig. 5. Generally speaking, four metrics of w/o (i.e. $\lambda = 0$) are consistently larger than those of imposing contrastive training (i.e. $\lambda > 0$). This reflects that adding contrastive refining to naive diffusion predictive learning can promote the forecasting capacity. Besides, we can see that the contrastive gain margin moderately fluctuates among various weights and datasets, and

Table 3: Forecasting performance promotion induced by applying denoising-based contrastive training to two existing conditional diffusion forecasters.

| Methods | | TimeDiff | | | | CSDI | | | |
|---|---|---|---|---|---|---|---|---|---|
| Metrics | | MSE | Promotion | CRPS | Promotion | MSE | Promotion | CRPS | Promotion |
| ETTh1 | 96 | 0.4143 | -2.93% | 0.3491 | 11.44% | 0.6559 | 40.44% | 0.4371 | 24.56% |
| | 168 | 0.4715 | -7.23% | 0.3753 | 10.00% | 0.5894 | 29.53% | 0.3851 | 25.76% |
| | 336 | 0.5073 | -2.63% | 0.4025 | 10.32% | 0.9920 | 5.15% | 0.5644 | 9.30% |
| | 720 | 0.5291 | 6.19% | 0.4338 | 14.44% | 0.7744 | 23.18% | 0.7010 | -17.78% |
| Exchange | 96 | 0.0901 | 18.54% | 0.1722 | 26.69% | 0.1589 | 37.71% | 0.2082 | 28.23% |
| | 168 | 0.1588 | 22.54% | 0.2312 | 27.46% | 0.4096 | 49.12% | 0.3840 | 24.60% |
| | 336 | 0.6345 | -8.76% | 0.4293 | 21.55% | 0.5664 | 8.33% | 0.4110 | 14.12% |
| | 720 | 0.9735 | -7.03% | 0.6941 | 2.62% | 1.3642 | 1.26% | 0.6392 | 13.89% |

a modest weight between 0.0005 and 0.005 can lead to better improvement. See Appendix A.9 for more detailed analysis on the influence of negative number $N$ and temperature $\tau$.

**Generality of contrastive training.** We add the step-wise denoising contrastive training presented in 4 to two existing diffusion forecasters to validate its generality on conditional time series diffusion learning. From the results shown in Table 3, it is obvious that CSDI's generative forecasting ability can be further enhanced by contrastive diffusion training. Its hybrid attention network can represent complex temporal patterns more properly by handling more OOD negative samples. While for TimeDiff which owns extra pre-trained auto-regressive conditioning encoders, CRPS values constantly decrease but some unexpected increases appear on MSE. It may stem from the side effect of redundant contrastive procedure conveyed to the well-behaved deterministic pre-training strategy.

## 5 RELATED WORK

**Channel-oriented multivariate forecasting.** Recent progress on multivariate deterministic prediction (Liu et al., 2023; Lu et al., 2023; Chen et al., 2024a; Han et al., 2024) indicate that learning channel-centric temporal properties (including single-channel dynamics and cross-channel correlations) is of significant importance. Both channel-independent and channel-fusing time series processing are crucial to improve the forecasting performance. But the effectiveness of such channel manipulation structures is rarely investigated in diffusion-based multivariate probabilistic forecasting, where the extra influence of imposed channel noise in varying degrees should also be addressed. To tackle this barrier, we blend both channel-independent and channel-mixing modules in the conditional diffusion denoiser to boost its forecasting ability on multivariate cases.

**Time series diffusion models.** Diffusion models have been actively applied to tackle a wide scope of time series tasks, including synthesis (Yuan & Qiao, 2024; Narasimhan et al., 2024), forecasting (Rasul et al., 2021), imputation (Tashiro et al., 2021) and anomaly detection (Chen et al., 2023). Their common goal is to derive a high-quality conditional temporal distribution aligned with diverse input contexts, such as statistical properties in constrained generation (Coletta et al., 2024) and historical records. A valid solution is to inject useful temporal properties into iterative diffusion learning (Yuan & Qiao, 2024; Biloš et al., 2023) or to develop gradient-based guidance schemes (Coletta et al., 2024). But there are still rooms to enhance them from the aspect of training methods and denoiser architectures. To bridge this gap for multivariate forecasting, we exclusively design a channel-aware denoiser and explicitly enhance the predictive mutual information between past observations and future forecasts by an adapted temporal contrastive diffusion learning. Even though several works have applied contrastive diffusion to cross-modal content creation (Wang et al., 2024b; Zhu et al., 2022), its efficacy on time series generative modeling have not yet been well explored. And reasonable interpretations on such contrastive diffusion merits are also scanty. See Appendix A.2 for more detailed related work, which also covers universal temporal contrastive learning.

## 6 CONCLUSION

In this work, we propose the channel-aware contrastive conditional diffusion model named CCDM for probabilistic forecasts on multivariate time series. CCDM can capture intrinsic prediction-related temporal information hidden in observed conditioning time series using an efficient channel-centric denoiser architecture and information-maximizing denoising-based contrastive refinement. Extensive experiments demonstrate the exceptional forecasting capability of CCDM over existing time series diffusion models. In future work, we plan to reduce the training costs imposed by additional temporal contrastive learning, and extend this contrastive diffusion method to general time series analysis and other cross-domain synthesis tasks.

ETHICS STATEMENT

Our work is only aimed at faithful multivariate probabilistic forecasting for human good, so there is no involvement of human subjects or conflict of interests as far as the authors are aware of.

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

# A APPENDIX

## A.1 THEORETIC RESULTS

### A.1.1 PROOF FOR PROPOSITION 1

Below, we shed light on how to derive the upper bound of diffusion-induced probabilistic forecasting error shown in Proposition 1. We utilize the KL-divergence between the real distribution $q^{te}(\mathbf{y}_0|\mathbf{x})$ of test time series and approximated predictive distribution $p_\theta^{te}(\mathbf{y}_0|\mathbf{x})$ by conditional diffusion models to represent the probabilistic forecasting error

$$\mathcal{D}_{KL}\left[q^{te}(\mathbf{y}_0|\mathbf{x})||p_\theta^{te}(\mathbf{y}_0|\mathbf{x})\right] = \mathbb{E}_{q^{te}(\mathbf{y}_0|\mathbf{x})}\left[\log q^{te}(\mathbf{y}_0|\mathbf{x})\right] - \mathbb{E}_{q^{te}(\mathbf{y}_0|\mathbf{x})}\left[\log p_\theta^{te}(\mathbf{y}_0|\mathbf{x})\right]. \quad (8)$$

The first term in Eq. 8 is unrelated to conditional diffusion learning and thus can be prescribed as a constant $C_1$ based on the information quantity of real test data

$$\mathbb{E}_{q^{te}(\mathbf{y}_0|\mathbf{x})}\left[\log q^{te}(\mathbf{y}_0|\mathbf{x})\right] = -\frac{1}{q^{te}(\mathbf{x})}\mathbb{E}_{q^{te}(\mathbf{y}_0,\mathbf{x})}\left[\log q^{te}(\mathbf{y}_0|\mathbf{x})\right] = -\frac{H(\mathbf{y}_0|\mathbf{x})}{q^{te}(\mathbf{x})} = C_1. \quad (9)$$

The second term Eq. 8 is the expected log-likelihood over $q^{te}(\mathbf{y}_0|\mathbf{x})$, which is identical to the learning objective of vanilla conditional diffusion models in (Ho et al., 2020). Akin to the step-wise denoising loss derivation in (Ho et al., 2020), we can obtain the upper bound of the error via Jensen's inequality and decompose it into $K+1$ items $\mathcal{V}_0, ..., \mathcal{V}_K$:

$$-\mathbb{E}_{q^{te}(\mathbf{y}_0|\mathbf{x})}\left[\log p_\theta^{te}(\mathbf{y}_0|\mathbf{x})\right] = -\mathbb{E}_{q^{te}(\mathbf{y}_0|\mathbf{x})}\left[\log \int q^{te}(\mathbf{y}_{1:K}|\mathbf{y}_0)\frac{p_\theta^{te}(\mathbf{y}_{0:K}|\mathbf{x})}{q^{te}(\mathbf{y}_{1:K}|\mathbf{y}_0)}d\mathbf{y}_{1:K}\right]$$

$$\leq -\mathbb{E}_{q^{te}(\mathbf{y}_0|\mathbf{x})}\left[\mathbb{E}_{q^{te}(\mathbf{y}_{1:K}|\mathbf{y}_0)}\left[\log \frac{p_\theta^{te}(\mathbf{y}_{0:K}|\mathbf{x})}{q^{te}(\mathbf{y}_{1:K}|\mathbf{y}_0)}\right]\right]$$

$$= \mathbb{E}_{q^{te}(\mathbf{y}_0|\mathbf{x})}\left[\mathcal{V}_0 + \sum_{k=2}^{K}\mathcal{V}_{k-1} + \mathcal{V}_K\right], \quad (10)$$

where

$$\mathcal{V}_K = \mathcal{D}_{KL}\left[q^{te}(\mathbf{y}_K|\mathbf{y}_0)||p_\theta^{te}(\mathbf{y}_K|\mathbf{x})\right] = 0, \quad (11)$$

as $q^{te}(\mathbf{y}_K|\mathbf{y}_0)$ and $p_\theta^{te}(\mathbf{y}_K|\mathbf{x})$ are both standard Gaussian. And since the reverse transitions at each diffusion step can be shaped in explicit Gaussian forms, we can write out

$$\mathcal{V}_{k-1} = \mathbb{E}_{q^{te}(\mathbf{y}_k|\mathbf{y}_0)}\left[\mathcal{D}_{KL}\left[q^{te}(\mathbf{y}_{k-1}|\mathbf{y}_k,\mathbf{y}_0)||p_\theta^{te}(\mathbf{y}_{k-1}|\mathbf{y}_k,\mathbf{x})\right]\right]$$

$$= \mathbb{E}_{q^{te}(\mathbf{y}_k|\mathbf{y}_0)}\left[\mathcal{D}_{KL}\left[\mathcal{N}(\mathbf{y}_{k-1};\boldsymbol{\mu}_k(\mathbf{y}_k,\mathbf{y}_0),\tilde{\beta}_k\mathbf{I})||\mathcal{N}(\mathbf{y}_{k-1};\boldsymbol{\mu}_\theta(\mathbf{y}_k,\mathbf{x},k),\tilde{\beta}_k\mathbf{I})\right]\right]$$

$$= \mathbb{E}_{q^{te}(\mathbf{y}_k|\mathbf{y}_0)}\left[\frac{1}{2\tilde{\beta}_k^2}\left[\|\boldsymbol{\mu}_\theta(\mathbf{y}_k,\mathbf{x},k) - \boldsymbol{\mu}_k(\mathbf{y}_k,\mathbf{y}_0)\|_2^2\right]\right]$$

$$= \mathbb{E}_{\mathbf{y}_0,\boldsymbol{\epsilon}_k}\left[\frac{1}{2\tilde{\beta}_k^2}\left[\left\|\frac{1}{\sqrt{\alpha_k}}\left(\mathbf{y}_k - \frac{\beta_k}{\sqrt{1-\bar{\alpha}_k}}\boldsymbol{\epsilon}_\theta\left(\mathbf{y}_k,\mathbf{x},k\right)\right) - \frac{1}{\sqrt{\alpha_k}}\left(\mathbf{y}_k - \frac{\beta_k}{\sqrt{1-\bar{\alpha}_k}}\boldsymbol{\epsilon}_k\right)\right\|_2^2\right]\right]$$

$$= \mathbb{E}_{\mathbf{y}_0,\boldsymbol{\epsilon}_k}\left[\frac{\beta_k^2}{2\tilde{\beta}_k^2\alpha_k(1-\bar{\alpha}_k)}\left[\|\boldsymbol{\epsilon}_\theta\left(\sqrt{\bar{\alpha}_k}\mathbf{y}_0 + \sqrt{1-\bar{\alpha}_k}\boldsymbol{\epsilon}_k,\mathbf{x},k\right) - \boldsymbol{\epsilon}_k\|_2^2\right]\right], \quad (12)$$

where $\tilde{\beta}_k = \frac{1-\bar{\alpha}_{k-1}}{1-\bar{\alpha}_k}\beta_k$ and $\mathcal{V}_0$ is actually a special case of Eq. 12 when $k=1$

$$\mathcal{V}_0 = -\mathbb{E}_{q(\mathbf{y}_1|\mathbf{y}_0)}\left[\log p_\theta(\mathbf{y}_0|\mathbf{y}_1,\mathbf{x})\right]$$

$$= \mathbb{E}_{q(\mathbf{y}_1|\mathbf{y}_0)}\left[\log(2\pi)^{\frac{HD}{2}}\tilde{\beta}_1 + \frac{1}{2\tilde{\beta}_1^2}\|\mathbf{y}_0 - \boldsymbol{\mu}_\theta(\mathbf{y}_1,\mathbf{x},k=1)\|_2^2\right]$$

$$= \mathbb{E}_{\mathbf{y}_0,\boldsymbol{\epsilon}_1}\left[\frac{\beta_1^2}{2\tilde{\beta}_1^2\alpha_1(1-\bar{\alpha}_1)}\left[\|\boldsymbol{\epsilon}_\theta\left(\sqrt{\bar{\alpha}_1}\mathbf{y}_0 + \sqrt{1-\bar{\alpha}_1}\boldsymbol{\epsilon}_1,\mathbf{x},k=1\right) - \boldsymbol{\epsilon}_1\|_2^2\right]\right] + C_2. \quad (13)$$

Overall, if we let $A_k = \frac{\beta_k^2}{2\tilde{\beta}_k^2 \alpha_k (1 - \bar{\alpha}_k)}$, $C = C_1 + C_2$, we can derive the ultimate upper bound of probabilistic forecasting error in a concise form as follows:

$$\mathcal{D}_{KL}\left[q^{te}(\mathbf{y}_0|\mathbf{x})||p_\theta^{te}(\mathbf{y}_0|\mathbf{x})\right] \leq \mathbb{E}_{\mathbf{x},\mathbf{y}_0,\boldsymbol{\epsilon}_k,k}\left[A_k\left\|\boldsymbol{\epsilon}_\theta\left(\sqrt{\bar{\alpha}_k}\mathbf{y}_0 + \sqrt{1 - \bar{\alpha}_k}\boldsymbol{\epsilon}_k, \mathbf{x}, k\right) - \boldsymbol{\epsilon}_k\right\|_2^2\right] + C,$$
(14)

which finalizes the proof of Proposition 1. It shows that for unknown test time series, the diffusion-based generative forecasting performance is associated with the generalization capability of the trained conditional denoising network on total step-wise noise regression.

### A.1.2 ANALYSIS ON INFORMATION-THEORETIC CONTRASTIVE DIFFUSION LEARNING

Here, we vindicate that maximizing predictive mutual information $I(\mathbf{y}_0; \mathbf{x})$ is equivalent to minimizing the KL-divergence $\mathcal{D}_{KL}[q(\mathbf{y}_0|\mathbf{x})||p_\theta(\mathbf{y}_0|\mathbf{x})]$ from the genuine predictive distribution to the diffusion approximated distribution. The detailed proof is presented as follows:

$$
\begin{aligned}
I(\mathbf{y}_0; \mathbf{x}) &= \mathbb{E}_{q(\mathbf{y}_0,\mathbf{x})}\left[\log \frac{p_\theta(\mathbf{y}_0|\mathbf{x})}{q(\mathbf{y}_0)}\right] \\
&= \mathbb{E}_{q(\mathbf{y}_0,\mathbf{x})}\left[\log \frac{p_\theta(\mathbf{y}_0|\mathbf{x})}{q(\mathbf{y}_0|\mathbf{x})} \cdot \frac{q(\mathbf{y}_0|\mathbf{x})}{q(\mathbf{y}_0)}\right] \\
&= \mathbb{E}_{q(\mathbf{x})}\left[\int p_\theta(\mathbf{y}_0|\mathbf{x}) \log \frac{p_\theta(\mathbf{y}_0|\mathbf{x})}{q(\mathbf{y}_0|\mathbf{x})}\mathrm{d}\mathbf{y}_0\right] + \mathbb{E}_{q(\mathbf{x}|\mathbf{y}_0)}\left[\int q(\mathbf{y}_0) \cdot \frac{q(\mathbf{y}_0|\mathbf{x})}{q(\mathbf{y}_0)}\mathrm{d}\mathbf{y}_0\right] \\
&= \mathbb{E}_{q(\mathbf{x})}\left[\mathcal{D}_{KL}(p_\theta(\mathbf{y}_0|\mathbf{x})||q(\mathbf{y}_0|\mathbf{x}))\right] - \mathbb{E}_{q(\mathbf{x}|\mathbf{y}_0)}\left[\mathcal{D}_{KL}(q(\mathbf{y}_0)||q(\mathbf{y}_0|\mathbf{x}))\right] \\
&\leq \mathbb{E}_{q(\mathbf{x})}\left[\mathcal{D}_{KL}(p_\theta(\mathbf{y}_0|\mathbf{x})||q(\mathbf{y}_0|\mathbf{x}))\right].
\end{aligned}
$$
(15)

Apparently, $I(\mathbf{y}_0; \mathbf{x})$ is upper bounded by $\mathcal{D}_{KL}[q(\mathbf{y}_0|\mathbf{x})||p_\theta(\mathbf{y}_0|\mathbf{x})]$ and can be maximized by minimizing $\mathcal{D}_{KL}[q(\mathbf{y}_0|\mathbf{x})||p_\theta(\mathbf{y}_0|\mathbf{x})]$ on provided historical observations $\mathbf{x}$. It is widely-acknowledged that minimizing $\mathcal{D}_{KL}[q(\mathbf{y}_0|\mathbf{x})||p_\theta(\mathbf{y}_0|\mathbf{x})]$ is an effective proxy for the maximum likelihood training Song et al. (2021); Zhang et al. (2024). It can lead to better log-likelihood for diffusion models since vanilla combination of an array of weighted noise regression losses in Eq. 1 can not directly optimize the log-likelihood $\log p_\theta(\mathbf{y}_0|\mathbf{x})$ (Ho et al., 2020). Besides, Song et al. (2021); Zhang et al. (2024) have demonstrated that integrating the maximum likelihood training manner with the naive score matching objective can acquire a significantly better generation quality. Accordingly, in this time series probabilistic forecasting work, we propose to explicitly maximize $I(\mathbf{y}_0; \mathbf{x})$ through the devised denoising-based InfoNCE loss in Eq. 4, which can serve to improve the prediction-related likelihood of time series diffusion models and further enhance the forecasting capability.

### A.2 ADDITIONAL DISCUSSIONS ON RELATED WORKS

**Channel-oriented multivariate forecasting.** How to properly manage various channel-centric temporal properties (i.e. single-channel dynamics and cross-channel correlations) has been attached greater importance in recent multivariate forecasting works (Chen et al., 2024a; Han et al., 2024) for two reasons. One is that traditional transformer-based models (Zhou et al., 2021; Wu et al., 2021; Zhou et al., 2022; Liu et al., 2022) only focus on improving the expressivity and efficiency of long-range temporal dependency, which can not obviously discriminate roles of disparate channels and entice some unsatisfactory outcomes. Besides, channel-independent predictors (Nie et al., 2022; Zeng et al., 2023; Das et al., 2023a) utilize a shared network to uniformly treat all channels and display that the single-channel separate prediction can outperform multi-channel mixing settings. Whilst this channel-independent structure fail to handle those complex temporal modes where the auxiliary information from other channels could also be helpful. Latest progress (Liu et al., 2023; Lu et al., 2023; Chen et al., 2024a; Han et al., 2024) reflect that both channel-independence and channel-fusion are crucial for versatile time series predictors. However, the significance of proper channel manipulation is rarely probed in multivariate diffusion forecasters, and the additional influence of channel noise imposed in different extents should also be considered. To tackle this barrier, we blend both channel-independence and channel-fusion modules in diffusion denoiser to boost its forecasting ability on multivariate cases.

**Time series diffusion models.** Due to the remarkable capacity to generate high-fidelity samples, diffusion models are actively exploited to grasp the stochastic dynamics and temporal correlations

for a variety of time series tasks, including synthesis (Yuan & Qiao, 2024; Narasimhan et al., 2024), forecasting (Rasul et al., 2021), imputation (Tashiro et al., 2021) and anomaly detection (Chen et al., 2023). Common goals of these tasks are to derive a high-quality conditional temporal distribution aligned with diverse input contexts, such as statistical properties in constrained generation (Coletta et al., 2024) and historical records. To this end, the key challenge lies in how to design a potent temporal conditioning mechanism to empower the conditional backward generation. An intuitive way is to integrate useful temporal properties such as trend-seasonality (Yuan & Qiao, 2024), continuity (Biloš et al., 2023) and multi-scale modes (Shen et al., 2023; Fan et al., 2024) to empirically boost the utilization efficiency of conditioning data in the learnable denoising process. Another track is to develop gradient-based guidance schemes to satisfy given constraints via differentiable (Coletta et al., 2024) or objective-oriented optimization (Kollovieh et al., 2024). Even this plethora of time series diffusion models, there are still rooms to enhance them from the aspect of training manners and denoiser architectures. To bridge this gap for multivariate forecasting, we exclusively design a channel-aware denoiser network and recast the problem of estimating conditional predictive distribution in the paradigm of mutual information maximization, which can enhance the consistency between past conditioning and future predicted time series. On top of original conditional likelihood maximization via step-wise noise regression, we adapt temporal contrastive learning to further augment conditional diffusion training. In future work, we hope to extend such innovations to benefit other time series analysis tasks.

**Time series contrastive learning.** Time series contrastive learning primarily aims to obtain self-supervised universal temporal representations which can enable an array of downstream tasks with few shots (Trirat et al., 2024; Lee et al., 2023; Franceschi et al., 2019; Wang et al., 2024a). This line of research focus on developing efficient representation learning methods to pre-train temporal feature extractors in two vital senses, containing contrastive loss design and positive and negative sample pair construction. With respect to the deterministic time series prediction task, there also exist specialized decomposed contrastive pre-training approaches (Woo et al., 2021; Wang et al., 2022) to investigate disentangled seasonal and trend representations, which can relieve the subsequent prediction on volatile temporal evolution. While in this work, we devise an end-to-end denoising-based contrastive learning to ameliorate conditional denoiser training rather than the common pre-training fashion on general temporal representation networks. We realize this contrastive refinement in an identical form of step-wise noise regression to seamlessly align with vanilla diffusion training, whereas other popular methods often design the temporal feature similarity-based objective to govern the training process. Moreover, we alter both temporal variations and point magnitudes in the time series augmentation stage, which can construct more useful negative samples for the contrastive denoiser improvement.

### A.3 TRAINING ALGORITHM

We elucidate the step-wise denoising-based contrastive diffusion training algorithm in Algorithm 1.

---

**Algorithm 1** Step-wise contrastive conditional diffusion training procedure.

---

**Input:** Lookback time series $\mathbf{x} \in \mathbb{R}^{L \times D}$; target time series $\mathbf{y}_0 \in \mathbb{R}^{H \times D}$; lookback length $L$; prediction horizon $H$; variate number $D$; diffusion step number $K$; negative sample number $N$; contrastive loss weight $\lambda$; temperature coefficient $\tau$;

**repeat**
1: Draw step $k \sim \mathbb{U}[1, .., K]$.
2: Draw noise $\boldsymbol{\epsilon} \sim \mathcal{N}(\mathbf{0}, \mathbf{I})$ to calculate the naive diffusion loss $\mathcal{L}_k^{denoise}$ in Eq. 1.
3: Draw noise $\boldsymbol{\epsilon}' \sim \mathcal{N}(\mathbf{0}, \mathbf{I})$ to calculate the denoising-based contrastive loss $\mathcal{L}_k^{contrast}$ in Eq. 4.
4: Obtain a set of negative time series $\{\mathbf{y}_0^n\}_{n=1}^N$ using the hybrid augmentation in Appendix A.9.1.
5: Compute the contrastive conditional diffusion loss $\mathcal{L}_k^{CCDM} = \mathcal{L}_k^{denoise} + \lambda \mathcal{L}_k^{contrast}$ in Eq. 6.
6: Optimize the conditional denoising network $\boldsymbol{\epsilon}_\theta(\cdot)$ using the gradient $\nabla_\theta \mathcal{L}_k^{CCDM}$.
**until** converged

---

## A.4 DATASET DESCRIPTION

We present the dataset usage in Table 4, where the channel number $D$, sampling rate, train/val/test split size and own field are clarified. We also provide accessible repositories for these datasets below:

1) `ETTh1`: https://github.com/zhouhaoyi/ETDataset
2) `Exchange`: https://github.com/laiguokun/multivariate-time-series-data
3) `Weather`: https://www.bgc-jena.mpg.de/wetter/
4) `Appliance`: https://archive.ics.uci.edu/dataset/374/appliances+energy+prediction
5) `Electricity`: https://archive.ics.uci.edu/dataset/321/electricityloaddiagrams20112014
6) `Traffic`: https://pems.dot.ca.gov/

Table 4: Detailed dataset description. Size indicates the split lengths of individual points for training, validation and testing division respectively.

| Dataset | Variate number $D$ | Sampling frequency | Split size | Field |
|---------|--------------------|--------------------|------------|-------|
| ETTh1 | 7 | Hourly | (8640, 2880, 2880) | Energy |
| Exchange | 8 | Daily | (5311, 758, 1517) | Finance |
| Weather | 21 | 10min | (34560, 5760, 11520) | Weather |
| Appliance | 28 | 10min | (13814, 1973, 3947) | Energy |
| Electricity | 321 | Hourly | (17280, 2880, 5760) | Energy |
| Traffic | 862 | Hourly | (11520, 2880, 2880) | Traffic |

## A.5 EVALUATION METRICS

To assess the accuracy and reliability of estimated multivariate predictive distribution, we adopt four common metrics quantify both deterministic and probabilistic forecasting performance of generated prediction intervals. The MSE (Mean Squared Error) and MAE (Mean Absolute Error) are employed to quantify the mean difference between the obtained median forecast and true target. The CRPS (Continuous Ranked Probability Score) and CRPS_sum are employed to characterize the divergence between the generated prediction uncertainties and the real observed time series distribution. Suppose $\mathbf{y}_0$ is the ground-truth time series, $\{\hat{\mathbf{y}}_0^{(s)}\}_{s=1}^S$ is the produced prediction set, and let its $50\%$-quantile trajectory $\bar{\mathbf{y}}_0$ signify the point forecast, then two metrics can be calculated in a point-wise form over all channels and timestamps:

$$MSE = \frac{1}{HD} \left\| \mathbf{y}_0 - \bar{\mathbf{y}}_0 \right\|_2^2; \tag{16}$$

$$MAE = \frac{1}{HD} \left| \mathbf{y}_0 - \bar{\mathbf{y}}_0 \right|; \tag{17}$$

$$CRPS = \frac{1}{HD} \sum_{d=1}^{D} \sum_{t=1}^{H} \int_R (F(\hat{y}_{td}) - \mathbb{I}\{y_{td} \leq \hat{y}_{td}\})^2 \mathrm{d}\hat{y}_{td}; \tag{18}$$

$$CRPS\_sum = \frac{1}{H} \sum_{t=1}^{H} \int_R (F(\hat{z}_t) - \mathbb{I}\{z_t \leq \hat{z}_t\})^2 \mathrm{d}\hat{z}_t; \tag{19}$$

where $y_{td}$ indicates the $t$-th point value of the $d$-th univariate time series, $z_t = \sum_{i=1}^{D} y_{ti}$ is the sum of $D$ point observations at time $t$. $F$ is the empirical cumulative distribution function.

## A.6 EXPERIMENTAL CONFIGURATIONS

In Table 5, we detail the conditional diffusion model configurations on different forecasting scenarios, including the channel-aware DiT compositions and diffusion noise scheduling. We simply preserve the layers of input and output channel-independent dense encoders identical to the depth of attention modules, i.e. $n_{att} = n_{enc} = n_{dec} = 2$. One observation is that the designed channel-centric conditional denoising network can be easily scalable with diverse forecasting scenarios by

merely adjusting the hidden representation dimension $e_{hid}$, which changes compatibly with the prediction horizon $H$.

In Table 6, we shed light on the concrete contrastive training configurations for the main comparison outcomes presented in Table 1. We adopt the two-stage separate training on Weather, Electricity and Traffic datasets to reduce the training time and memory consumption. The best contrastive weight is chosen from $\{0.001, 0.0005, 0.0001, 0.00005\}$. Due to the GPU memory limitation, we have to turn down the negative sample number and batch size on Electricity and Traffic datasets with hundreds of channels, which could restrict the resulting final forecast performance. The initial learning rates are also displayed for the full reproduction on the newly adopted benchmark. We adopt the Adam optimizer with its weight decay of 1e-6 and a MultiStepLR learning rate scheduler to optimize the parameters of contrastive diffusion model.

Table 5: Diffusion forecaster configurations on different forecasting setups.

| Forecasting setup | | DiT blocks | | | Noise schedule (quadratic) | | |
|---|---|---|---|---|---|---|---|
| Lookback length $L$ | Prediction horizon $H$ | Depth $n_{att}$ | Heads | Hidden dim $e_{hid}$ | $\beta_1$ | $\beta_K$ | Steps $K$ |
| 48 | 96 | 2 | 8 | 128 | 0.0001 | 0.5 | 50 |
| 96 | 168 | 2 | 8 | 256 | 0.0001 | 0.2 | 100 |
| 192 | 336 | 2 | 8 | 512 | 0.0001 | 0.1 | 200 |
| 336 | 720 | 2 | 8 | 728 | 0.0001 | 0.1 | 200 |

## A.7 RUNTIME ANALYSIS

We compare the both training and inference time costs of disparate diffusion forecasters in Table 7. It is obvious that the auxiliary contrastive learning indeed aggravates the burden of vanilla denoising diffusion training for the sake of a higher quality of multivariate predictive distribution. Thus we adopt the two-stage separate strategy to accelerate the training process. The sequential generation procedure of our CCDM method is notably faster than other models, which indicates the designed channel-centric denoiser architecture can be efficiently scalable to diverse forecasting settings. Besides, the deterministic autoregressive pretraining in TimeDiff, hybrid attention layers in CSDI and point-wise amortized diffusion in TimeGrad can magnify their time consumption to different extents.

## A.8 MORE EXPERIMENTAL RESULTS

### A.8.1 ADDITIONAL COMPARISONS ON SHORT-TERM PROBABILISTIC FORECASTING

We also verify the capability of CCDM on short-term probabilistic forecasting in Table 8, at which previous diffusion forecasters are displayed to be adept. We follow the same setting in CSDI (Tashiro et al., 2021) with lookback window of 168 and prediction horizon of 24. Another two diffusion forecasters i.e. TimeDiT (Cao et al., 2024) and LDT (Feng et al., 2024), which similarly repurpose the DiT architecture are involved to show the advantage of the proposed channel-centric manipulation in temporal conditional denoising. We can see that CCDM consistently outperform other baselines on the short-term setup, which further validate the superior forecasting capacity of CCDM. Besides, we replace the CiDM module with residual connections by normal channel-independent linear layers as in iTransformer, and entice a moderate decline on prediction outcomes in Table 8. It reflects that adding residual shortcuts to channel-independent MLP encoders can indeed boost the expressivity for dynamic temporal variations, and verify the virtue of residual CiDM modules in TiDE (Das et al., 2023a) again.

### A.8.2 ADDITIONAL COMPARISONS WITH UP-TO-DATE BASELINE MODELS

To further verify the forecasting capability of CCDM, we involve extra four classes of up-to-date models for a more comprehensive comparison: 1) iTransformer (Liu et al., 2023), which devises a channel-centric transformer to capture the complex intra-variate and inter-variate temporal correlations; 2) SimMTM (Dong et al., 2024), which designs a mask-based pretraining scheme to refine the predictive learning process; 3) mr-Diff (Shen et al., 2023), which endows the useful multi-scale

Table 6: Contrastive training configurations corresponding to forecasting results in Table 1.

| Setup | | Contrastive weight $\lambda$ | Negative number $N$ | Batch size | Initial rate | Training mode |
|---|---|---|---|---|---|---|
| ETTh1 | 96 | 0.001 | 64*2 | 32 | 0.001 | End-to-end |
| | 168 | 0.001 | 64*2 | 32 | 0.001 | End-to-end |
| | 336 | 0.001 | 64*2 | 32 | 0.0002 | End-to-end |
| | 720 | 0.001 | 64*2 | 32 | 0.0002 | End-to-end |
| Exchange | 96 | 0.001 | 64*2 | 32 | 0.001 | End-to-end |
| | 168 | 0.001 | 64*2 | 32 | 0.001 | End-to-end |
| | 336 | 0.001 | 64*2 | 32 | 0.0002 | End-to-end |
| | 720 | 0.001 | 64*2 | 32 | 0.0002 | End-to-end |
| Weather | 96 | 0.001 | 64*2 | 32 | 0.0001 | Two-stage |
| | 168 | 0.001 | 64*2 | 32 | 0.0001 | Two-stage |
| | 336 | 0.0001 | 64*2 | 32 | 0.00002 | Two-stage |
| | 720 | 0.001 | 64*2 | 32 | 0.00002 | Two-stage |
| Appliance | 96 | 0.001 | 64*2 | 32 | 0.001 | End-to-end |
| | 168 | 0.001 | 64*2 | 32 | 0.001 | End-to-end |
| | 336 | 0.0001 | 64*2 | 32 | 0.0002 | End-to-end |
| | 720 | 0.00005 | 64*2 | 32 | 0.0002 | End-to-end |
| Electricity | 96 | 0.0001 | 32*2 | 24 | 0.0001 | Two-stage |
| | 168 | 0.00005 | 32*2 | 12 | 0.0001 | Two-stage |
| | 336 | 0.00005 | 32*2 | 8 | 0.00002 | Two-stage |
| | 720 | 0.00005 | 24*2 | 8 | 0.00002 | Two-stage |
| Traffic | 96 | 0.0005 | 32*2 | 4 | 0.0001 | Two-stage |
| | 168 | 0.0001 | 24*2 | 4 | 0.0001 | Two-stage |
| | 336 | 0.00005 | 16*2 | 4 | 0.00002 | Two-stage |
| | 720 | 0.00005 | 12*2 | 4 | 0.00002 | Two-stage |

Table 7: Time cost comparison of diffusion forecasters on different sizes of prediction tasks. Both training time [s] of one epoch and inference time [ms] of one step are provided.

| Size | | CCDM | | TimeDiff | | CSDI | | TimeGrad | |
|---|---|---|---|---|---|---|---|---|---|
| | | Train [s] | Infer [ms] | Train [s] | Infer [ms] | Train [s] | Infer [ms] | Train [s] | Infer [ms] |
| D=8 | H=96 | 18.67 | 3.63 | 14.11 | 3.00 | 4.78 | 3.63 | 2.22 | 349.42 |
| | H=168 | 28.11 | 4.37 | 18.56 | 3.00 | 6.33 | 3.58 | 3.89 | 603.05 |
| | H=336 | 80.78 | 4.71 | 24.00 | 2.98 | 10.67 | 3.72 | 5.32 | 1163.55 |
| | H=720 | 166.44 | 4.97 | 26.22 | 3.05 | 18.78 | 3.58 | 9.33 | 2571.37 |
| D=28 | H=96 | 66.67 | 3.76 | 25.44 | 4.00 | 34.89 | 3.76 | 7.67 | 374.23 |
| | H=168 | 203.11 | 4.38 | 37.11 | 3.94 | 50.78 | 3.68 | 12.22 | 605.80 |
| | H=336 | 441.74 | 4.71 | 33.22 | 4.29 | 97.22 | 3.66 | 21.67 | 1170.64 |
| | H=720 | 903.00 | 4.75 | 34.67 | 4.50 | 181.56 | 6.45 | 50.22 | 2551.13 |
| D=321 | H=96 | 573.22 | 4.59 | 657.67 | 17.92 | 84.78 | 9.08 | 48.56 | 357.51 |
| | H=168 | 1131.89 | 4.70 | 859.44 | 19.48 | 145.89 | 17.20 | 86.22 | 630.14 |
| | H=336 | 3173.89 | 4.83 | 1190.33 | 20.61 | 376.11 | 47.77 | 171.44 | 1188.07 |
| | H=720 | 4039.56 | 5.09 | 1269.56 | 22.71 | 546.67 | 70.13 | 330.67 | 2672.31 |
| D=862 | H=96 | 1466.14 | 4.54 | 185.78 | 46.67 | 104.22 | 25.12 | 80.56 | 369.04 |
| | H=168 | 1884.77 | 4.52 | 193.89 | 47.89 | 118.33 | 47.71 | 146.67 | 620.23 |
| | H=336 | 3202.85 | 5.17 | 284.89 | 49.09 | 228.44 | 96.06 | 289.11 | 1186.83 |
| | H=720 | 4678.78 | 7.86 | 463.56 | 55.01 | 417.33 | 193.62 | 545.67 | 2591.93 |

Table 8: Comparisons of short-term forecasting capacity on two datasets with $L = 168, H = 24$.

| Methods | Exchange | | | | Electricity | | | |
|---|---|---|---|---|---|---|---|---|
| | MSE | MAE | CRPS | CRPS_sum | MSE | MAE | CRPS | CRPS_sum |
| CCDM | **0.0309** | **0.1173** | **0.0917** | **0.5246** | **0.0881** | **0.1780** | **0.1325** | **9.9455** |
| CCDM-w/o CiDM | 0.0323 | 0.1205 | 0.0983 | 0.5576 | 0.1067 | 0.1998 | 0.1627 | 12.1184 |
| CSDI | 0.0704 | 0.1774 | 0.1393 | 0.7714 | 0.1117 | 0.2028 | 0.1580 | 13.4852 |
| TimeDiff | 0.0313 | 0.1257 | 0.1257 | 0.6857 | 0.1285 | 0.2512 | 0.2509 | 19.0025 |
| TimeDiT | 0.0657 | 0.1685 | 0.1252 | 0.7196 | 0.1066 | 0.1965 | 0.1507 | 12.9503 |
| LDT | 0.0656 | 0.1616 | 0.1125 | 0.6750 | 0.0998 | 0.1859 | 0.1473 | 12.7360 |

Table 9: Overall comparisons with four kinds of up-to-date forecasters w.r.t MSE and MAE on three real-world datasets. The best and second-best results are boldfaced and underlined.

| Methods | | CCDM | | iTransformer | | SimMTM | | mr-Diff | | Moirai | |
|---|---|---|---|---|---|---|---|---|---|---|---|
| Metrics | | MSE | MAE | MSE | MAE | MSE | MAE | MSE | MAE | MSE | MAE |
| ETTh1 | 96 | **0.3715** | **0.3900** | 0.4117 | 0.4159 | 0.3953 | 0.4069 | 0.4024 | 0.3987 | 0.4053 | 0.4087 |
| | 168 | **0.4137** | **0.4170** | 0.4515 | 0.4415 | 0.4335 | 0.4336 | 0.4397 | 0.4244 | 0.4505 | 0.4395 |
| | 336 | 0.5146 | 0.4629 | 0.5437 | 0.4905 | **0.4688** | **0.4551** | 0.4935 | 0.4558 | 0.5738 | 0.5076 |
| | 720 | 0.5545 | 0.5227 | 0.5746 | 0.5313 | **0.5228** | **0.5175** | 0.5621 | 0.5237 | 0.8054 | 0.6146 |
| | Avg | 0.4636 | **0.4482** | 0.4954 | 0.4698 | **0.4551** | 0.4533 | 0.4744 | 0.4507 | 0.5588 | 0.4926 |
| Weather | 96 | 0.2452 | **0.2320** | 0.2503 | 0.2710 | **0.2434** | 0.2524 | 0.3841 | 0.3515 | 0.2546 | 0.2634 |
| | 168 | **0.2407** | **0.2417** | 0.2774 | 0.2920 | 0.2585 | 0.2655 | 0.3563 | 0.3253 | 0.2749 | 0.2863 |
| | 336 | **0.2840** | **0.2814** | 0.3271 | 0.3267 | 0.3047 | 0.3113 | 0.4793 | 0.3745 | 0.3259 | 0.3223 |
| | 720 | 0.5599 | 0.4603 | 0.3768 | 0.3684 | **0.3552** | **0.3546** | 0.5031 | 0.4031 | 0.3842 | 0.3695 |
| | Avg | 0.3325 | 0.3039 | 0.3079 | 0.3145 | **0.2905** | **0.2960** | 0.4307 | 0.3636 | 0.3099 | 0.3104 |
| Electricity | 96 | 0.1987 | **0.2704** | 0.2011 | 0.2825 | 0.2261 | 0.3106 | **0.1960** | 0.3123 | 0.2065 | 0.2849 |
| | 168 | **0.1575** | **0.2481** | 0.1579 | 0.2554 | 0.1774 | 0.2800 | 0.1908 | 0.3037 | 0.1666 | 0.2614 |
| | 336 | **0.1651** | **0.2597** | 0.1666 | 0.2656 | 0.1826 | 0.2900 | 0.2048 | 0.3177 | 0.1726 | 0.2677 |
| | 720 | **0.1959** | **0.2858** | 0.1982 | 0.2947 | 0.2128 | 0.3138 | 0.2277 | 0.3344 | 0.1995 | 0.2960 |
| | Avg | 0.1793 | **0.2660** | **0.1810** | 0.2746 | 0.1997 | 0.2986 | 0.2048 | 0.3170 | 0.1863 | 0.2775 |
| 1st count | | **18** | | 1 | | 10 | | 1 | | 0 | |

temporal features into the cascaded time series diffusion model; 4) Moirai (Woo et al., 2024), which is a universal time series foundation model forged on large-scale datasets. We compare their deterministic forecasting capability on three real-world datasets and present the MSE and MAE results in Table 9. We can observe that CCDM and SimMTM can achieve the state-of-art and second-best ranks respectively. It reveals that designing complementary learning strategies like contrastive refinement in CCDM or masked pretraining in SimMTM beyond naive predictive training is able to enhance the forecasting capacity on specific time series.

### A.8.3 FULL RESULTS ON ABLATION STUDY

We illuminate the full forecasting outcomes corresponding to the ablation study of Section 4.3 in Table 10. In a nutshell, the performance promotion margins derived from such denoiser architecture and contrastive refinement innovations vary among different forecasting scenarios. It still requires careful settings on channel-aware denoising networks and auxiliary contrastive training to achieve the optimal results for a specific time series field and prediction setup.

## A.9 MORE ANALYSIS ON TEMPORAL CONTRASTIVE REFINEMENT

### A.9.1 NEGATIVE TIME SERIES AUGMENTATION

To enable contrastive learning for time series diffusion models, we consider the following four types of augmentation methods to produce negative sequences $\mathbf{y}_0^{(n)}$.

- Intra-series variation shuffling. It alters the ground-truth temporal variations of each univariate time series by patch shuffling, since recovering the correct dynamic evolution is a vital challenge for time series diffusion models. As shown in Fig. 6, we divide a given sequence into an array of sub-series patches and randomly shuffle their orders to change original temporal dependencies.

- Magnitude scaling. It scales up or scales down the magnitudes of individual time points, as an ideal prediction interval should well cover every point forecasts without any of them falling outside. Thus, for each positive target $\mathbf{y}_0$, we uniformly sample a scaling factor $a_d$ between $[0, 0.5] \cup [1.5, 2.0]$ and impose it on each channel by $a_d \cdot \mathbf{y}_0^d \in \mathbb{R}^H$.

- Jittering. It samples a random Gaussian noise from $\mathcal{N}(0, 0.3)$ and adds it to the ground-truth time series $\mathbf{y}_0$.

- Cutout. It randomly masks out the true values on $10\%$ timestamps from input $\mathbf{y}_0$ by zeros.

Table 10: Complete forecasting results by masking denoising-based temporal contrastive refinement or channel-mixing DiT blocks.

| Methods | | w/o contrastive refinement | | | | w/o channel-wise DiT | | | |
|---------|---|------|-------------|------|-------------|------|-------------|------|-------------|
| Metrics | | MSE | Degradation | CRPS | Degradation | MSE | Degradation | CRPS | Degradation |
| ETTh1 | 96 | 0.4447 | 19.70% | 0.3199 | 12.01% | 0.3903 | 5.06% | 0.2963 | 3.75% |
| | 168 | 0.5223 | 26.25% | 0.3402 | 12.39% | 0.5800 | 40.20% | 0.6674 | 120.48% |
| | 336 | 0.6416 | 24.68% | 0.3917 | 15.51% | 0.5381 | 4.57% | 0.4699 | 38.57% |
| | 720 | 0.5944 | 7.20% | 0.5038 | 3.75% | 0.8740 | 57.62% | 0.8928 | 83.86% |
| | Avg | 0.5508 | 18.81% | 0.3889 | 10.08% | 0.5956 | 28.47% | 0.5816 | 64.62% |
| Exchange | 96 | 0.1057 | 16.80% | 0.1677 | 8.54% | 0.0959 | 5.97% | 0.1598 | 3.43% |
| | 168 | 0.1986 | 21.25% | 0.2338 | 8.29% | 0.2200 | 34.31% | 0.2777 | 28.62% |
| | 336 | 0.4532 | 2.84% | 0.3557 | 1.14% | 0.4735 | 7.44% | 0.3870 | 10.04% |
| | 720 | 1.2290 | 5.18% | 0.6038 | 2.97% | 1.1802 | 1.00% | 0.5975 | 1.89% |
| | Avg | 0.4966 | 6.59% | 0.3403 | 4.04% | 0.4924 | 5.69% | 0.3555 | 8.68% |
| Weather | 96 | 0.2825 | 15.21% | 0.1936 | 6.02% | 0.2919 | 19.05% | 0.2012 | 10.19% |
| | 168 | 0.3349 | 39.14% | 0.2167 | 14.17% | 0.4199 | 74.45% | 0.2981 | 57.06% |
| | 336 | 0.2932 | 3.24% | 0.2313 | 3.72% | 0.3825 | 34.68% | 0.2873 | 28.83% |
| | 720 | 0.6158 | 9.98% | 0.4365 | 7.14% | 0.8428 | 50.53% | 0.5478 | 34.46% |
| | Avg | 0.3816 | 14.77% | 0.2695 | 7.50% | 0.4843 | 45.65% | 0.3336 | 33.07% |
| Appliance | 96 | 0.7097 | 13.97% | 0.4291 | 10.34% | 0.7473 | 20.01% | 0.4546 | 16.89% |
| | 168 | 0.7313 | 16.71% | 0.4374 | 8.81% | 0.7853 | 25.33% | 0.6070 | 51.00% |
| | 336 | 0.9254 | 1.48% | 0.5083 | 0.93% | 1.0660 | 16.90% | 0.6971 | 38.42% |
| | 720 | 1.7215 | 10.36% | 0.9525 | 10.83% | 1.8744 | 20.16% | 1.1336 | 31.91% |
| | Avg | 1.0220 | 9.86% | 0.5818 | 8.04% | 1.1183 | 20.21% | 0.7231 | 34.28% |
| Electricity | 96 | 0.2142 | 12.92% | 0.2266 | 10.75% | 0.2296 | 21.03% | 0.2198 | 7.43% |
| | 168 | 0.1689 | 7.24% | 0.2033 | 7.40% | 0.1779 | 12.95% | 0.2041 | 7.82% |
| | 336 | 0.1714 | 3.82% | 0.2035 | 2.62% | 0.1744 | 5.63% | 0.2048 | 3.28% |
| | 720 | 0.2002 | 2.19% | 0.2242 | 2.66% | 0.2073 | 5.82% | 0.2260 | 3.48% |
| | Avg | 0.1887 | 6.55% | 0.2144 | 5.77% | 0.1973 | 11.41% | 0.2137 | 5.43% |
| Traffic | 96 | 1.0345 | 0.52% | 0.4226 | 8.05% | 1.2831 | 24.68% | 0.4741 | 21.22% |
| | 168 | 0.6936 | 0.80% | 0.3113 | 1.17% | 0.7682 | 11.64% | 0.3869 | 25.74% |
| | 336 | 0.6913 | 3.44% | 0.3572 | 8.77% | 0.8472 | 26.77% | 0.4329 | 31.82% |
| | 720 | 0.9561 | 13.93% | 0.5610 | 30.34% | 1.1351 | 35.26% | 0.5762 | 33.88% |
| | Avg | 0.8439 | 4.68% | 0.4130 | 13.34% | 1.0084 | 25.08% | 0.4675 | 28.29% |

Table 11: Forecasting results by different negative time series augmentation methods on CCDM.

| Augmentation methods | MSE | MAE | CRPS | CRPS_sum |
|---------------------|------|------|------|----------|
| CCDM (Scaling+Variation) | 0.4137 | 0.4170 | 0.3027 | 1.3594 |
| Scaling | 0.4148 | 0.4173 | 0.3031 | 1.3584 |
| Variation | 0.4145 | 0.4184 | 0.3046 | 1.3645 |
| Jittering | 0.4236 | 0.4221 | 0.3070 | 1.3677 |
| Cutout | 0.4507 | 0.4381 | 0.3180 | 1.4534 |

We attest the effect of these four negative construction methods on ETTh1 dataset with $L = 96, H = 168$ and report prediction results in Table 11. We can find that utilizing scaling and variation augmentation methods incurs modestly better quality of prediction intervals than normal Gaussian jittering and zero cutout. Thus in the devised CCDM, we combine the scaling and variation methods to produce more informative negative instances at each diffusion step.

### A.9.2 INFLUENCE OF CONTRASTIVE LOSS FORM

As claimed in (Oord et al., 2018), the density ratio function $f(\cdot)$ in the softmax-formed InfoNCE loss can be any positive real-valued types. To seamlessly align with the standard denoising diffusion training paradigm, we specifically dictate $f(\cdot)$ using the step-wise noise regression form as presented in Eq. 3a, 3b, which adopts the same MSE loss to optimize the conditional denoising network. In fact, we can also prescribe $f(\cdot)$ as another similarity-based form, which is widely employed in vision

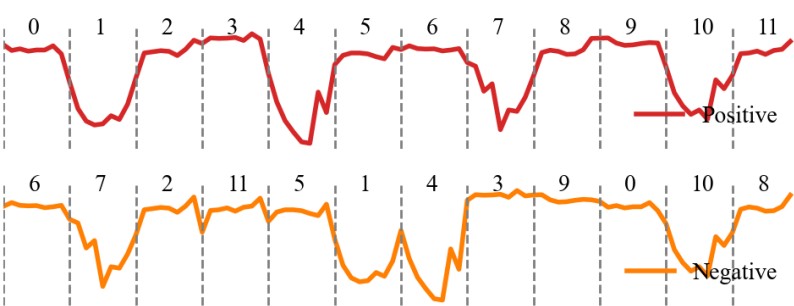

Figure 6: One diagram of the variation-based time series augmentation method.

Table 12: Forecasting results by two forms of density ratio functions in contrastive loss.

| Contrastive loss type | Exchange | | | | Electricity | | | |
|---|---|---|---|---|---|---|---|---|
| | MSE | MAE | CRPS | CRPS_sum | MSE | MAE | CRPS | CRPS_sum |
| MSE regression | 0.0830 | 0.1995 | 0.1480 | 0.8331 | 0.1987 | 0.2704 | 0.2046 | 20.6836 |
| Cosine similarity | 0.0864 | 0.2031 | 0.1508 | 0.8729 | 0.2010 | 0.2714 | 0.2056 | 20.9938 |

representation learning (Chen et al., 2020). We provide this similarity-typed design for density ratio function as follows:

$$f_{k,\boldsymbol{\epsilon}'}(\mathbf{y}_0, \mathbf{x}; \theta) = \exp(\mathrm{sim}(\boldsymbol{\epsilon}', \boldsymbol{\epsilon}_\theta(\sqrt{\bar{\alpha}_k}\mathbf{y}_0 + \sqrt{1 - \bar{\alpha}_k}\boldsymbol{\epsilon}', \mathbf{x}, k))/\tau); \tag{20a}$$

$$f_{k,\boldsymbol{\epsilon}'}(\mathbf{y}_0^{(n)}, \mathbf{x}; \theta) = \exp(\mathrm{sim}(\boldsymbol{\epsilon}', \boldsymbol{\epsilon}_\theta(\sqrt{\bar{\alpha}_k}\mathbf{y}_0^{(n)} + \sqrt{1 - \bar{\alpha}_k}\boldsymbol{\epsilon}', \mathbf{x}, k))/\tau); \tag{20b}$$

where $\mathrm{sim}(\cdot)$ indicates the cosine similarity between the ground-truth and predicted noise. The spirit of Eq. 20a, 20b is that the predicted noise of positive time series should be more similar to the imposed noise label $\boldsymbol{\epsilon}'$, while that for negative instances is repelled from the true $\boldsymbol{\epsilon}'$. We validate the efficacy of these two disparate density ratio forms for time series contrastive diffusion learning, and report the forecasting outcomes on two real-world datasets with $L = 48, H = 96$ in Table 12. We can easily see that the MSE noise regression form is slightly better than the cosine similarity type, which suggests that aligning the additional contrastive training with naive denoising manner is more effective to enhance time series diffusion models.

### A.9.3 INFLUENCE OF NEGATIVE NUMBER

It is claimed in previous works on visual contrastive representation learning Oord et al. (2018); Chen et al. (2020) that a larger number of negative samples within a training iteration can bring out more informative latent features for downstream vision recognition tasks. To probe the influence of number of negative sample $N$ on the specialized contrastive time series diffusion model for multivariate forecasting, we change $N$ in the range from 16 to 256 and showcase pertaining outcomes in Fig. 7. We can observe that the optimal $N$ is 192, 128 and 16 on three datasets and two quantitative metrics of each dataset exhibit distinctive changing trends. This phenomenon suggests that the real impact of negative sample number on contrastive training gains is relatively intractable, which is not amenable to the law in visual contrastive self-supervised pretraining. It could also be caused by the substantially smaller amount of training corpus in time series than images. We should determine the best number of negative instances in light of concrete data characteristics along with other training hyper-parameters. Even though we can not empirically derive a valid guideline to design the optimal contrastive training configuration, the remarkable forecasting outcomes achieved by CCDM in Table 1 and 9 can reveal that: simply instantiating CCDM using the uniform setting provided in Table 5 without any extra hyper-parameter search is sufficient to attain more excellent forecasting capability versus other baseline models on a wide variety of real-world datasets and prediction scenarios.

### A.9.4 INFLUENCE OF TEMPERATURE COEFFICIENT

The proposed denoising-based contrastive diffusion loss in Eq. 4 is in a canonical softmax form. According to the gradient analysis for the universal softmax-based contrastive loss in Wang & Liu

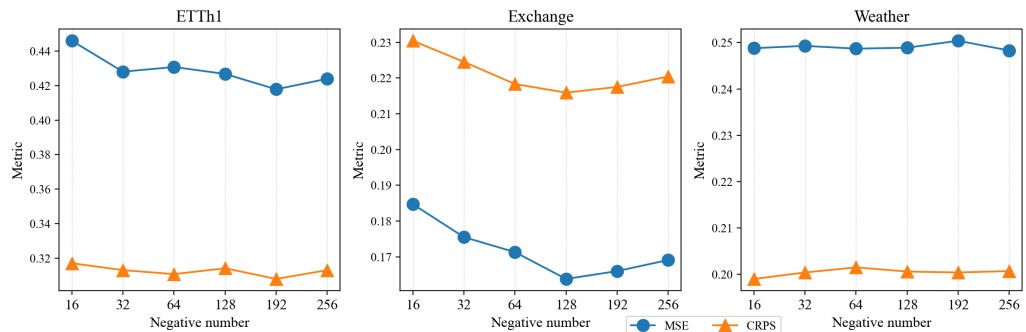

Figure 7: Forecasting results by different numbers of negative samples.

Table 13: Forecasting results by different temperature coefficients.

| Temperature $\tau$ | ETTh1 | | Exchange | | Weather | |
|---|---|---|---|---|---|---|
| | MSE | CRPS | MSE | CRPS | MSE | CRPS |
| 0.05 | 0.4391 | 0.3124 | 0.1728 | 0.2216 | 0.2515 | 0.2015 |
| 0.1 | 0.4267 | 0.3142 | 0.1638 | 0.2159 | 0.2489 | 0.2006 |
| 0.5 | 0.4730 | 0.3252 | 0.1583 | 0.2152 | 0.2493 | 0.2022 |
| 1.0 | 0.5082 | 0.3389 | 0.2031 | 0.2449 | 0.2505 | 0.2025 |

(2021), the temperature $\tau$ is a critical factor to control the penalty magnitude on various negative samples. To attain the contrastive improvement on conditional denoiser training, maintaining $\tau$ within an appropriate interval is significant. We assign four values to $\tau$ and illustrate quantitative results in Table 13. We can apparently observe that $[0.05, 0.1]$ could be a reasonable range on `ETTh1` and $[0.1, 0.5]$ is also valid for other two datasets.

## A.10    MORE ANALYSIS ON CHANNEL-AWARE DiT ARCHITECTURE

As discussed in Section 3.1, the proposed channel-aware DiT architecture mainly differs from existing time series denoising networks in two ways: 1) *Multi-head attention usage for temporal correlations modeling.* We alter the naive point-wise attention over the time dimension to a channel-wise attention along the variate axis. 2) *Conditioning scheme of past observed time series* $\mathbf{x}$. We directly concatenate the conditioning $\mathbf{x}$ with corrupted $\mathbf{y}_k$ and capture their temporal correlations by subsequent channel-wise DiT blocks. To demonstrate the impact of attention usage and past conditioning scheme separately, we compare three curated CCDM variants with DiT-based TimeDiT, LDT and attention-based CSDI on two real-world datasets. The respective average MSE and CRPS values over four prediction horizons are presented in Table 14.

In detail, the attention axis column in Table 14 contains two options, including channel-wise attention for inter-variate correlations or point-wise attention for intra-variate temporal dependencies. The conditioning scheme column consists of three entries: 1) The proposed $\mathbf{x} - \mathbf{y}_k$ mixing DiT. It concatenates the temporal encoding of past observed $\mathbf{x}$ and step-wise corrupted $\mathbf{y}_k$ along the channel dimension and feed-forward them into the follow-up DiT blocks to fully exploit the predictive information in $\mathbf{x}$. 2) Vallina adaLN DiT, which handles $\mathbf{x}$ and diffusion step embedding using the uniform linear adaLN layers. 3) 1D-CNN encoding, which simply processes the local features in $\mathbf{x}$ and adds it to $\mathbf{y}_k$ latent embedding. Note that to *ensure a fair architecture comparison*, the ad-hoc CCDM variants in top three lines, i.e. the devised channel-wise Mix-DiT in standard CCDM, variant point-wise Mix-DiT and channel-wise DiT are *trained without the auxiliary contrastive loss*.

According to the ablation study results in Table 14, we can observe that both channel-wise correlation modeling and $\mathbf{x} - \mathbf{y}_k$ mixing conditioning scheme indeed lead to more satisfactory forecasting results. In particular, the mixing conditioning regime can benefit the denoising network to a much larger margin than ordinary adaLN and 1D-CNN modules, which suggests that directly fusing $\mathbf{x}$ and $\mathbf{y}_k$ by DiT blocks can prevent from the potential predictive information loss. Besides, managing the

Table 14: Ablation studies on channel-wise attention and past conditioning schemes.

| | Architecture design | | Exchange | | | | Electricity | | | |
|---|---|---|---|---|---|---|---|---|---|---|
| Models | Attention axis | Past conditioning scheme | MSE | Degradation | CRPS | Degradation | MSE | Degradation | CRPS | Degradation |
| channel-wise Mix-DiT | channel | $\mathbf{x} - \mathbf{y}_k$ mixing DiT | **0.4699** | **0.00%** | **0.3403** | **0.00%** | **0.1887** | **0.00%** | **0.2144** | **0.00%** |
| point-wise Mix-DiT | time | $\mathbf{x} - \mathbf{y}_k$ mixing DiT | 0.5379 | 14.47% | 0.3583 | 5.29% | 0.1929 | 2.23% | 0.2279 | 6.30% |
| channel-wise DiT | channel | adaLN DiT | 0.5699 | 21.28% | 0.3958 | 16.31% | 0.2141 | 13.46% | 0.2382 | 11.10% |
| TimeDiT | time | adaLN DiT | 0.6374 | 35.65% | 0.4209 | 23.68% | 0.2598 | 37.68% | 0.2967 | 38.39% |
| LDT | time | adaLN DiT | 0.6119 | 30.22% | 0.4084 | 20.01% | 0.2329 | 23.42% | 0.2813 | 31.20% |
| CSDI | time+channel | 1D-CNN encoding | 0.7649 | 62.78% | 0.5051 | 48.43% | 0.4012 | 112.61% | 0.3069 | 43.14% |

complex temporal properties from a channel-centric perspective in diffusion forecasting can mitigate the side effect of noise injection training and give rise to higher-quality prediction intervals.

### A.11 MORE SHOWCASES ON PREDICTION INTERVALS

In Fig. 8-13 below, we visualize more prediction intervals generated by the proposed CCDM on six datasets. The legend for each figure is identical to Fig. 4. For each task's result visualization, we just display the first 7 or 8 variates and present two random samples on the $L = 48, H = 96$ setting. Moreover, in Fig. 14 below, we visually compare the quality of prediction intervals and point forecasts produced by four different models on each channel of ETTh1. We can clearly see that the prediction intervals generated by contrastive diffusion CCDM hold better accuracy, sharpness and reliability to encompass the real observations versus other models. We can also observe that the faithfulness of the approximated predictive distribution can be enhanced after introducing auxiliary contrastive training to time series diffusion models.

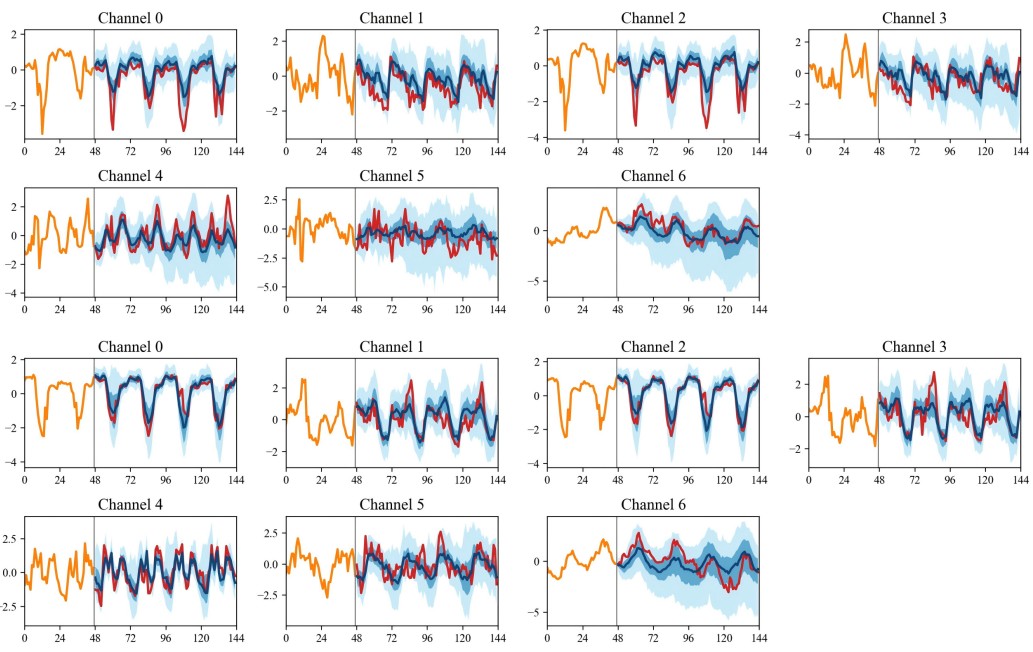

Figure 8: ETTh1 prediction intervals of total 7 channels.

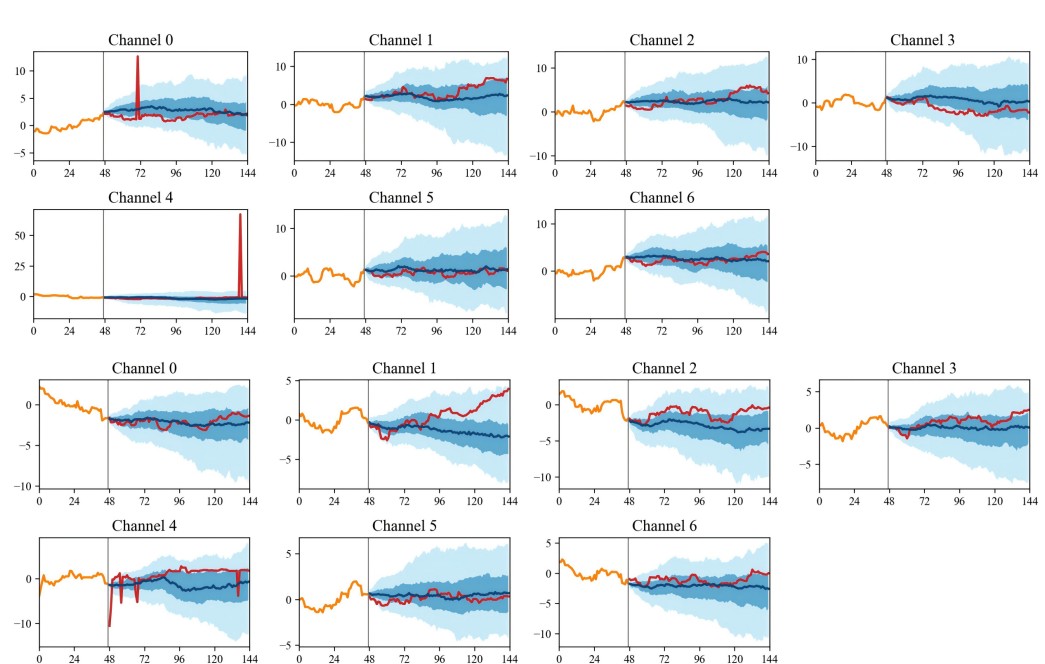

Figure 9: Exchange prediction intervals of total 7 channels.

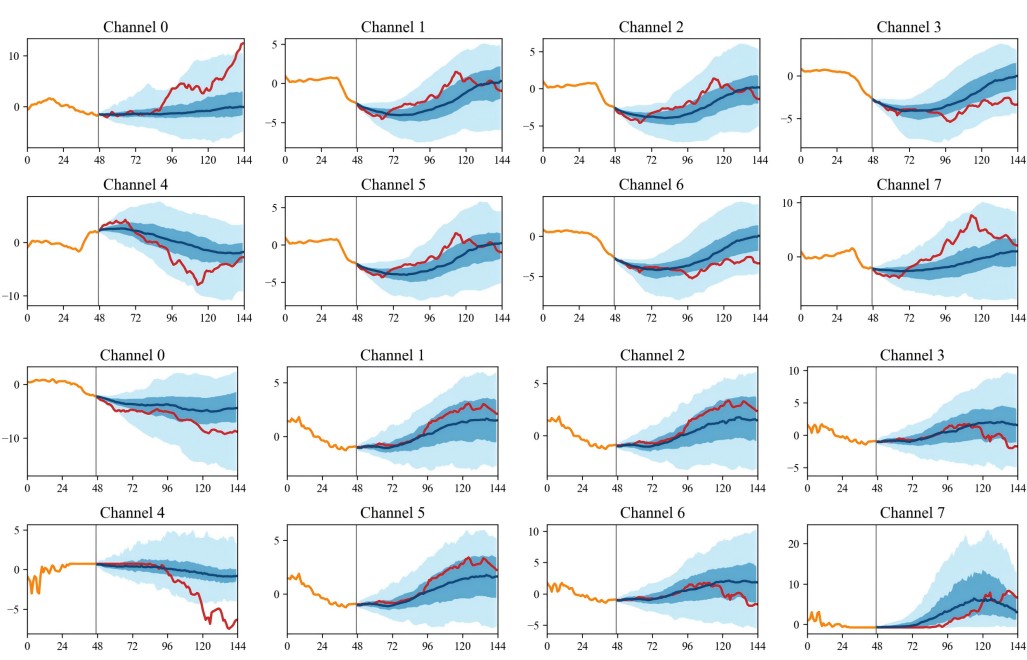

Figure 10: Weather prediction intervals of first 8 channels.

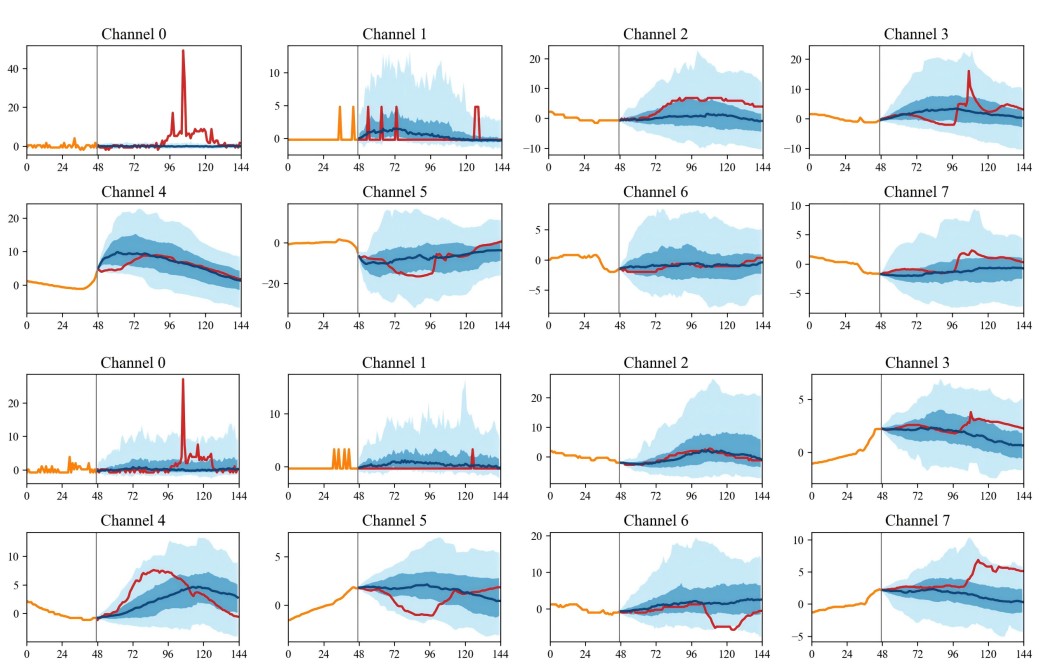

Figure 11: Appliance prediction intervals of first 8 channels.

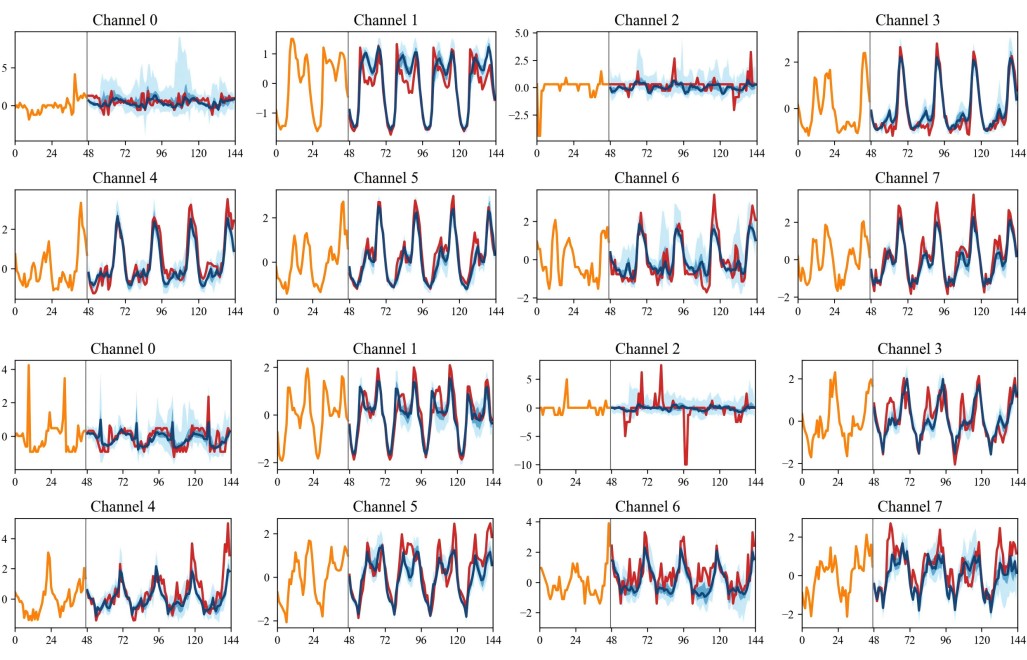

Figure 12: Electricity prediction intervals of first 8 channels.

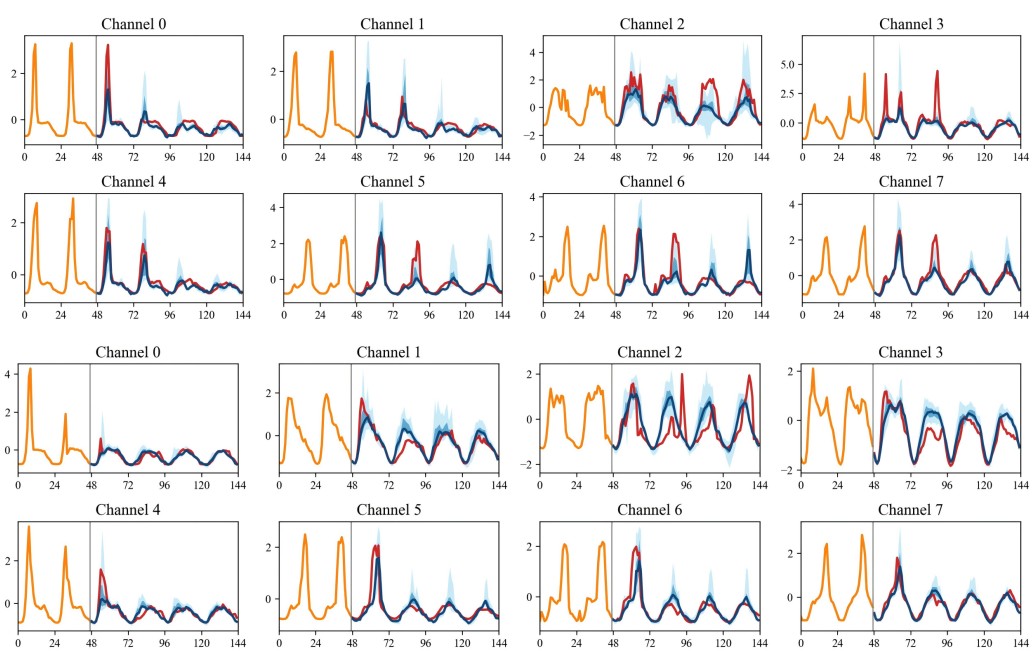

Figure 13: Traffic prediction intervals of first 8 channels.

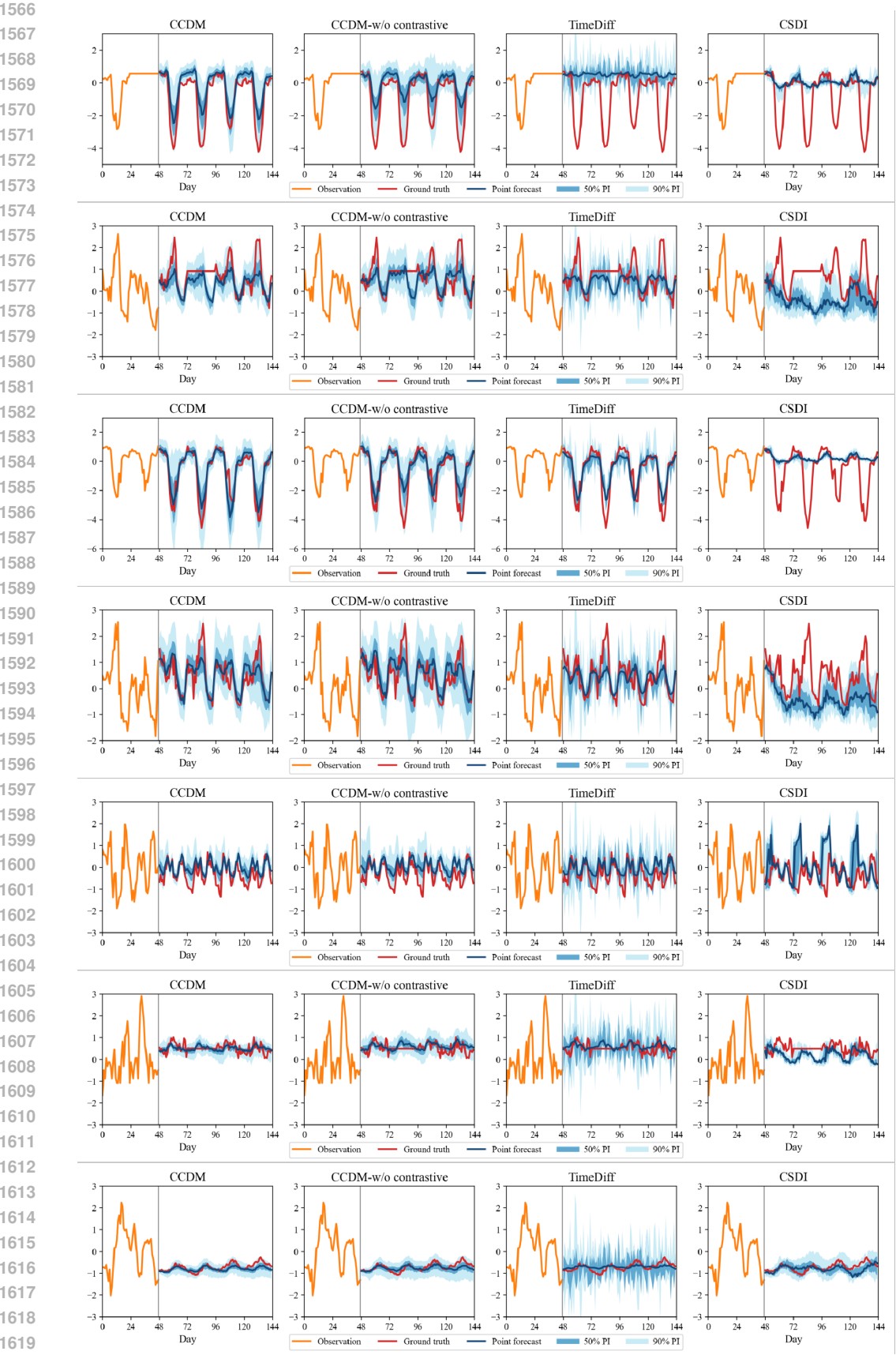

Figure 14: Comparison of generated point forecasts and prediction intervals on 7 ETTh1 channels.

