# OpenReview forum: "Channel-aware Contrastive Conditional Diffusion for Multivariate Probabilistic Time Series Forecasting"
_ICLR.cc/2025/Conference — Submitted to ICLR 2025_

### Official Review · Reviewer_4FVe · 2024-10-23

**Soundness:** 1
**Presentation:** 2
**Contribution:** 1
**Rating:** 3
**Confidence:** 4

**Summary:**

This paper innovatively designs a composite channel-aware conditional denoising network (CCDM) that merges a channel-independent dense encoder to extract univariate dynamics and a channel diffusion transformer to aggregate cross-variate correlations. And a time-contrast learning based on self-organized denoising is designed to explicitly amplify the predictive mutual information between past observations and future predictions. The method can consistently complement the plain stepwise denoising diffusion training to improve the prediction accuracy and generalization of unknown test time series.

**Strengths:**

Learning generalisable multivariate probability distributions based on a step-wise denoising paradigm is a pivotal problem where there are two challenging obstacles, firstly how to establish cross-channel temporal correlation connections between observed and denoising predicted sequences, and subsequently how to improve the capability to mine implicit representations from a given time series.

To address these two issues, the authors propose two innovative architectures:

* The channel-independent dense encoder and the channel-diffusion transformer to extract univariate and aggregated cross-variate correlations, respectively.

* Self-organized denoising-based temporal contrast learning to consistently complement plain stepwise denoising-diffusion training.

**Weaknesses:**

**Weakness 1:**

CCDM proposes contrastive learning for self-organizing denoising as a consistent complement to naive stepwise denoising diffusion training. Contrastive learning is a typical discriminative self-supervised learning. Compared with generative self-supervised learning, contrastive learning only needs to determine whether the sample is positive or not, and the low difficulty of the training task may lead to the model not being fully trained.

In addition, CCDM establishes the training paradigm based on a contrastive learning strategy that maximizes the mutual information between input and output. However, the model will be applied to the generative task based on prediction. The discriminative representation information learned by the model during the training phase varies dramatically for downstream generation tasks. To the best of our knowledge, large models trained in a self-supervised manner often need to design efficient fine-tuning strategies to adapt them to specific downstream tasks, while CCDMS directly reason on concrete datasets after being trained with pairwise learning.

**Weakness 2:**

There has been a lot of research (Crossformer[1], LIFT[2], SAMformer[3]) on building cross-channel connections between multivariate time series. The hybrid channel-aware denoising architecture proposed by CDDM includes two main structures, Channel-wise Diffusion Transformer and Channel-independent Dense Module (CiDM). CiDM, which is not originally from this article but from TiDE[4], The Channel-wise Diffusion Transformer looks like it combines the Dimension-Segment-Wise (DSW) structure designed in Crossformer[1] with the conditional diffusion model. These shortcomings make CCDM an incremental summary work rather than innovative research.

**Weakness 3:**

The lack of advanced baselines leads to the inability to verify the competitiveness of the proposed CCDM. Specifically, only four baselines based on Diffusion are shown in Table 1, and both SSSD and CSDI are published in 2021.

More extensive performance comparisons are expected to fully validate the effectiveness of the proposed method. The proposed baseline can be introduced in five sections:

* Models based on pre-trained LLM alignment to TS, OneFitsAll, TimeLLM, LLM4TS.

* Pre-trained time series base models on unified time series datasets from multiple domains, such as Timer, Moriai, Moment, UniTime.

* Small datasets to train and test on specific datasets, e.g. PatchTST, iTransformer[5], ModernTCN[6].

* Recent diffusion-based time series probabilistic prediction models, such as Diffuison-TS[7], mr-Diff[8], MG-TSD[9].

* Self-supervised prediction models based on contrastive learning and mask reconstruction SimMTM[10].

The CCDM is expected to be compared with at least one competitive model in each forecasting paradigm to indicate the plausibility of the model design.

Even compared with the four baselines in Table 1, CCDM has almost no performance improvement on all six datasets. Specifically, on the ETTh dataset, the MSE of CCDM is only 0.0017 lower than that of TimeDiff. On the Exchange, Appliance and Weather datasets, CCDM shows worse performance than TimeDiff and CSDI. On Electricity and Traffic datasets, the average improvement of CCDM in MSE metric is less than 10%.

**Weakness 4:**

In the ablation experiment (Table 2), after removing contrastive refinement, the model improves only 1.2% and 1.08% on ECL and Traffic datasets.

In addition, another confusing result is that ETTh and Weather have a small number of channels (7 and 21), while ECL has a huge number of channels (321). However, after removing the channel-wise design, On the contrary, the performance in ETTh and Weather drastically decreases (23.34% and 40.28% in MSE), and the performance in the ECL barely changes (5.71% in MSE).

This may indicate that channel-wise designs fail to capture cross-channel latent representations from datasets with a large number of channels and instead perform well in other datasets with a smaller number of channels. The results of ablation experiments are negative and difficult to understand for the main thrust of this paper, "establishing cross-channel connections in Diffusion models".

**Reference:**

1) Zhang, Yunhao and Junchi Yan. “Crossformer: Transformer Utilizing Cross-Dimension Dependency for Multivariate Time Series Forecasting.” ICLR 2023.

2) Ilbert, Romain et al. “SAMformer: Unlocking the Potential of Transformers in Time Series Forecasting with Sharpness-Aware Minimization and Channel-Wise Attention.” ICML 2024.

3) Zhao, Lifan and Yanyan Shen. “Rethinking Channel Dependence for Multivariate Time Series Forecasting: Learning from Leading Indicators.” ICLR 2024

4) Das, Abhimanyu et al. “Long-term Forecasting with TiDE: Time-series Dense Encoder.” ArXiv abs/2304.08424 (2023): n. pag.

5) Liu, Yong et al. “iTransformer: Inverted Transformers Are Effective for Time Series Forecasting.” ArXiv abs/2310.06625 (2023): n. pag.

6) Luo, Donghao and Xue Wang. “ModernTCN: A Modern Pure Convolution Structure for General Time Series Analysis.” ICLR 2024.

7) Yuan, Xinyu and Yan Qiao. “Diffusion-TS: Interpretable Diffusion for General Time Series Generation.” ICLR 2024.

8) Shen, Lifeng et al. “Multi-Resolution Diffusion Models for Time Series Forecasting.” ICLR 2024.

9) Fan, Xinyao et al. “MG-TSD: Multi-Granularity Time Series Diffusion Models with Guided Learning Process.” ICLR 2024.

10) Dong, Jiaxiang et al. “SimMTM: A Simple Pre-Training Framework for Masked Time-Series Modeling.” NIPS 2023.

**Questions:**

**Question 1:**

See Weakness 3 for our concerns about the experimental results, and we believe that the introduction of competitive and up-to-date baselines is considered necessary.  In addition, the authors are expected to explain the phenomena in the ablation experiments, details of which can be referred to Weakness 4.

**Question 2:**

It is mentioned in the main text that "density ratio function can be any types of positive real functions". Is there a clear strategy to determine the best density ratio function for different time-series data domains? Does designing different density ratio functions indirectly affect the training effect? How to construct similar and dissimilar instances to prevent model collapse? We believe that the construction of negative samples is critical to applying CCDM to a wide and diverse real-world setting.

**Question 3:**

As far as we know, contrastive learning tends to consume a lot of computational resources. Specifically, we often need tens of thousands of iterations to fully train the diffusion model, and the total training overhead is staggering if we need to generate a set of n negative samples based on each positive example at each iteration. Therefore, theoretical analysis complexity and experimental results on time overhead of CCDM are necessary for their potential real-world applications.

---

> ### Author Response · Authors · 2024-11-23
> **Response to Reviewer 4FVe (Part 1)**
>
> *Thank you very much for your careful review and precious advice. We have made great effort to address your questions and concerns.*
>
> **W1:** **Concerns on the motivation and behaviors of the proposed complementary contrastive learning.**
>
> Indeed, contrastive learning have been shown to discover effective vision or language representations by learning to discriminate between positive and negative sample pairs. But this might not be a valid manner as well in time series forecasting, since time series do not have the explicit and human-recognizable temporal structures, like the definite semantics of image and text data. In this work, we aim to explicitly maximize the mutual predictive information between limited past observations and future forecasts, which is beneficial to improve the accuracy and reliability of the generated prediction intervals. To achieve this goal, we specifically design the time series contrastive learning adapted from the classic InfoNCE loss to implement the intention of mutual prediction-related information maximization. Meanwhile, in order to improve the training efficiency of standard denoising diffusion, we particularly endow a step-wise noise regression form into the InfoNCE loss which can be seamlessly consistent with the naive denoising training manner. The additional contrastive training can help time series diffusion models to discern the low-probability areas (constituted by diffusion paths of negative samples) in the target predictive distribution and keep away from them. Besides, CCDM can be learned end-to-end on specific datasets without laborious self-supervised pretraining on large-scale corpus and fine-tuning on downstream tasks. We additionally provide more detailed theoretical interpretations on the benefits of the devised information-theoretic contrastive diffusion training in $\underline{\text{Appendix A.1.2 and Theoretical Insights Section of the revised paper}}$.
>
> **W2:** **Insufficient statement on the novelty of the proposed channel-wise denoising network.**
>
> As the reviewer said, channel-centric time series structure design has been broadly investigated in the field of multivariate deterministic forecasting, such as the excellent work of iTransformer and Crossformer. But this issue has not been well explored in the specific multivariate diffusion-based probabilistic forecasting task. In this area, a key architectural challenge is how to design an effective conditional time series denoising network, which is able to identify the faithful multi-channel temporal properties with varying degrees of diffused noise disturbing each variate sequences. Improper treatment for the diffused noise can cause training instability and hurt the capacity to tackle long-term dependencies and complex inter-channel correlations. In this regard, we propose a composite channel-aware conditional denoiser with both channel-independent CiDM and channel-mixing DiT. It can tackle the side effect of noise corruption and recover the plausible heterogeneous temporal correlations.

---

> > ### Comment · Reviewer_4FVe · 2024-11-26
> >
> > Thank you for your response. I have carefully reviewed the author's rebuttal and other reviewer comments. My main worry  and concern of confusion remains unresolved:
> >
> > 1. Contrastive learning, as a discriminative self-supervised pre-training paradigm, is not suitable for the generative downstream paradigm based on prediction. Therefore, the existing pre-training paradigm based on contrastive learning performs poorly on prediction tasks. The authors need to intuitively explain **why CCDM achieves good performance on prediction tasks even though it is based on contrastive learning paradigm?** For example, CCDM designs a discriminative/generative hybrid self-supervised paradigm.
> >
> > 2. Regarding whether CCDM is innovative at the methodological level? As mentioned in Weakness 2, the channel diffusion transformer simply combines the **DSW module (proposed by Crossformer)** with the conditional diffusion model, and **CiDM also comes from TiDE (As described in the manuscript)**, so the authors need to explain the innovative contribution of CCDM from the most naive point of view of modeling time series.
> >
> > 3. Regarding the experimental results of CCDM presented in Table 1 (in fact, the authors did not number the tables during the rebuttal, we strongly encourage the authors to do so), **CCDM showed lower performance than SimMTM on average across all three datasets**. And we observe that the prediction error of CCDM is much larger than that of SimMTM on long-term prediction metrics (e.g. ETTh1-720 and Weather-720). Does this indicate that **CCDM is difficult to capture long-term time dependence?** Moreover, Electricity-avg in Table 1 seems to suffer from labeling errors. The author should take part in the discussion more seriously.
> >
> > 4. Regarding the phenomenon in ablation experiments. According to the authors, **the channel-wise module, which is designed for multivariate time series prediction, shows worse results when dealing with datasets with huge channel dimensions (ECL,Traffic)**. However, iTransformer, which is also oriented to modeling large number of inter-channel, shows excellent performance on ECL and Traffic datasets. Therefore we believe that "the channel dimension in ECL is considerably larger than that in ETTh1 and Weather" is not a reasonable explanation. The authors need more theoretical basis or experimental data to support their views.
> >
> > 5. A discussion of the time cost. Discussions about time overhead and computational complexity are crucial for long-term time series forecasting tasks, so **"not straightforward to conduct theoretical time complexity analysis"** does not excuse the authors. In fact, the complexity analysis of time series only needs to consider the time step size, the model dimension, and the number of diffusion-denoising iterations. As we worry, contrastive learning needs to build a set of **N** negative samples for each positive sample in each iteration, and CCDM as a diffusion model often needs additional **T** rounds of iterative noise reduction, so it is necessary to analyze the time complexity of the training process and inference process of CCDM from a theoretical point of view.
> >
> > In fact, since the authors **did little to address our concerns and confusions regarding CCDM**, we cautiously decided to temporarily downgrade the CCDM score.

---

> ### Author Response · Authors · 2024-11-23
> **Response to Reviewer 4FVe (Part 2)**
>
> **W3.1\&Q1.1:** **More comparisons with up-to-date competitive forecasting models.**
>
> To further verify the forecasting capability of CCDM, we implement extra four classes of baseline models for a more comprehensive comparison, including iTransformer [5], SimMTM [6], mr-Diff [7] and Moirai [8]. We compare their deterministic forecasting capability on three real-world datasets and present the MSE and MAE results as follows:
>
> | Methods |  | CCDM |  | iTransformer |  | SimMTM |  | mr-Diff |  | Moirai |  |
> | :---: | :---: | :---: | :---: | :---: | :---: | :---: | :---: | :---: | :---: | :---: | :---: |
> | Metrics |  | MSE | MAE | MSE | MAE | MSE | MAE | MSE | MAE | MSE | MAE |
> | ETTh1 | 96 | **0.3715** | **0.3900** | 0.4117 | 0.4159 | 0.3953 | 0.4069 | 0.4024 | 0.3987 | 0.4053 | 0.4087 |
> |  | 168 | **0.4137** | **0.4170** | 0.4515 | 0.4415 | 0.4335 | 0.4336 | 0.4397 | 0.4244 | 0.4505 | 0.4395 |
> |  | 336 | 0.5146 | 0.4629 | 0.5437 | 0.4905 | **0.4688** | **0.4551** | 0.4935 | 0.4558 | 0.5738 | 0.5076 |
> |  | 720 | $\underline{0.5545}$ | $\underline{0.5227}$ | 0.5746 | 0.5313 | **0.5228** | **0.5175** | 0.5621 | 0.5237 | 0.8054 | 0.6146 |
> |  | Avg | $\underline{0.4636}$ | **0.4482** | 0.4954 | 0.4698 | **0.4551** | 0.4533 | 0.4744 | 0.4507 | 0.5588 | 0.4926 |
> | Weather | 96 | $\underline{0.2452}$ | **0.2320** | 0.2503 | 0.2710 | **0.2434** | 0.2524 | 0.3841 | 0.3515 | 0.2546 | 0.2634 |
> |  | 168 | **0.2407** | **0.2417** | 0.2774 | 0.2920 | 0.2585 | 0.2655 | 0.3563 | 0.3253 | 0.2749 | 0.2863 |
> |  | 336 | **0.2840** | **0.2814** | 0.3271 | 0.3267 | 0.3047 | 0.3113 | 0.4793 | 0.3745 | 0.3259 | 0.3223 |
> |  | 720 | 0.5599 | 0.4603 | 0.3768 | 0.3684 | **0.3552** | **0.3546** | 0.5031 | 0.4031 | 0.3842 | 0.3695 |
> |  | Avg | 0.3325 | $\underline{0.3039}$ | 0.3079 | 0.3145 | **0.2905** | **0.2960** | 0.4307 | 0.3636 | 0.3099 | 0.3104 |
> | Electricity | 96 | $\underline{0.1987}$ | **0.2704** | 0.2011 | 0.2825 | 0.2261 | 0.3106 | **0.1960** | 0.3123 | 0.2065 | 0.2849 |
> |  | 168 | **0.1575** | **0.2481** | 0.1579 | 0.2554 | 0.1774 | 0.2800 | 0.1908 | 0.3037 | 0.1666 | 0.2614 |
> |  | 336 | **0.1651** | **0.2597** | 0.1666 | 0.2656 | 0.1826 | 0.2900 | 0.2048 | 0.3177 | 0.1726 | 0.2677 |
> |  | 720 | **0.1959** | **0.2858** | 0.1982 | 0.2947 | 0.2128 | 0.3138 | 0.2277 | 0.3344 | 0.1995 | 0.2960 |
> |  | Avg | $\underline{0.1793}$ | **0.2660** | **0.1810** | 0.2746 | 0.1997 | 0.2986 | 0.2048 | 0.3170 | 0.1863 | 0.2775 |
> | 1st count |  | **18** |  | 1 |  | $\underline{10}$ |  | 1 |  | 0 |  |
>
>
> We can observe that CCDM and SimMTM can achieve the state-of-art and second-best ranks respectively. It reveals that designing complementary learning strategies like contrastive refinement in CCDM or masked pretraining in SimMTM versus standard predictive training is able to enhance the forecasting capacity on specific time series. This justifies the design principles in our work.
>
> We have incorporated these additional comparison outcomes in $\underline{\text{Appendix A.8.2 and Table 9 of the revised paper}}$.
>
> **W3.2\&W4.1:** **Improvement on CCDM forecasting results presented in Table 1, 2.**
>
> Following the training implementation in CSDI, we add a MultiStepLR scheduler in Pytorch to adaptively adjust the learning rate during contrastive diffusion training. Besides, we modulate the weight decay of Adam optimizer to $1e-6$, which can regularize the network parameters learned by CCDM using L2-norm to avoid overfitting. These two operations can further improve the contrastive training stability and efficiency, as the step-wise noise injection in diffusion and the erratic variability of real-world time series could render time series diffusion hard to attain the optimal learning effectiveness. Furthermore, we slightly increase the batch size and negative sample number for Electricity and Traffic datasets. We present both the updated and original average results in Table 1 and performance decrease by w/o contrastive learning in Table 2 as follows:
>
> | Outcomes | Updated version in revised paper |  |  |  | Initial version in original paper |  |  |  |
> | :---: | :---: | :---: | :---: | :---: | :---: | :---: | :---: | :---: |
> | Dataset | MSE | CRPS | MSE decrease | CRPS decrease | MSE | CRPS | MSE decrease | CRPS decrease |
> | ETTh1 | 0.4636 | 0.3533 | 18.81\% | 10.08\% | 0.4769 | 0.3600 | 15.97\% | 8.55\% |
> | Weather | 0.3325 | 0.2507 | 14.77\% | 7.50\% | 0.3414 | 0.2563 | 13.00\% | 4.77\% |
> | Appliance | 0.9303 | 0.5385 | 9.86\% | 8.04\% | 0.9543 | 0.5454 | 7.18\% | 5.99\% |
> | Electricity | 0.1771 | 0.2027 | 6.55\% | 5.77\% | 0.1864 | 0.2111 | 1.20\% | 1.57\% |
> | Traffic | 0.8062 | 0.3644 | 4.68\% | 13.34\% | 0.8346 | 0.3762 | 1.08\% | 8.73\% |
>
>
> We can see that CCDM outcomes exhibit a moderate growth after we refine the training settings. The detailed overall comparison and component ablation results have been updated in $\underline{\text{Table 1, 2, 10 of the revised paper}}$.

---

> ### Author Response · Authors · 2024-11-23
> **Response to Reviewer 4FVe (Part 3)**
>
> **W4.2\&Q1.2:** **Explanation on the results of channel-aware DiT ablation study for ECL dataset.**
>
> There could be two possible reasons to explain this phenomenon: 1) As the channel dimension in ECL is considerably larger than that in ETTh1 and Weather, it would be much harder for channel-wise attention to discover the plausible cross-channel correlations by diffusion training on limited ECL data. 2) According to the claims in Section 5.3.2 of a nice benchmark paper [1], the inter-variate correlations in ECL are relatively weaker than those in another two datasets. Thus, adding channel-wise blocks to capture the cross-channel temporal properties on time series with weak variate correlations may not bring a benefit as prominent as that with strong variate correlations.
>
> **Q2.1:** **Influence of different forms of density ratio functions on temporal contrastive learning.**
>
> To seamlessly align with the standard denoising diffusion training paradigm, we specifically dictate the density ration function $f(\cdot)$ using the step-wise noise regression form, which adopts the same MSE loss to optimize the conditional denoising network. In fact, we can also prescribe $f(\cdot)$ as another similarity-based form, which is widely employed in vision representation learning [3]. We present this similarity-based design for density ratio function as:
>
> $$
> f_{k,\epsilon{}'}(\mathbf{y} _{0},\mathbf{x};\theta)=\exp (\mathrm{sim}(\epsilon{}', \epsilon _{\theta}(\sqrt{\bar{\alpha} _{k}}\mathbf{y} _{0}+\sqrt{1-\bar{\alpha} _{k}} \epsilon{}' ,\mathbf{x},k))/\tau);
> $$
>
> $$
> f_{k,\epsilon{}'}(\mathbf{y} _{0}^{(n)},\mathbf{x};\theta)=\exp (\mathrm{sim}(\epsilon{}', \epsilon _{\theta}(\sqrt{\bar{\alpha} _{k}}\mathbf{y} _{0}^{(n)}+\sqrt{1-\bar{\alpha} _{k}} \epsilon{}' ,\mathbf{x},k))/\tau);
> $$
>
> where $\mathrm{sim}(\cdot)$ indicates the cosine similarity between the ground-truth and predicted noise. The essence of above two equations is that the predicted noise of positive time series should be more similar to the imposed noise label $\epsilon{}'$, while that for negative instances is repelled from the true $\epsilon{}'$. We validate the efficacy of these two disparate density ratio forms for time series contrastive diffusion learning, and report the forecasting outcomes on two real-world datasets with $L=48, H=96$ as follows:
>
> | Contrastive loss type | Exchange |  |  |  | Electricity |  |  |  |
> | :---: | :---: | :---: | :---: | :---: | :---: | :---: | :---: | :---: |
> |  | MSE | MAE | CRPS | CRPS\_sum | MSE | MAE | CRPS | CRPS\_sum |
> | Noise regression | 0.0830 | 0.1995 | 0.1480 | 0.8331 | 0.1987 | 0.2704 | 0.2046 | 20.6836 |
> | Cosine similarity | 0.0864 | 0.2031 | 0.1508 | 0.8729 | 0.2010 | 0.2714 | 0.2056 | 20.9938 |
>
>
> We can clearly observe that the MSE noise regression form is slightly better than the cosine similarity type, which suggests that aligning the additional contrastive training with naive denoising manner is more effective to enhance time series diffusion models.
>
> We have contained this part of discussions in $\underline{\text{Appendix A.9.2 and Table 12 of the revised paper}}$.

---

> ### Author Response · Authors · 2024-11-23
> **Response to Reviewer 4FVe (Part 4)**
>
> **Q2.2:** **Comparisons for diverse types of negative time series augmentation methods.**
>
> To enable contrastive learning for time series diffusion models, we consider the following four types of augmentation methods to produce negative sequences, and the latter two operations are borrowed from [2]:
>
> - **Intra-series variation shuffling.** It alters the ground-truth temporal variations along each channel by patch shuffling, since recovering the correct dynamic evolution is a vital challenge for time series diffusion models.
> - **Magnitude scaling.** It scales up or scales down the magnitudes of individual time points, as an ideal prediction interval should well cover every point forecasts without any of them falling outside. Thus, we uniformly sample a scaling factor between $\left [ 0, 0.5 \right ] \cup \left [ 1.5, 2.0 \right ]$ and multiply each univariate time series by this factor.
> - **Jittering.** It samples a random Gaussian noise from $\mathcal{N}(0, 0.3)$ and adds it to the ground-truth time series.
> - **Cutout.** It randomly masks out the true values on $10\%$ timestamps from real sequences by zeros.
>
> We attest the effect of these four negative construction methods on ETTh1 dataset with $L=96, H=168$ and report prediction results below:
>
> | Augmentation methods | MSE | MAE | CRPS | CRPS\_sum |
> | :---: | :---: | :---: | :---: | :---: |
> | CCDM (Scaling+Variation) | 0.4137 | 0.4170 | 0.3027 | 1.3594 |
> | Scaling | 0.4148 | 0.4173 | 0.3031 | 1.3584 |
> | Variation | 0.4145 | 0.4184 | 0.3046 | 1.3645 |
> | Jittering | 0.4236 | 0.4221 | 0.3070 | 1.3677 |
> | Cutout | 0.4507 | 0.4381 | 0.3180 | 1.4534 |
>
>
> We can find that utilizing scaling and variation augmentation methods incurs modestly better quality of prediction intervals than normal Gaussian jittering and zero cutout. Thus in the devised CCDM, we combine the scaling and variation methods to produce more informative negative instances at each diffusion step. We also integrate this clarification on negative time series augmentations in $\underline{\text{Section A.9.1 and Table 11 of the revised paper}}$.
>
> **Q3:** **Time overhead analysis on contrastive training.**
>
> As the reviewer commented, introducing additional contrastive learning to time series diffusion forecasters can give rise to larger computational time consumption. We just leverage this auxiliary scheme to trade training time efficiency for prediction accuracy. It is not straightforward to conduct theoretical time complexity analysis for contrastive training, since it could be affected by a variety of potential factors, including the volume of dataset, batch size in each iteration, negative sample number, model architecture, hardware capacity and so on. In $\underline{\text{Section A.7 and Table 7 of the revised paper}}$, we present both training and inference time of CCDM versus other baseline models like other contrastive time series forecasting work have implemented [4]. And corresponding model dimension, batch size and negative sample size are provided in $\underline{\text{Table 5, 6 of the revised paper}}$. We can see that although CCDM costs more training time, its sequential inference procedure is notably faster than other models. Besides, for the large-scale Electricity and Traffic dataset, we only impose contrastive training to the last 20 epochs to reduce the time overhead.

---

> ### Author Response · Authors · 2024-11-23
> **Response to Reviewer 4FVe (Part 5)**
>
> [1] Xiangfei Qiu, Jilin Hu, Lekui Zhou, Xingjian Wu, Junyang Du, Buang Zhang, Chenjuan Guo, Aoying Zhou, Christian S. Jensen, Zhenli Sheng and Bin Yang. TFB: Towards Comprehensive and Fair Benchmarking of Time
> Series Forecasting Methods. PVLDB, 17(9): 2363 - 2377, 2024.
>
> [2] Luo D, Cheng W, Wang Y, et al. Time series contrastive learning with information-aware augmentations. Proceedings of the AAAI Conference on Artificial Intelligence. 2023, 37(4): 4534-4542.
>
> [3] Ting Chen, Simon Kornblith, Mohammad Norouzi, and Geoffrey Hinton. A simple framework for contrastive learning of visual representations. In International conference on machine learning, pp. 1597–1607. PMLR, 2020.
>
> [4] Gerald Woo, Chenghao Liu, Doyen Sahoo, Akshat Kumar, and Steven Hoi. Cost: Contrastive learning of disentangled seasonal-trend representations for time series forecasting. In International Conference on Learning Representations, 2021.
>
> [5] Liu Y, Hu T, Zhang H, et al. itransformer: Inverted transformers are effective for time series forecasting. arXiv preprint arXiv:2310.06625, 2023.
>
> [6] Dong J, Wu H, Zhang H, et al. Simmtm: A simple pre-training framework for masked time-series modeling. Advances in Neural Information Processing Systems, 2024, 36.
>
> [7] Shen L, Chen W, Kwok J. Multi-Resolution Diffusion Models for Time Series Forecasting. The Twelfth International Conference on Learning Representations. 2024.
>
> [8] Woo G, Liu C, Kumar A, et al. Unified training of universal time series forecasting transformers. arXiv preprint arXiv:2402.02592, 2024.
>
> *We sincerely hope our responses address some of your concerns. If you have any further questions, please feel free to ask.*

---

### Official Review · Reviewer_D78R · 2024-10-31

**Soundness:** 3
**Presentation:** 2
**Contribution:** 2
**Rating:** 3
**Confidence:** 3

**Summary:**

This work proposes CCDM which is able to train a diffusion model for time series forecasting via the combination of a denoising loss and a contrastive loss, which they call a "neural mutual information" perspective.  This allows for the training of probabilistic forecasts in a deep learning setting, following other recent works.  By allowing for probabilistic forecasts, they are able to train a more accurate diffusion model when compared to historical regression-only style models.  Comparing across standard benchmark datasets and against several other baseline methods, favorable performance is achieved in the MSE (regression) metric and CRPS (probabilistic) metric.  Several ablation studies then confirm that the contrastive loss function is useful in diffusion model training.

**Strengths:**

The topic of how to combine the diffusion models of deep learning with the probabilistic forecasting of time series seems like an important and emerging area.

The specific success of contrastive learning in aiding the diffusion model in CRPS performance is demonstrated through ablation experiments.

Good performance is achieved in many datasets and on many tasks.

**Weaknesses:**

The theoretical insights do not seem cohesive.

The contribution seems extremely simple or are not well highlighted.  After reading, I have the understanding that the major or only contribution is the application of contrastive learning to diffusion models, as depicted in Figure 1.  If that is the case, it is not properly highlighted and the challenges are not sufficiently discussed.

Further probabilistic analysis like the one in Figure 4 could be helpful for emphasizing the impact of this work.

**Questions:**

Can you clarify what is the difference between your proposed "neural mutual information perspective" and the combination of a denoising loss and contrastive loss?

Can you explain what the contributions are compared to the existing works [1] and [2]?  From reading the related work section, I got the impression that the other diffusion-based methods which "repurpose DiT" should be the closest related works; however, getting to the experiments it seems these two methods are not compared.

Is it possible to provide error bars for Figure 5? I understand the claim is that contrastive learning is helpful; however, looking at these figures, I get the completely opposite impression that actually the contrastive learning is doing almost nothing.  Perhaps it is possible to further explain these results.

If a key contribution of the work is the method to apply contrastive learning in the time series domain, can you elaborate on what the specific challenges of applying contrastive learning to time series data are?


[1] "Timedit: General-purpose diffusion transformers for time series foundation model" Defu Cao et al. 2024.

[2] "Latent diffusion transformer for probabilistic time series forecasting" Shibo Feng et al. 2024

---

> ### Author Response · Authors · 2024-11-23
> **Response to Reviewer D78R (Part 1)**
>
> *Many thanks for your meticulous review and insightful questions. In the following, we make great effort to address each of your concerns and questions.*
>
> **W1:** **The theoretical insights do not seem cohesive.**
>
> Both [1] and [2] have demonstrated that denoising diffusion models can benefit from additional contrastive training, which can lead to a higher-quality approximation for the target distribution, but they work from two distinct perspectives to interpret this merit. Specifically, the former reference starts from the view of neural mutual information maximization to state two complementary training objectives, see **Q1** response below for details. While the latter reference stands from the view of out-of-distribution (OOD) evaluation, claiming that learning on extra negative samples can help diffusion models to recognize the low-probability regions in the target distribution. Since the less powerful training objectives and distribution shifts are actually two key issues in the scope of time series diffusion forecasting, the Theoretical Insight part just claims that introducing contrastive learning can tackle these two problems simultaneously.
>
> **W2:** **Insufficient statement on major challenges and contributions.**
>
> Here, we want to highlight the main contributions, as well as clarifying the related challenges tackled by this paper more clearly as follows:
>
> - **First composite channel-aware architecture for temporal conditional denoising.** In the time series forecasting community, it remains an open issue on the design of effective conditional denoising network, which is able to identify the faithful channel-specific and cross-channel temporal properties with varying degrees of diffused noise disturbing each variate sequences. To this end, we propose a composite channel-aware conditional denoising network consisting of both channel-independent dense encoders and channel-mixing diffusion transformers, which can tackle the side effect of noise corruption and recover the plausible heterogeneous temporal correlations.
> - **Novel integration of time series contrastive learning with diffusion forecasting for more efficient exploitation of useful predictive information.** Given limited historical time series, it is an under-explored issue to find effective schemes for enhancing the utilization efficiency of the implicit temporal predictive information. The auxiliary training methods leveraged by existing diffusion forecasters need to expose prior knowledge on task-specific temporal properties, and can be inconsistent with standard step-wise denoising training. To this end, we devise a complementary denoising-based temporal contrastive refinement to boost diffusion training efficiency and explicitly amplify the prediction-related mutual information between generated forecasts and past observations.
> - **Comprehensive evaluation on multivariate probabilistic forecasting.** Previous generative forecasting methods only verify their performance on short-term settings and do not release their capability on more challenging scenarios with long-term dependencies. To this end, we construct a more comprehensive benchmark with varying prediction horizons and channel numbers to completely compare different diffusion forecasters' capacity. The proposed CCDM can achieve the state-of-art performance on diverse setups.
>
> We have restated the key challenges, motivations and contributions in $\underline{\text{Introduction section of the revised paper}}$.
>
> **W3:** **More probabilistic showcases like Figure 4.**
>
> In $\underline{\text{Figure 14 of the revised paper}}$, we visually compare the quality of prediction intervals and point forecasts produced by four different models on each channel of ETTh1. We can clearly see that the prediction intervals generated by contrastive diffusion CCDM hold better accuracy, sharpness and reliability to encompass the real observations versus other models. We can also observe that the faithfulness of the approximated predictive distribution can be enhanced after introducing auxiliary contrastive training to time series diffusion models.

---

> ### Author Response · Authors · 2024-11-23
> **Response to Reviewer D78R (Part 2)**
>
> **Q1:** **Clarifying the difference between "neural mutual information perspective" and the combination of denoising and contrastive loss?**
>
> According to the motivation clarified in **W2** response, we aim to explicitly maximize the prediction-related mutual information between past observations $\mathbf{x}$ and future forecasts $\mathbf{y} _{0}$. As claimed in [1], standard generative diffusion learned by log-likelihood maximization (i.e. $\max p _{\theta}(\mathbf{y} _{0}|\mathbf{x})$) and derived denoising loss is a forward predictive way to maximize the mutual information between $\mathbf{x}$ and $\mathbf{y} _{0}$. [1] both theoretically and empirically proved that combining predictive learning and contrastive learning can enhance the effectiveness of neural mutual information maximization on target tasks. Inspired by this, we propose to complement the contrastive learning to naive denoising diffusion learning. Besides, we specifically design a novel noise regression form for the canonical InfoNCE contrastive loss to make the additional contrastive training seamlessly align with original denoising training.
>
> **Q2:** **Distinctions and comparisons with DiT-based TimeDiT [5] and LDT [6].**
>
> In addition to the novel contrastive learning framework, the proposed composite channel-aware DiT distinctly leverage the unified channel-centric strategy to model the multivariate predictive distribution, and can differ from TimeDiT and LDT in two ways below:
>
> - **Multi-head attention usage for temporal correlations modeling.** We switch the naive point-wise attention over the time dimension to a channel-wise attention along the variate axis. It allows to represent complex cross-channel correlations in given conditioning time series $\mathbf{x}$ and corrupted targets $\mathbf{y} _{k}$, apart from the temporal dependencies captured by channel-independent dense encoders.
> - **Incorporation scheme of conditioning time series.** We directly concatenate the conditioning $\mathbf{x}$ with corrupted $\mathbf{y} _{k}$ and capture their temporal features by the subsequent channel-wise attention. This operation can fully exploit the useful predictive information in limited historic observations. Whereas TimeDiT and LDT simply pass the given $\mathbf{x}$ to linear adaLN layers, which may cause the predictive information loss for DiT blocks.
>
> Furthermore, we compare CCDM with TimeDiT and LDT on Exchange and Electricity datasets with lookback window of 168 and prediction horizon of 24. The comparison outcomes are presented below:
>
> | Methods | Exchange |  |  |  | Electricity |  |  |  |
> | :---: | :---: | :---: | :---: | :---: | :---: | :---: | :---: | :---: |
> |  | MSE | MAE | CRPS | CRPS\_sum | MSE | MAE | CRPS | CRPS\_sum |
> | CCDM | **0.0309** | **0.1173** | **0.0917** | **0.5246** | **0.0881** | **0.1780** | **0.1325** | **9.9455** |
> | TimeDiT | 0.0657 | 0.1685 | 0.1252 | 0.7196 | 0.1066 | 0.1965 | 0.1507 | 12.9503 |
> | LDT | 0.0656 | 0.1616 | 0.1125 | 0.6750 | 0.0998 | 0.1859 | 0.1473 | 12.7360 |
>
>
> We can see that channel-aware CCDM can obviously outperform TimeDiT and LDT. We have updated above discussion on DiT adaptation differences and comparisons in $\underline{\text{Section 3.1 and Appendix A.8.1 of the revised paper}}$.
>
> **Q3:** **Explaining Figure. 5 more clearly to demonstrate the positive effect of contrastive learning.**
>
> Following the reviewer's suggestion, we have replotted Figure. 5 to exhibit the positive effect of contrastive learning more clearly. As shown in $\underline{\text{Fig. 5 of the revised paper}}$, where we display MSE, MAE, CRPS and CRPS\_sum incurred by assigning various values to the contrastive weight $\lambda$. Fig. 5 aims to illustrate two points: 1) how different weights (i.e. $\lambda=0.01, 0.005, 0.001, 0.0005, 0.0001$) of the proposed contrastive loss in Eq. 6 can influence the diffusion forecasting capacity. 2) Adding contrastive training to naive diffusion predictive learning can actually enhance the forecasting capability, as the four metric outcomes of $\lambda=0$ (i.e. without contrastive refinement) is significant larger than those of $\lambda>0$ (i.e. imposing different degrees of contrastive learning).

---

> ### Author Response · Authors · 2024-11-23
> **Response to Reviewer D78R (Part 3)**
>
> **Q4:** **Elaboration on the specific challenges of applying contrastive learning to time series modeling.**
>
> There are two specific challenges on integrating contrastive learning to time series generative modeling:
>
> - **Constructing effective negative time series instances to inform temporal contrastive learning.** As claimed in [3], [4], how to select appropriate time series augmentation methods is a key factor for ultimate contrastive learning effect, since time series do not have diverse and human-recognizable temporal structures like images and languages. In this work, we consider four kinds of augmentation methods including intra-series variation shuffling, scaling, jittering and cutout to produce negative samples for contrastive diffusion training. More detailed discussions can be found in $\underline{\text{Section A.9.1 and Table 11 of the revised paper}}$.
> - **Adapting an efficient task-specific contrastive learning objective for the target time series analysis task.** In this specific probabilistic forecasting scenario, we wish to maximally utilize the beneficial predictive information in historical conditioning time series. For this purpose, we specifically leverage an information-theoretical contrastive learning objective to explicitly maximize the prediction-related mutual information between past observations and future forecasts. Besides, we specifically prescribe this contrastive loss as a noise regression form to seamlessly align with the standard denoising diffusion training. More detailed explanations can be seen in $\underline{\text{Appendix A.9.2 and Table 12 of the revised paper}}$.
>
> [1] Yao-Hung Hubert Tsai, Yue Wu, Ruslan Salakhutdinov, and Louis-Philippe Morency. Self-supervised learning from a multi-view perspective. In International Conference on Learning Representations, 2020.
>
> [2] Yunshu Wu, Yingtao Luo, Xianghao Kong, Evangelos E Papalexakis, and Greg Ver Steeg. Your diffusion model is secretly a noise classifier and benefits from contrastive training. arXiv preprint arXiv:2407.08946, 2024.
>
> [3] Luo D, Cheng W, Wang Y, et al. Time series contrastive learning with information-aware augmentations. Proceedings of the AAAI Conference on Artificial Intelligence. 2023, 37(4): 4534-4542.
>
> [4] Woo G, Liu C, Sahoo D, et al. Cost: Contrastive learning of disentangled seasonal-trend representations for time series forecasting. arXiv preprint arXiv:2202.01575, 2022.
>
> [5] Cao D, Ye W, Zhang Y, et al. Timedit: General-purpose diffusion transformers for time series foundation model. arXiv preprint arXiv:2409.02322, 2024.
>
> [6] Feng S, Miao C, Zhang Z, et al. Latent diffusion transformer for probabilistic time series forecasting. Proceedings of the AAAI Conference on Artificial Intelligence. 2024, 38(11): 11979-11987.
>
> *Thank you again for your thorough review, and we hope our responses can address your questions.*

---

> > ### Comment · Reviewer_D78R · 2024-11-23
> >
> > Thank you for your detailed response. After a review of the draft modifications and author response, I have come to understand the two major contributions of the work are 1. the novel integration of contrastive learning for time series (e.g. through the equations in Figure 1) and 2. composite channel-aware conditioning for integration of the diffusion model. Although the experiments do make significant progress towards these two goals, I do not think the writing or positioning of the paper clarifies this. In particular, for point 1, I do not feel that the discussion of the neural mutual information maximization perspective provides a clearer understanding of contrastive learning (Q1) or that the theoretical results are clarified why they are useful. For point 2, the architectural changes with respect to other architectures is not sufficiently highlighted. Although in your author response, you have provided promising benchmarks against the existing Dit approaches, this alone does not demonstrate which of your two major components is contributing to these benefits. Other concerns like evaluated Dit approaches on a different task than your main paper and the lack of confidence intervals for Figure 5 contribute to the feeling that the paper does not sufficiently prove its claims.

---

> > > ### Author Response · Authors · 2024-11-25
> > > **Response to Reviewer D78R (Part 1)**
> > >
> > > We greatly appreciate for your instant response and insightful suggestions to improve this paper from the side of contribution demonstration. Hope what we present below can address your concerns and questions.
> > >
> > > **Q1:** **More lucid justification on the benefits of contrastive diffusion learning from the proposed neural mutual information perspective.**
> > >
> > > Here, we demonstrate that the proposed information-theoretic denoising-based contrastive learning can improve the maximum likelihood training of diffusion models, which can give rise to more effective generative forecasting learning. In $\underline{\text{Appendix A.1.2 and Theoretical Insights Section of the revised paper}}$, we prove that maximizing $I _{\theta}(\mathbf{y} _{0}; \mathbf{x})$ is equivalent to minimizing KL-divergence $\mathcal{D} _{KL} \left[ q(\mathbf{y} _{0} | \mathbf{x}) || p _{\theta}(\mathbf{y} _{0}|\mathbf{x}) \right]$ between the real predictive distribution and diffusion-model-approximated distribution. It is well-known that minimizing $\mathcal{D} _{KL} \left[ q(\mathbf{y} _{0}|\mathbf{x})||p _{\theta}(\mathbf{y} _{0}|\mathbf{x}) \right]$ can be an efficient surrogate for the maximum likelihood learning to improve the log-likelihood $\log p _{\theta} (\mathbf{y} _{0}|\mathbf{x})$ of diffusion models [1], [2]. It can lead to better log-likelihood for diffusion models since vanilla combination of an array of weighted noise regression losses in Eq. 1 can not directly optimize the log-likelihood $\log p _{\theta}(\mathbf{y} _{0}|\mathbf{x})$ [2]. Besides, [1], [2] have demonstrated that integrating the maximum likelihood training manner with the naive score matching (i.e. noise regression) objective can acquire a significantly better generation quality. As learning the faithful predictive likelihood is necessary for time series probabilistic forecasting [3], complementing mutual information-theoretic contrastive training can gain better likelihood and thus improve the forecasting capacity of time series diffusion models. We hope these proof and interpretations could address the reviewer's concerns on the understanding of contrastive learning benefits from a mutual information perspective.
> > >
> > > **Q2:** **More detailed analysis on the contributions of individual architectural components.**
> > >
> > > As discussed in **1st-round Q2** response above, the proposed channel-aware DiT architecture mainly differs from existing time series denoising networks in two ways: 1) *Multi-head attention usage for temporal correlations modeling.* We alter the naive point-wise attention over the time dimension to a channel-wise attention along the variate axis. 2) *Conditioning scheme of past observed time series $\mathbf{x}$.* We directly concatenate the conditioning $\mathbf{x}$ with corrupted $\mathbf{y} _{k}$ and capture their temporal correlations by subsequent channel-wise DiT blocks. To demonstrate the impact of attention usage and past conditioning scheme separately, we compare three curated CCDM variants with DiT-based TimeDiT, LDT and attention-based CSDI on two real-world datasets following the same setup in the main paper. The respective average MSE and CRPS values over four prediction horizons are presented as follows:
> > >
> > > | Datasets | Architecture design |  | Exchange |  |  |  | Electricity |  |  |  |
> > > | :---: | :---: | :---: | :---: | :---: | :---: | :---: | :---: | :---: | :---: | :---: |
> > > | Models | Attention axis | Past conditioning scheme | MSE | Degradation | CRPS | Degradation | MSE | Degradation | CRPS | Degradation |
> > > | channel-wise Mix-DiT | channel | $\mathbf{x}-\mathbf{y} _{k}$ mixing DiT | **0.4699** | **0.00\%** | **0.3403** | **0.00\%** | **0.1887** | **0.00\%** | **0.2144** | **0.00\%** |
> > > | point-wise Mix-DiT | time | $\mathbf{x}-\mathbf{y} _{k}$ mixing DiT | 0.5379 | 14.47\% | 0.3583 | 5.29\% | 0.1929 | 2.23\% | 0.2279 | 6.30\% |
> > > | channel-wise DiT | channel | adaLN DiT | 0.5699 | 21.28\% | 0.3958 | 16.31\% | 0.2141 | 13.46\% | 0.2382 | 11.10\% |
> > > | TimeDiT | time | adaLN DiT | 0.6374 | 35.65\% | 0.4209 | 23.68\% | 0.2598 | 37.68\% | 0.2967 | 38.39\% |
> > > | LDT | time | adaLN DiT | 0.6119 | 30.22\% | 0.4084 | 20.01\% | 0.2329 | 23.42\% | 0.2813 | 31.20\% |
> > > | CSDI | time+channel | 1D-CNN encoding | 0.7649 | 62.78\% | 0.5051 | 48.43\% | 0.4012 | 112.61\% | 0.3069 | 43.14\% |

---

> > > ### Author Response · Authors · 2024-11-25
> > > **Response to Reviewer D78R (Part 2)**
> > >
> > > From the table we can see that the metric degradation values are computed based on the proposed channel-wise Mixing DiT in the first row. In detail, the attention axis column in this Table contains two options, including channel-wise attention for inter-variate correlations or point-wise attention for intra-variate temporal dependencies. The conditioning scheme column consists of three entries: 1) The proposed $\mathbf{x}-\mathbf{y} _{k}$ mixing DiT. It concatenates the temporal encoding of past observed $\mathbf{x}$ and step-wise corrupted $\mathbf{y} _{k}$ along the channel dimension and feed-forward them into the follow-up DiT blocks to fully exploit the predictive information in $\mathbf{x}$. 2) Vallina adaLN DiT, which handles $\mathbf{x}$ and diffusion step embedding using the uniform linear adaLN layers. 3) 1D-CNN encoding, which simply processes the local features in $\mathbf{x}$ and adds it to $\mathbf{y} _{k}$ latent embedding. Note that to *ensure a fair architecture comparison*, the ad-hoc CCDM variants in top three lines, i.e. the devised channel-wise Mix-DiT in standard CCDM, variant point-wise Mix-DiT and channel-wise DiT are *trained without the auxiliary contrastive loss*.
> > >
> > > According to the ablation study results in this Table above, we can observe that both channel-wise correlation modeling and $\mathbf{x}-\mathbf{y} _{k}$ mixing conditioning scheme indeed lead to more satisfactory forecasting results. In particular, the mixing conditioning regime can benefit the denoising network to a much larger margin than ordinary adaLN and 1D-CNN modules, which suggests that directly fusing $\mathbf{x}$ and $\mathbf{y} _{k}$ by DiT blocks can prevent from the potential predictive information loss. Besides, managing the complex temporal properties from a channel-centric perspective in diffusion forecasting can mitigate the side effect of noise injection training and give rise to higher-quality prediction intervals. Following the reviewer's suggestions, we have incorporated such analysis results in $\underline{\text{Appendix A.10 and Table 14 of the revised paper}}$. We hope these studies and clarifications would address the reviewer's concerns on the design of model architectures.
> > >
> > > **Q3:** **Lack of error bars/confidence intervals for Figure 5.**
> > >
> > > As the reviewer suggested, we calculate the error bars/confidence intervals for four metric values in Figure 5 by practicing CCDM of varying contrastive weights for 10 times re-training and independent evaluation runs, as prior diffusion forecasting work done [4]. To implement this, we set 10 different random seeds which can govern the random factors in each contrastive diffusion practice, such as diffusion noise sampling, denoiser parameters initialization and negative time series augmentations. According to $\underline{\text{Figure 5 of the revised paper}}$, we can clearly see that after adding auxiliary contrastive learning of different degrees (i.e. $\lambda > 0$), the mean and standard error of four evaluation metrics can exhibit noteworthy decreases in contrast to naive denoising training (i.e. $\lambda = 0$), which indicates the proposed contrastive training can improve the accuracy and robustness of diffusion forecasters.
> > >
> > > [1] Zhang J, Liu D, Zhang S, et al. Contrastive sampling chains in diffusion models. Advances in Neural Information Processing Systems, 2024, 36.
> > >
> > > [2] Song Y, Durkan C, Murray I, et al. Maximum likelihood training of score-based diffusion models. Advances in neural information processing systems, 2021, 34: 1415-1428.
> > >
> > > [3] Salinas D, Flunkert V, Gasthaus J, et al. DeepAR: Probabilistic forecasting with autoregressive recurrent networks. International journal of forecasting, 2020, 36(3): 1181-1191.
> > >
> > > [4] Xinyao Fan, Yueying Wu, Chang Xu, Yuhao Huang, Weiqing Liu, and Jiang Bian. Mg-tsd: Multi-granularity time series diffusion models with guided learning process. arXiv preprint arXiv:2403.05751, 2024.
> > >
> > > *We sincerely thank again for your valuable questions and we hope our responses can give you a better understanding of this work. We are really glad to talk about any related questions with you!*

---

> > > > ### Comment · Reviewer_D78R · 2024-11-27
> > > >
> > > > Although this most recent table is a solid analysis more clearly highlighting the specific benefits of this work, my remaining concerns and the unaddressed concerns of the other reviewers make me believe this paper is unfit to be accepted in its current form.  My major concern, as it has been from the beginning, is that the experiments can only possibly support the contribution of "contrastive learning improves time series prediction"; however, the experiments are not designed with this goal in mind.  For instance, any discussion of the augmentation methods/ negative sampling (a key component of contrastive methods) is only in the appendix.  Other concerns like not discussing architectural differences and unclear conclusions from theoretical insights leave the overall claims of the paper confused.  Hence despite being a promising direction of exploration, I am lowering my score to recommend rejection.
> > > >
> > > > To reiterate the most concrete of these concerns
> > > >
> > > > - The theoretical results provide no clear insights to the reader
> > > >
> > > > - My concerns about Figure 5 were covered up instead of being further explored
> > > >
> > > > - No comprehensive benchmarks on the most similar existing methods

---

### Official Review · Reviewer_GGe5 · 2024-11-02

**Soundness:** 3
**Presentation:** 3
**Contribution:** 3
**Rating:** 8
**Confidence:** 4

**Summary:**

The authors describe a novel diffusion model CCDM for multivariate probabilistic time series forecasting. They propose a channel-wise diffusion model with a novel training loss based on ideas from contrastive learning.
The training input $x$ and target time series $y_k$, degraded by the diffusion forward process at stage $k$, are first encoded with a known per-channel CiDM module, resulting in two equal-sized latent representations. They are concatenated and input to a standard channel-wise DiT. The transformer output is decoded yielding the estimated diffusion noise $\epsilon_k$.
Instead of training the model with the classical log-likelihood diffusion loss, the authors propose to add a contrastive loss term which takes into account one positive and $N$ negative target samples for each input sample utilizing the InfoNCE loss as introduced by Oord et. al. for general time series applications. They prove an upper bound of the denoising diffusion-induced forecasting error and finally propose an algorithm to generate the negative samples.
In experiments, the model shows improved performance compared to SOTA models on different datasets. An ablation study shows the influence of the contrastive weight on the performance and reveals that the choice of the number of negative samples is critical and needs to be found empirically by hyperparameter search.

**Strengths:**

The idea of extending the diffusion loss with a contrastive loss is new and interesting. The architecture of the model is built from known building blocks, but not known in this combination. The authors give a thorough prove concerning the properties of the forecasting error. They present a comprehensive bibliography.

**Weaknesses:**

A general weakness of the interesting idea is that (p 20, section A10) the real effect of $N$ is intractable and the optimal value must be found by hyperparameter search.

**Questions:**

p 3: $y_0$ and $y_K$ is introduced one section too late

p 4: wording: accounts for … and can be any types of … (unclear)

p 5: replacing point-wise attention by channes-wise attention. (Could you elaborate this shortly?)

p 6 l 275: both a positive sample

p 7 l 327: efficiency

p 9: While the tables in the appendix show the influence of the constrastive learning, it is really hard to interpret the positive effect from figure 5. Can this be stated more clearly?

---

> ### Author Response · Authors · 2024-11-23
> **Response to Reviewer GGe5**
>
> *We are grateful for your positive support and beneficial comments. Below we address each question and concern.*
>
> **W1:** **Concerns on extra hyper-parameter searching for effective contrastive refinement.**
>
> As the review suggested,  empirically determining the optimal contrastive training configuration can vary case by case. However, the remarkable forecasting outcomes achieved by CCDM as shown in $\underline{\text{Table 1, 9 of the revised paper}}$ can reveal that: simply initializing CCDM using the uniform setting provided in Table 5 without any extra hyper-parameter search is sufficient to acquire more excellent forecasting capability versus other baseline models on a wide variety of real-world datasets and prediction scenarios. We have added this part of discussions to $\underline{\text{Appendix A.9.3 of the revised paper}}$.
>
> **Q1:** **p3: $\mathbf{y} _{0}$ and $\mathbf{y} _{K}$ is introduced one section too late.**
>
> $\mathbf{y} _{0}$ denotes the clean ground-truth time series at diffusion step 0, while $\mathbf{y} _{K}$ indicates the prior Gaussian noise at diffusion step K. We want to calrify that we have introduced them at Section 2.1 and 2.2.
>
> **Q2:** **Unclear statement in p4: accounts for … and can be any types of … ().**
>
> As proved in the foundational contrastive representation learning work [1], the density ratio function $f(\mathbf{y} _{0},\mathbf{x})=\frac{q(\mathbf{y} _{0}|\mathbf{x})}{q(\mathbf{y} _{0})}$ can be any positive real-valued forms to support the contrastive training governed by the derived InfoNCE loss. The noise regression form $f(\mathbf{y} _{0},\mathbf{x})=\exp (-||\epsilon{}'-\epsilon _{\theta}(\cdot)||^{2} _{2}/\tau)$ proposed by this work is inspired by this fact. In $\underline{\text{Section 3.2 and Appendix A.9.2 of the revised paper}}$, we provide more detailed discussions on this density ratio function design.
>
> **Q3:** **Unclear statement in p5: replacing point-wise attention by channel-wise attention.**
>
> Actually, previous DiT-based diffusion forecasters such as LDT and TimeDiT place attention mechanism along the time dimension to capture the temporal dependencies among each time points, which we call it point-wise attention. While we put the attention module along the channel axis to represent the cross-channel temporal correlations, which we call it channel-wise attention. More clear discussions on these two different DiT adaptations can be found in $\underline{\text{Section 3.1, paragraph 3 of the revised paper}}$.
>
> **Q4:** **p6 \l 275: both a positive sample.**
>
> We have addressed the typo in the revised paper.
>
> **Q5:** **p7 \l 327: efficiency.**
>
> We have corrected this word mistake in the revised paper.
>
> **Q6:** **Rendering Figure. 5 more clear and meaningful to show the positive effect of contrastive refinement.**
>
> We have depicted a new version of Figure. 5 to exhibit the positive effect of contrastive learning more clearly. As shown in $\underline{\text{Fig. 5 of the revised paper}}$, where we display MSE, MAE, CRPS and CRPS\_sum incurred by assigning various values to the contrastive weight $\lambda$. Fig. 5 aims to illustrate two points: 1) how different weights (i.e. $\lambda=0.01, 0.005, 0.001, 0.0005, 0.0001$) of the proposed contrastive loss in Eq. 6 can influence the diffusion forecasting capacity. 2) Adding contrastive training to naive diffusion predictive learning can actually enhance the forecasting capability, as the four metric outcomes of $\lambda=0$ (i.e. without contrastive refinement) is significant larger than those of $\lambda>0$ (i.e. imposing different degrees of contrastive learning).
>
> [1] Aaron van den Oord, Yazhe Li, and Oriol Vinyals. Representation learning with contrastive predictive coding. arXiv preprint arXiv:1807.03748, 2018.
>
> *We sincerely appreciate your positive feedback again and hope our response can address your concerns. If you have any further questions, please feel free to ask.*

---

> > ### Comment · Reviewer_GGe5 · 2024-11-26
> > **Reply**
> >
> > Thank you for your reply which answers my questions.

---

> > > ### Author Response · Authors · 2024-11-27
> > > **Response to Reviewer GGe5**
> > >
> > > We greatly appreciate the reviewer for your careful reading and positive support for this work! Sincerely thank you again and welcome to ask any further questions you are interested in.

---

### Official Review · Reviewer_m6S8 · 2024-11-04

**Soundness:** 2
**Presentation:** 2
**Contribution:** 2
**Rating:** 3
**Confidence:** 4

**Summary:**

The paper presents CCDM (Channel-aware Contrastive Conditional Diffusion Model), a novel approach for multivariate probabilistic time series forecasting. CCDM introduces two key innovations: a channel-aware conditional denoising network and a denoising-based temporal contrastive refinement. The channel-aware architecture incorporates channel-independent dense encoders and channel-mixing diffusion transformers to efficiently capture intra-variate and inter-variate temporal correlations. This design allows for better scalability across different prediction horizons and channel numbers. The denoising-based temporal contrastive refinement explicitly maximizes the prediction-related mutual information between past observations and future forecasts, complementing the standard denoising diffusion training.

The authors provide theoretical insights into the benefits of their approach from both neural mutual information and distribution generalization perspectives. They claim that CCDM exhibits superior forecasting capability compared to current state-of-the-art diffusion forecasters. The paper also presents a proposition on the upper bound of forecasting error for conditional diffusion models, linking the efficacy of conditional diffusion forecasters to the step-wise noise regression accuracy of the trained denoising network on unknown test time series.

**Strengths:**

The proposed CCDM (Channel-aware Contrastive Conditional Diffusion Model) introduces two key innovations that address important challenges in the field. First, it employs a channel-aware conditional denoising network that efficiently captures both intra-variate and inter-variate temporal correlations. This architecture, combining channel-independent dense encoders and channel-mixing diffusion transformers, allows for better scalability across different prediction horizons and channel numbers. Second, CCDM implements a denoising-based temporal contrastive refinement that explicitly maximizes the prediction-related mutual information between past observations and future forecasts. This approach complements the standard denoising diffusion training and improves forecasting accuracy and generalization on unknown test data.

Empirically, CCDM demonstrates superior forecasting capability compared to state-of-the-art diffusion forecasters. The method is also designed to be efficient, end-to-end, and seamlessly integrated with original simplified denoising diffusion optimization.

**Weaknesses:**

1. Lack of novelty and motivation. The paper's core contributions lack novelty in several aspects. The proposed Diffusion loss and mutual information concepts have been previously explored in the literature. Furthermore, the integration of contrastive learning loss within the Diffusion architecture seems arbitrary, and adding mutual information unnecessarily increases the complexity of the approach. While the authors attempt to justify their design choices with theoretical arguments, the discussion in lines 310-326 largely reiterates existing work (Tsai et al., 2020). The authors' decision to use KL loss approximation, necessitated by their inability to compute the upper bound in Proposition 1, is a standard approach in the field. This makes it difficult to justify the relationship between the two types of loss functions as a meaningful contribution.


2. Presentation: The paper's visual presentation suffers from several clarity issues. Figure 1's bidirectional arrows create ambiguity regarding the complementary learning pathways, and it's unclear when the losses described in equation 5 and 6 are applied. Figure 2 compounds this confusion by failing to clearly explain the implementation and timing of negative time series augmentation. Additionally, the distinction between channel-independent and channel-dependent approaches is not well articulated. The inclusion of CiDM appears redundant given that the Diffusion Transformer depicted in Figure 3 inherently supports multivariate input processing. This design choice is particularly questionable given the absence of any ablation studies to justify its necessity or demonstrate its effectiveness. These presentation issues significantly impact the reader's ability to understand the proposed architecture and its implementation details.

3. The experimental results lack clarity and comprehensive evaluation in several key aspects. First, although the proposed design makes no explicit claims about improving either short-term or long-term performance, the paper would benefit from including short-term prediction results using CSDI's benchmark datasets for better comparison. Second, the two-stage training process remains ambiguous throughout the paper. While line 289 claims to "improve the conditional denoiser behaviors in out-of-distribution (OOD) regions," no empirical evidence is provided to support this OOD performance improvement. Given the inherent uncertainty in Diffusion models, the absence of mean and variance statistics for the experimental results is a significant oversight. This statistical ambiguity is particularly problematic in Table 8, where the ablation results showing different components' contributions appear inconsistent and potentially arbitrary, making it difficult to draw meaningful conclusions about the model's effectiveness.

**Questions:**

1. Lack of novelty in motivation:
	- The introduction of diffusion loss and mutual information is not novel.
	- The use of contrastive learning loss in a diffusion architecture is not intuitive.
	- Introducing mutual information makes the problem more complex.
	2.	Insufficient theoretical justification:
	- The Theoretical Insights section (lines 310-326) appears to largely repeat (Tsai et al., 2020).
	- Proposition 1’s upper bound cannot be calculated in practice, which is why KL loss is used as an approximation. This is common knowledge and fails to effectively link the use of two different losses.
	3.	Unclear figures and methodology:
	-  Figure 1’s bidirectional arrows are confusing, raising questions about when to use the losses in equations 5 and 6.
	- Figure 2 adds to this confusion, leaving unclear when and how negative time series augmentation is used.
	- In Figure 3, it’s unclear why CiDM is necessary when Diffusion Transformer seems can naturally handle multivariate input. No ablation study is provided to justify CiDM's important.
	4.	Experimental results are difficult to follow:
	- Given that the proposed design doesn’t specifically aim to improve short-term or long-term performance, the reviewer suggests including results from the CSDI dataset (short-term prediction).
	- The two-stage training process is not clearly explained.
	- Despite claiming improved performance in out-of-distribution (OOD) regions (line 289), no evidence is provided to support this.
	- Given diffusion models’ inherent uncertainty, providing mean and variance of the data is crucial. This issue is particularly noticeable in Table 8, where different components’ contributions to model performance appear random.
	5.	Incomplete evaluation:
	- The paper lacks results on short-term prediction datasets (like those used in CSDI).
	- There’s no clear demonstration of OOD performance improvement.
	- The absence of mean and variance data for the uncertainty inherent in diffusion models is a significant omission.

---

> ### Author Response · Authors · 2024-11-23
> **Response to Reviewer m6S8 (Part 1)**
>
> *Thank you very much for your careful review and constructive suggestions! Hope our response can address your questions and concerns.*
>
> **W1.1\&Q1:** **Concerns on novelty and contribution of combing contrastive learning with denoising diffusion.**
>
> Here, we would like to reexplain and clarify the core motivations and contributions of the proposed contrastive denoising diffusion forecaster CCDM. It is an under-explored issue on how to enhance the utilization efficiency of the implicit temporal predictive information hidden in limited historical time series. The auxiliary training methods leveraged by existing diffusion forecasters need to expose prior knowledge on task-specific temporal properties and can not be consistent with standard step-wise denoising training. To this end, we propose to devise a complementary temporal contrastive learning scheme to improve diffusion training efficiency and explicitly maximize the prediction-related mutual information between generated forecasts and past observations.
>
> Our inspiration to integrate contrastive learning with denoising diffusion models come from two perspectives: 1) [1] has proved that combining both reconstruction-based generative learning and information-theoretic contrastive learning can enhance the usefulness of discovered task-specific representations. 2) [2] has shown that diffusion models can benefit from conducting more OOD evaluations on low-probability samples. Besides, we specifically design a noise regression form for the contrastive loss to seamlessly align with standard denoising diffusion training, which thus is efficient to implement. We are one of the first to integrate time series contrastive learning with diffusion forecasting for more efficient exploitation of useful predictive information.
>
> **W1.2\&Q2:** **Concerns on Theoretical insights and Proposition 1 in Section 3.3.**
>
> Here, we want to interpret the role of the theoretical insights subsection more clearly. In this part, we aim to demonstrate how naive time series diffusion models can benefit from auxiliary contrastive learning to approximate the target predictive distribution more faithfully. [1] and [2] demonstrate two disparate perspectives to motivate combining auxiliary contrastive learning with naive denoising diffusion. We further adapt these claims for the specific time series diffusion forecasting task as clarified in **W1.1\&Q1** and **W3.3\&Q4.3\&Q5.2** response. We have presented additional theoretical interpretations on the benefits of information-theoretic contrastive learning in $\underline{\text{Appendix A.1.2 and Theoretic Insights Section of the revise paper}}$.
>
> As the reviewer suggested, the upper bound of probabilistic diffusion forecasting can not be calculated in exact closed-form. However, we want to clarify that utilizing the KL-divergence can represent such upper
> bound. Moreover, it reflects that the forecasting capacity of conditional diffusion models is closely associated with the step-wise noise regression accuracy of trained denoising network on OOD test time series. We believe this is still important for guiding CCDM training. In this regard, resorting to temporal contrastive learning or other auxiliary training regimes is significant to boost conditional denoiser behaviors and the ultimate prediction outcomes.
>
> **W2.1\&Q3.1\&Q3.2:** **Improvement for Figure 1,2,3.**
>
> We apologize for the ambiguities in Figure 1,2,3. For Figure 1, we have deleted the bidirectional arrows and let two branches represent two complementary lower bounds for predictive mutual information. For Figure 2, the detailed negative time series augmentation methods can be found in $\underline{\text{Appendix A.9.1 of the revised paper}}$, where we discuss about four kinds of negative construction methods and validate their effectiveness for contrastive learning. For Figure 3, the channel-independence and channel-dependence are articulated in $\underline{\text{Section 3.1 of the revised paper}}$, where we also talk about the distinctions from other DiT-based time series diffusion models. The necessity of CiDM modules has been verified in **W2.2\&Q3.3** response below.

---

> ### Author Response · Authors · 2024-11-23
> **Response to Reviewer m6S8 (Part 2)**
>
> **W2.2\&Q3.3:** **Lack of ablation study on the necessity of CiDM components in DiT blocks.**
>
> We leverage the channel-independent dense modules (CiDMs) in TiDE [3] to capture the univariate temporal dependencies along each channel. TiDE has verified that MLP-based CiDM with residual connections is an excellent structure to represent dynamic temporal variations. To verify its necessity in the proposed channel-centric denoising network, we replace it by a normal channel-independent linear encoder as shown in iTransformer [4], and report the respective results on a short-term forecasting setup in CSDI below:
>
> | Methods | Exchange |  |  |  | Electricity |  |  |  |
> | :---: | :---: | :---: | :---: | :---: | :---: | :---: | :---: | :---: |
> |  | MSE | MAE | CRPS | CRPS\_sum | MSE | MAE | CRPS | CRPS\_sum |
> | CCDM | **0.0309** | **0.1173** | **0.0917** | **0.5246** | **0.0881** | **0.1780** | **0.1325** | **9.9455** |
> | CCDM-w/o CiDM | 0.0323 | 0.1205 | 0.0983 | 0.5576 | 0.1067 | 0.1998 | 0.1627 | 12.1184 |
>
>
> We can see that if CiDM is replaced by by ordinary MLP entices, a moderate and consistent decline is observed for prediction accuracy. It reflects that adding residual shortcuts to channel-independent MLP encoders can indeed boost the expressivity for dynamic temporal variations, and verifies the virtue of residual CiDM modules in TiDE [3] again. We have incorporated this CiDM ablation study in $\underline{\text{Appendix A.8.1 and Table 8 of the revised paper}}$.
>
> **W3.1\&Q4.1\&Q5.1:** **Lack of short-term forecasting results as shown in CSDI.**
>
> We verify the capability of CCDM on short-term probabilistic forecasting at which previous diffusion forecasters are displayed to be adept. As you suggested, we follow the same short-term setting in CSDI with lookback window of 168 and prediction horizon of 24. We report the short-term prediction results below:
>
> | Methods | Exchange |  |  |  | Electricity |  |  |  |
> | :---: | :---: | :---: | :---: | :---: | :---: | :---: | :---: | :---: |
> |  | MSE | MAE | CRPS | CRPS\_sum | MSE | MAE | CRPS | CRPS\_sum |
> | CCDM | **0.0309** | **0.1173** | **0.0917** | **0.5246** | **0.0881** | **0.1780** | **0.1325** | **9.9455** |
> | CSDI | 0.0704 | 0.1774 | 0.1393 | 0.7714 | 0.1117 | 0.2028 | 0.1580 | 13.4852 |
> | TimeDiff | 0.0313 | 0.1257 | 0.1257 | 0.6857 | 0.1285 | 0.2512 | 0.2509 | 19.0025 |
> | TimeDiT | 0.0657 | 0.1685 | 0.1252 | 0.7196 | 0.1066 | 0.1965 | 0.1507 | 12.9503 |
> | LDT | 0.0656 | 0.1616 | 0.1125 | 0.6750 | 0.0998 | 0.1859 | 0.1473 | 12.7360 |
>
>
> We can clearly observe that CCDM consistently outperforms other baselines on this short-term forecasting setup, which further validate the superior forecasting capacity of CCDM. We have incorporated this short-term comparisons in $\underline{\text{Appendix A.8.1 and Table 8 of the revised paper}}$.
>
> **W3.2\&Q4.2:** **The two-stage training process is not clearly explained.**
>
> We apologize for unclear description for such two-stage training method. As introducing extra negative samples to every training batch and iteration will induce additional time overhead. When the data volume is extremely large, the training time consumption will be prohibitive. Hence, we propose a two-stage training method to reduce the training costs on large-scale datasets like Weather, ECL and Traffic. Concretely, we first train a naive low-cost diffusion model by the standard denoising loss in 100 epochs, and then proceed to train such intermediate model by the auxiliary contrastive loss in extra 20 to 30 epochs. $\underline{\text{Table 1, 8, 9, 10 of the revised paper}}$ illustrated that such two-stage strategy with less training costs can attain satisfactory forecasting outcomes as well.

---

> ### Author Response · Authors · 2024-11-23
> **Response to Reviewer m6S8 (Part 3)**
>
> **W3.3\&Q4.3\&Q5.2:** **No empirical evidence to support the OOD performance improvement.**
>
> The OOD generalization issue, a.k.a temporal distribution shift is an imperative and fundamental issue in time series forecasting. Since the real-world time series are always non-stationary and stochastic, the intrinsic temporal dependencies and inter-variate correlations can alter over time, which will render the training distribution different from the unknown test distribution. Previous works [5], [6], [7], [8], [9] have statistically demonstrated the OOD phenomenon is ubiquitous in real-world time series datasets (e.g. six datasets utilized in our work), and incorporating auxiliary learning strategies to combat with test distribution shift can enhance the forecasting performance, like the invariant learning in [6] and adversarial training in [7].
>
> In this work, we find that the proposed complementary contrastive learning can also improve the forecasting capability of naive time series diffusion models on OOD test time series data. In Fig. 3 of [5], we can observe that the distribution of unknown test time series over each step is hard to overlap with that of collected training sets. We verify this by comparing the proposed contrastive diffusion CCDM with three plain denoising diffusion forecasters purely trained by step-wise noise regression, including CCDM-w/o contrastive, CSDI, TimeGrad. We choose six real-world datasets from the field of Energy, Weather, Finance and Traffic, which have been shown to have the OOD issue in [5]. We present the comparison results below:
>
> | Models | CCDM |  | CCDM-w/o contrastive |  | CSDI |  | TimeGrad |  |
> | :---: | :---: | :---: | :---: | :---: | :---: | :---: | :---: | :---: |
> | Metrics | MSE | CRPS | MSE | CRPS | MSE | CRPS | MSE | CRPS |
> | ETTh1 | **0.4636** | **0.3533** | 0.5508 | 0.3889 | 1.0642 | 0.5941 | 1.1804 | 0.6144 |
> | Exchange | **0.4659** | **0.3271** | 0.4966 | 0.3403 | 0.7649 | 0.5051 | 1.8398 | 1.0189 |
> | Weather | $\underline{\text{0.3325}}$ | $\underline{\text{0.2507}}$ | 0.3816 | 0.2695 | **0.3065** | **0.2344** | 0.5001 | 0.3274 |
> | Appliance | **0.9303** | **0.5385** | 1.0220 | 0.5818 | 1.0478 | 0.5638 | 1.9637 | 0.9000 |
> | Electricity | **0.1771** | **0.2027** | 0.1887 | 0.2144 | 0.4012 | 0.3069 | 0.3989 | 0.3330 |
> | Traffic | **0.8062** | **0.3644** | 0.8439 | 0.4130 | 1.4577 | 0.6104 | 1.1935 | 0.4650 |
>
>
> We can see that contrastive diffusion model CCDM can achieve the state-of-art results across various datasets, which reflects that adding the proposed contrastive training can improve the forecasting capacity on OOD test time series. More detailed outcomes over diverse prediction horizons and other baseline models can be found in $\underline{\text{Table 1, 2, 8, 9, 10 of the revised paper}}$.
>
> **W3.4\&Q4.4\&Q5.3:** **The absence of mean and variance statistics to quantify uncertainties inherent in diffusion models.**
>
> As the reviewer suggested, time series diffusion models will randomly generate a group of possible forecasting trajectories according to its approximated predictive distribution, which constitute a prediction interval. In probabilistic forecasting area, both the deterministic prediction accuracy and probabilistic prediction uncertainties should be well quantified. Adhering to many other canonical time series generative forecasting models [10], [11], we also utilize MSE, MAE, CRPS and CRPS\_sum metrics to assess the quality of diffusion generated prediction intervals. In detail, the MSE (Mean Squared Error) and MAE (Mean Absolute Error) are employed to quantify the mean difference between the obtained median forecast and true target, which can indicate the mean statistics as you mentioned. The CRPS (Continuous Ranked Probability Score) and CRPS\_sum are employed to characterize the divergence between the generated prediction uncertainties and the real observed time series distribution, which can indicate the variance statistics as you mentioned. And Table 8 is easy to interpret, since auxiliary contrastive training strategy and channel-aware DiT denoising network are totally two disparate designs to improve time series diffusion forecasters, they will exert different influences on the forecasting outcomes. We have updated the descriptions of evaluation metrics in $\underline{\text{Appendix A.5 of the revised paper}}$, which aims to quantify the accuracy and uncertainties of diffusion generated forecasts.

---

> ### Author Response · Authors · 2024-11-23
> **Response to Reviewer m6S8 (Part 4)**
>
> [1] Yao-Hung Hubert Tsai, Yue Wu, Ruslan Salakhutdinov, and Louis-Philippe Morency. Self-supervised learning from a multi-view perspective. In International Conference on Learning Representations, 2020.
>
> [2] Yunshu Wu, Yingtao Luo, Xianghao Kong, Evangelos E Papalexakis, and Greg Ver Steeg. Your diffusion model is secretly a noise classifier and benefits from contrastive training. arXiv preprint arXiv:2407.08946, 2024.
>
> [3] Abhimanyu Das, Weihao Kong, Andrew Leach, Shaan K Mathur, Rajat Sen, and Rose Yu. Long-term forecasting with tide: Time-series dense encoder. Transactions on Machine Learning Research, 2023a.
>
> [4] Yong Liu, Tengge Hu, Haoran Zhang, Haixu Wu, Shiyu Wang, Lintao Ma, and Mingsheng Long. itransformer: Inverted transformers are effective for time series forecasting. In The Twelfth International Conference on Learning Representations, 2023.
>
> [5] Kim T, Kim J, Tae Y, et al. Reversible instance normalization for accurate time-series forecasting against distribution shift. International Conference on Learning Representations. 2021.
>
> [6] Liu H, Kamarthi H, Kong L, et al. Time-Series Forecasting for Out-of-Distribution Generalization Using Invariant Learning. Forty-first International Conference on Machine Learning. 2024.
>
> [7] Lu W, Wang J, Sun X, et al. Out-of-distribution Representation Learning for Time Series Classification. The Eleventh International Conference on Learning Representations. 2023.
>
> [8] Chen M, Shen L, Fu H, et al. Calibration of Time-Series Forecasting: Detecting and Adapting Context-Driven Distribution Shift[C]. Proceedings of the 30th ACM SIGKDD Conference on Knowledge Discovery and Data Mining. 2024: 341-352.
>
> [9] Liu Y, Wu H, Wang J, et al. Non-stationary transformers: Exploring the stationarity in time series forecasting. Advances in Neural Information Processing Systems, 2022, 35: 9881-9893.
>
> [10] Kashif Rasul, Calvin Seward, Ingmar Schuster, and Roland Vollgraf. Autoregressive denoising diffusion models for multivariate probabilistic time series forecasting. In International Conference on Machine Learning, pp. 8857–8868. PMLR, 2021.
>
> [11] Xinyao Fan, Yueying Wu, Chang Xu, Yuhao Huang, Weiqing Liu, and Jiang Bian. Mg-tsd: Multi-granularity time series diffusion models with guided learning process. arXiv preprint arXiv:2403.05751, 2024.
>
> *We sincerely hope our responses address your concerns. If you have any further questions, please feel free to ask. Thank you.*

---

> > ### Comment · Reviewer_m6S8 · 2024-11-27
> > **Thanks for the response and there is remaining concerns**
> >
> > Thank you for considering my responses. I have carefully reviewed your replies, and there are three significant issues that remain unaddressed. The reviewer will provide their feedback on the other aspects if needed:
> >
> > 1. Regarding the novelty claim: The statement "enhancing utilization efficiency of implicit temporal predictive information in limited historical time series" is not particularly novel. This concept underlies most existing work on historical data-based prediction. Furthermore, the integration of contrastive learning with diffusion models appears disconnected from the visual representations in Figures 1-3, which fail to demonstrate this supposedly novel combination in a coherent structural framework.
> > 2. Regarding Proposition 1: My original concern about the practical calculation of the upper bound and the use of KL loss as an approximation was not adequately addressed. Your response that "KL-divergence can represent such upper bound" seems just repeat my concern.
> > 3. Regarding uncertainty quantification: Given your use of CRPS, it is essential to include mean and variance metrics to properly evaluate CCDM's uncertainty performance. Without these statistical measures, the model's probabilistic forecasting capabilities cannot be properly assessed.

---

### Author Response · Authors · 2024-11-23
**Overview of Responses and Revisions**

We sincerely appreciate all the reviewers for your valuable suggestions and insightful comments, which are really beneficial and instructive to improve our work. We have endeavored to address your concerns and existing weakness of this paper. Here, we summarized our major responses and revisions as follows:

- **Further clarifications on key challenges and contributions.** We provide more lucid explanations on the motivations and novelties of the proposed denoising-based contrastive learning and channel-aware architecture design to improve time series diffusion forecasting. We also present more clear analysis on the benefits of the complementary contrastive training both empirically and theoretically.
- **Additional baseline models, forecasting setups and evaluation metrics.** We incorporate short-term prediction setups and compare with other up-to-date non-diffusion-based models for more comprehensive evaluation. We utilize MAE and CRPS\_sum metrics and showcase more prediction intervals for better probabilistic analysis.
- **More sufficient analysis on denoising-based contrastive learning.** We demonstrate deeper analysis on the contrastive loss design, negative time series augmentation methods and training time overhead.
- **More architectural ablation studies.** We compare with other DiT-based models and conduct more component ablations to attest the necessity of channel-centric conditional denoiser design.
- **Experimental results refinement.** We ameliorate the contrastive training implementation and improve original prediction results presented in Table 1, 2.
- **Writing improvement.** We correct the writing typos and enhance the presentation in original submission.

We genuinely thank again for the careful comments and valuable suggestions from all reviewers, which are very helpful to promote the paper quality and impact. We are really willing to answer any further questions. Looking forward to the reviewers' feedback.

---

### Meta-Review · Area_Chair_UD2V · 2024-12-18

**Metareview:**

The paper proposes a new channel-aware diffusion model with contrastive losses for multivariate probabilistic forecasting. Adding contrastive learning to diffusion model seems like an interesting (if counter-intuitive) idea, and the channel-aware denoising architecture  seems reasonable (even though it is incremental incremental over prior work, as some reviewers noted ). However, a key concern of most of the reviewers (which the AC agrees with) were several weaknesses in the experimental section - ranging from lack of error bars, missing state-of-the-art baselines (especially on non-diffusion models ), unclear quality gains over the baselines on some datasets/metrics, and missing discussion on time complexity. The authors, to their credit, did significantly improve on the experimental section during the rebuttal process, and added more baselines and ablation studies. Nevertheless, the paper would greatly benefit from a exhaustive revision on the experiments, with a comprehensive set of baselines and more insightful ablation studies (especially on the contrastive learning piece).  This is interesting work, and I would urge the authors to resubmit the paper again with a stronger experimental section.

**Additional Comments On Reviewer Discussion:**

Several reviewers had questions on the novelty of the channel-aware denoising architecture and around the motivations for the contrastive learning piece, especially since the initial submission did not clearly explain these aspects. Furthermore, many of the reviewers expressed strong concerns  around the weak experimental evaluations - lack of error bars, missing state-of-the-art baselines, unclear quality gains over the baselines on some datasets/metrics, and missing discussion on time complexity. During the rebuttal, the authors added clearer explanations motivating their architectural choices, and more baselines, error bars, ablation studies, and short-term prediction experiments. However, several reviewers (including 4FVe  and D78R) felt that the updated experiments still left several questions unexplained, including  unclear results on multi-channel datasets, weaker performance on some baselines/datasets, and inadequate comparison against some existing methods.

---

### Decision · Program_Chairs · 2025-01-22

Reject